# SGD Finds then Tunes Features in Two-Layer Neural Networks with Near-Optimal Sample Complexity: A Case Study in the XOR problem

**Margalit Glasgow** [*]
Department of Computer Science
Stanford University
Stanford, CA 94305, USA
{mglasgow}@stanford.edu

## Abstract

In this work, we consider the optimization process of minibatch stochastic gradient descent (SGD) on a 2-layer neural network with data separated by a quadratic ground truth function. We prove that with data drawn from the $d$-dimensional Boolean hypercube labeled by the quadratic "XOR" function $y = -x_i x_j$, it is possible to train to a population error $o(1)$ with $d\mathrm{polylog}(d)$ samples. Our result considers simultaneously training both layers of the two-layer-neural network with ReLU activations via standard minibatch SGD on the logistic loss. To our knowledge, this work is the first to give a sample complexity of $\tilde{O}(d)$ for efficiently learning the XOR function on isotropic data on a standard neural network with standard training. Our main technique is showing that the network evolves in two phases: a *signal-finding* phase where the network is small and many of the neurons evolve independently to find features, and a *signal-heavy* phase, where SGD maintains and balances the features. We leverage the simultaneous training of the layers to show that it is sufficient for only a small fraction of the neurons to learn features, since those neurons will be amplified by the simultaneous growth of their second layer weights.

## 1 Introduction

Stochastic gradient descent (SGD) is the primary method of training neural networks in modern machine learning. Despite the empirical success of SGD, there are still many questions about why SGD is often able to efficiently find good local minima in the non-convex optimization landscape characteristic of training neural networks.

A growing body of work aims to theoretically understand the optimization dynamics and sample complexity of learning natural classes of functions via SGD on neural networks. A particularly well-understood regime in this regard is the neural tangent kernel (NTK)(Jacot et al., 2021a), where the network only moves a small distance from its initialization. However, in many cases, the NTK provably requires a poor sample complexity to generalize (Abbe et al., 2022).

More recent work aims to prove convergence guarantees for SGD on neural networks with tight sample complexity guarantees. A natural test-bed for this, which has garnered a lot of attention, is learning target functions that are inherently low-dimensional, depending only on a constant number of dimensions of the data (Chen & Meka, 2020; Chen et al., 2020; Nichani et al., 2022; Barak et al., 2022; Bietti et al., 2022; Mousavi-Hosseini et al., 2022; Refinetti et al., 2021; Abbe et al., 2021a; 2022; 2023). Such functions, often called *sparse* or *multi-index* functions, can be written as $f(x) := g(Ux)$, where $U \in \mathbb{R}^{k \times d}$ has orthogonal rows, and $g$ is a function on $\mathbb{R}^k$. Many works have shown that learning such target functions via SGD on neural networks is possible in much fewer samples than achievable by kernel methods (Chen et al., 2020; Bai & Lee, 2019; Damian et al., 2022; Abbe et al., 2021a; 2022; 2023). The results in these papers apply to a large class of ground

---

[*] https://web.stanford.edu/ mglasgow/index.html.

truth functions, and have greatly enhanced our understanding of the sample complexity necessary for learning via SGD on neural networks.

The limitation of the aforementioned works is that they typically modify the SGD algorithm in ways that don't reflect standard training practices, for example using layer-wise training, changing learning rates, or clipping. While providing strong guarantees on certain subclasses of multi-index functions, such modifications may limit the ability of SGD to learn broader classes of multi-index functions with good sample complexity. We discuss this more in the context of related work in Section 1.1.

The goal of this paper is to show that for a simple but commonly-studied problem, standard minibatch SGD on a two-layer neural network can learn the ground truth function in near-optimal sample complexity. In particular, we prove in Theorem 3.1 that a polynomial-width ReLU network trained via online minibatch SGD on the logistic loss will classify the boolean XOR function $f(x) := -x_i x_j$ with a sample complexity of $\tilde{O}(d)$.[1] We study the XOR function because it one of the simplest test-beds for a function which exhibits some of the core challenges of analyzing SGD on neural networks: a random initialization is near a saddle point, and the sample complexity attainable by kernel methods is suboptimal (see further discussion in Section 1.1).

Despite its simplicity, the prior theoretical understanding of learning the XOR function via SGD on standard networks is lacking. It is well-known that the NTK requires $\Theta(d^2)$ samples to learn this function (Wei et al., 2019; Ghorbani et al., 2021; Abbe et al., 2023). Wei et al. (Wei et al., 2019) showed that $\tilde{O}(d)$ samples statistically suffice, either by finding the global optimum of a two-layer network, or by training an infinite-width network, both of which are computationally intractable. Similar guarantees of $\tilde{O}(d)$ are given by Bai et al. (Bai & Lee, 2019) and Chen et al. (Chen et al., 2020); however, such approaches rely on drastically modifying the network architecture and training algorithm to achieve a quadratic neural tangent kernel. Abbe et al. (Abbe et al., 2023) proves a sample complexity of $\tilde{O}(d)$ for the XOR problem, but uses an algorithm which assumes knowledge of the coordinate system under which the data is structured, and is thus not rotationally invariant. It is also worth noting that several works have studied the XOR problem with non-isotropic data, where the cluster separation grows to infinity (Frei et al., 2022; Ben Arous et al., 2022), in some cases yielding better sample complexities.

The main approach in our work is showing that while running SGD, the network naturally evolves in two phases. In the first phase, which we call the *signal-finding* phase, the network is small, and thus we can show that a sufficient fraction of the neurons evolve independently, similarly to how they would evolve if the output of the network was zero. Phase 1 is challenging because it requires moving away from the saddle near where the network is initialized, which requires super-constant time (here we use "time" to mean the number of iterations times step size). This rules out using the mean field model approach as in Mei et al. (Mei et al., 2018b; 2019), or showing convergence to a lower-dimensional SDE as in Ben Arous et al. (Ben Arous et al., 2022), which both break down after constant time when directly applied to our setting. [2]

After the signal components in the network have become large enough to dominate the remaining components, the network evolves in what we call the *signal-heavy* phase. In this phase, we show inductively that throughout training, the signal components stay significantly larger than their counterparts. This inductive hypothesis allows us to approximate the output of the network on a sample $x$ by its *clean* approximation, given by a network where all the non-signal components have been removed. Under this approximation, the dynamics of the network are easier to compute, and we can show that the signal components will grow and rebalance until all four of the clusters in the XOR problem have sufficiently small loss. The division into signal-finding and signal-heavy phases is similar to the two phases of learning in e.g. Arous et al. (2021).

Our Phase 2 analysis leverages the simultaneous training of both layers to show that the dominance of the signal components will be maintained throughout training. In particular, we show once individual neurons become signal heavy, their second layer weights become large, and thus a positive feedback

---

[1] We consider this near-optimal in the sense that for algorithms that are rotationally invariant $\tilde{\Theta}(d)$ samples are required. See Section G for details.

[2] Ben Arous et al. (2022) considers a setting of high-dimensional SGD where a *constant* number of summary statistics sufficient to track the key features of the SGD dynamics and the loss, which can only be applied to constant-width 2-layer neural networks. Their coupling between high-dimensional SGD and a low-dimension SDE holds for $\Theta(1)$ time, which is not enough time to learn the XOR function, which requires $\Theta(\log(d))$ time.

cycle between the first and second layer weights of that neuron causes it to grow faster than non-signal-heavy neurons. This allows us to maintain the signal-heavy inductive hypothesis. If we only trained the first layer, and all second layer weights had equal absolute value, then unless we have strong control over the balance of the clusters, it would be possible for the non-signal components to grow at a rate which is on the same order as the rate of the signal components (see Remark 4.3).

## 1.1 RELATED WORK

**Learning Multi-Index Functions via Neural Networks**  Most related to our work is a body of work aiming to understand the sample complexity of learning multi-index functions via SGD on neural networks Bietti et al. (2022); Refinetti et al. (2021); Chen et al. (2020); Abbe et al. (2021a; 2022; 2023); Damian et al. (2022); Barak et al. (2022); Daniely & Malach (2020); Mousavi-Hosseini et al. (2022); Nichani et al. (2022); Ge et al. (2017); Mahankali et al. (2023); Ba et al. (2022); Dandi et al. (2023). Such functions are typically studied in either the Gaussian data setting where $x \sim \mathcal{N}(0, I_d)$, or in the Boolean hypercube setting, where $x \sim \text{Uniform}(\{\pm 1\}^d)$. In both cases, we have $f(x) := g(Ux)$, where $U$ projects $x$ onto a lower dimensional space of dimension $k$, and $g$ is an arbitrary function on $k$ variables. In the Boolean setting, $U$ projects onto a subset of $k$ coordinates of $x$, so in the case of the XOR function we study, $k = 2$ and $g$ is a quadratic function.

Chen and Meka (Chen & Meka, 2020) showed when $k$ is constant, and $g$ is a degree-$D$ polynomial for constant $D$, there exists a polynomial-time algorithm which learns such multi-index functions on Gaussian covariates in $\tilde{O}(d)$ samples. Such algorithms can also be emulated in the same sample complexity via SGD on neural networks designed to emulate arbitrary Statistical Query algorithms (Abbe & Sandon, 2020; Abbe et al., 2021b), though these networks bear little similarity to standard neural networks used in practice.

The sample complexity of learning multi-index functions via SGD on standard neural networks is an open and active area of research. It is known that the neural tangent kernel (and more generally, kernel methods) require $\Omega(d^D)$ samples (Hsu, 2021). A line of work by Abbe et al. (Abbe et al., 2021a; 2022; 2023) has conjectured that the sample complexity required for SGD is $\tilde{\Theta}(d^{\max(L-1,1)})$, where $L$ denotes the "leap complexity", a measure of hierarchical structure upper bounded by $D$, and which equals 2 for the XOR function. If true, this conjecture would place the sample complexity of SGD on standard neural networks squarely between that of kernel methods and arbitrary polynomial-time algorithms. When $L = 1$, Abbe et al. (2022) showed via a mean-field analysis that is possible to learn with $\Theta(d)$ samples via layer-wise training, where the first layer is trained until it learns the subspace $U$, and then the second layer is trained as a linear model. For $L > 1$, Abbe et al. (2023) provided a layer-wise SGD algorithm achieving the conjectured complexity, but which assumes knowledge of the coordinate system under which the data is structured. This means the algorithm is not-rotationally invariant, barring the network from learning more general multi-index functions. Other works have also used layer-wise training to give similar results for subclasses of multi-index functions (Damian et al., 2022; Mousavi-Hosseini et al., 2022; Barak et al., 2022); Mousavi-Hosseini et al. (2022) studies a setting where $k = 1$ and $L = 1$, while Damian et al. (2022); Barak et al. (2022) study settings where $L \geq 2$, and use just a single gradient step on on the first layer, which requires $\Omega(d^L)$ samples. Numerous other works (Tan & Vershynin, 2019; Bietti et al., 2022; Wu et al., 2023; Arous et al., 2021) have made progress in the setting of single-index functions ($k = 1$) when $L > 1$. In some cases, the result achieve tight guarantees that depend on a quantity called the "information exponent" of $g$, which is *equivalent to the leap complexity* when $k = 1$, though these methods require training only a single neuron in $\mathbb{R}^d$. The recent work Mahankali et al. (2023) considers training a single-index target function with $k = 2$ and degree 4 on a 2-layer neural network via vanilla gradient descent, and shows a sample complexity of $O(d^{3+\epsilon})$, which improves over kernel methods.

The above discussion highlights a gap in our understanding when $k \geq 2$ and $L \geq 2$. Indeed, such a setting is challenging because it requires learning multiple neurons, and escaping one (or more) saddles (Abbe et al., 2023). For this reason, we believe the XOR function (with $k, L = 2$) is a good stepping stone for understanding the behaviour of SGD on neural networks for more general functions with $k \geq 2, L \geq 2$. Note that other works (Bai & Lee, 2019; Chen et al., 2020) have achieved a near-optimal sample complexity of $\tilde{O}(d)$ for the XOR problems; these works use a non-standard architecture and training algorithm which puts SGD into a quadratic NTK regime. While such a regime can often attain sample complexities beating the standard (linear) NTK, in general this method yields complexities of $\tilde{O}(d^{D-1})$, which is larger than the rate achieved by Abbe et al. (2022) whenever

$L = 1$ and $D \geq 3$. We emphasize that our work achieves the near-optimal sample complexity $\tilde{O}(d)$ with a standard two-layer neural network, trained with standard minibatch SGD.

We note that many more works have explored both empirically (eg. (Woodworth et al., 2020; Chizat et al., 2019)) and theoretically (eg.(Li et al., 2020; Allen-Zhu & Li, 2020; Suzuki & Akiyama, 2020; Telgarsky, 2022; Jacot et al., 2021b)) the sample-complexity advantages of "rich" SGD training over the "lazy" NTK regime.

**Simultaneous Training of Layers.** While many of the works mentioned above use layer-wise training algorithms, the standard empirical practice is to train all layers simultaneously. Several theoretical works explore this setting, uncovering implicit biases of ReLU (or other homogeneous) networks trained simultaneously (Wei et al., 2019; Chizat & Bach, 2020; Lyu & Li, 2019; Lyu et al., 2021; Maennel et al., 2018). Under a variety of assumptions, these works have related the solutions found via gradient descent to margin-maximizing solutions. A much finer understanding of the implicit bias of simultaneous training is provided for a line of work on diagonal neural networks (Pesme & Flammarion, 2023; Even et al., 2023).

## 1.2 ORGANIZATION OF PAPER

In Section 2, we describe the data and training model. In Section 3 we state our result. In Section 4, we overview the proof techniques. We conclude in Section 5. All proofs are in the Appendix.

## 1.3 NOTATION

For a vector $v$, we use $\|v\|$ to denote the $\ell_2$ norm, and $\|v\|_1$ to denote the $\ell_1$ norm. We use $\|M\|_2$ to denote the spectral norm of a matrix $M$. All big-O notation is with respect to $d \to \infty$, and we use $\tilde{O}$ to suppress log factors in big-O notation. $\omega(1)$ denotes growing to infinity with $d$. We use $\mathbb{S}^{d-1}(r)$ to denote the sphere of radius $r$ in $d$ dimensions, and $\mathbf{1}(\cdot)$ to denote the indicator variable of an event.

## 2 MODEL AND SETTING

### 2.1 DATA.

We study the setting where the data comes from the Boolean hypercube $x \sim \text{Uniform}(\{-1, 1\}^d)$, and the label $y$ is given by $y(x) = \text{XOR}(x_1, x_2) := -x_1 x_2$.

Note that with $\mu_1 := e_1 - e_2$, and $\mu_2 := e_1 + e_2$, we can model the distribution as

$$(x, y) = \begin{cases} (\mu_1 + \xi, 1) & w.p.\ 1/4 & (-\mu_1 + \xi, 1) & w.p.\ 1/4 \\ (\mu_2 + \xi, -1) & w.p.\ 1/4 & (-\mu_2 + \xi, -1) & w.p.\ 1/4 \end{cases},$$

where $\xi \sim \text{Uniform}(0^2 \times \{-1, 1\}^{d-2})$ so that $\xi \perp \{\mu_1, \mu_2\}$. We will often write

$$x = z + \xi,$$

where $z$ is the projection of $x$ onto the space spanned by $e_1$ and $e_2$, and $\xi$ is the projection of $x$ orthogonal to $e_1$ and $e_2$. We denote this distribution by $P_d$, and throughout, it is implicitly assumed that all probabilities and expectations over $x$ are for $x \sim P_d$.

**Remark 2.1.** *While for simplicity, we state our results for the setting where the data comes from an axis-aligned Boolean hypercube, and where ground truth depends on the first two dimensions, the minibatch SGD algorithm and the initialization of the network will be rotationally invariant. Thus all our results hold for a Boolean hypercube with any basis.*

### 2.2 TRAINING.

**Model.** We train both layers of a two-layer ReLU network with width $p$:

$$\frac{1}{p} \sum_{j=1}^{p} a_j \sigma(w_j^T x),$$

where $\sigma(\alpha) = \max(0, \alpha)$ is the ReLU function. We will use the variable $\rho := \frac{1}{p} \sum_{j=1}^{p} \mathbf{1}_{(w_j, a_j)}$ to denote the empirical distribution of the neurons and their second layer weights. Thus we denote

$$f_\rho(x) := \mathbb{E}_{w,a \sim \rho} a \cdot \sigma(w^T x),$$

We will often abuse notation and write probabilities and expectations using $w \sim \rho$, and use $a_w$ to denote its associated second layer weight. We note that it is not necessarily the case the second layer weight $a_w$ is a *function* of $w$; we do this for the convenience of not indexing each pair as $(w_j, a_j)$.

**Initialization.** We initialize the network with $w_j \sim \text{Uniform}(\mathbb{S}^{d-1}(\theta))$ for a scale parameter $\theta$, such that $\|w_j\| = \theta$. We initialize the second layer as $a_j = \epsilon_j \|w_j\|$, where $\epsilon_j \sim \text{Uniform}(\pm 1)$.

**Minibatch SGD.** We train using minibatch SGD on the logistic loss function

$$\ell_\rho(x) := -2 \log \left( \frac{1}{1 + \exp(-y(x) f_\rho(x))} \right),$$

and define the population loss $L_\rho := \mathbb{E}_{x \sim P} \ell_\rho(x)$. We will use the shorthand $\ell'_\rho(x)$ to denote the derivative of $\ell_\rho(x)$ with respect to $f_\rho(x)$:

$$\ell'_\rho(x) := -\frac{2y(x) \exp(-y(x) f_\rho(x))}{1 + \exp(-y(x) f_\rho(x))}.$$

We use $\rho_t$ to denote the empirical distribution of the $p$ neurons $(w^{(t)}, a_w^{(t)})$ at iteration $t$. At each step, we perform the minibatch SGD update

$$w^{(t+1)} := w^{(t)} - \eta \nabla \hat{L}_\rho(w^{(t)}) \qquad a_w^{(t+1)} := a_w^{(t)} - \eta \nabla \hat{L}_\rho(a_w^{(t)}).$$

Here $\hat{L}_\rho = \frac{1}{m} \sum_{x^{(i)} \in M_t} \ell_\rho(x^{(i)})$ denotes the empirical loss with respect to a minibatch $M_t$ of $m$ random samples chosen i.i.d. from $P_d$ at step $t$, and for a loss function $L$ and a parameter $u$ in the network, $\nabla_u L := p \frac{\partial L}{\partial u}$ denotes the scaled partial derivative of the loss with respect to $u$, defined in particular for a neuron $(w, a_w)$, as follows: [3][4]

$$\nabla_w \hat{L}_\rho = \frac{1}{m} \sum_{x^{(i)} \in M_t} \frac{\partial}{\partial w} p \ell_\rho(x^{(i)}) = \frac{1}{m} \sum_{x^{(i)} \in M_t} a_w \ell'_{\rho_t}(x^{(i)}) \sigma'(w^T x^{(i)}) x^{(i)};$$

$$\nabla_{a_w} \hat{L}_\rho = \frac{1}{m} \sum_{x^{(i)} \in M_t} \frac{\partial}{\partial a_w} p \ell_\rho(x^{(i)}) = \frac{1}{m} \sum_{x^{(i)} \in M_t} \ell'_{\rho_t}(x^{(i)}) \sigma(x_i^T w).$$

## 3 MAIN RESULT

The following theorem is our main result.

**Theorem 3.1.** *There exists a constant $C > 0$ such that the following holds for any $d$ large enough. Let $\theta := 1/\log^C(d)$. Suppose we train a 2-layer neural network with minibatch SGD as in Section 2.2 with a minibatch size of $m \geq d/\theta$, width $1/\theta \leq p \leq d^C$, step size $d^{-C} \leq \eta \leq \theta$, and initialization scale $\theta$. Then for some $t \leq C \log(d)/\eta$, with probability $1 - d^{-\omega(1)}$, we have*

$$\mathbb{E}_{x \sim P_d}[\ell_{\rho_t}(x)] \leq (\log(d))^{-\Theta(1)}.$$

By setting $\eta = \theta$ and $m = d/\theta$, Theorem 3.1 states that we can learn the XOR function up to $\epsilon$ population loss in $\Theta(d \text{polylog}(d))$ samples and iterations on a polynomial-width network.

---

[3] Since the ReLU function is non-differentiable at zero, we define $\sigma'(0) = 0$.

[4] For convenience, we scale this derivative up by a factor of $p$ to correspond to the conventional mean-field scaling. If we didn't perform this scaling, we could achieve the same result by scaling the learning rate $\eta$.

Table 1: Summary of Notation used in Proof Overview and Proofs

| | | |
|---|---|---|
| $w_{\text{sig}} = \begin{cases} \frac{1}{2}\mu_1\mu_1^T w & a_w \geq 0 \\ \frac{1}{2}\mu_2\mu_2^T w & a_w < 0 \end{cases}$ | $w_{\text{opp}} = \begin{cases} \frac{1}{2}\mu_2\mu_2^T w & a_w \geq 0 \\ \frac{1}{2}\mu_1\mu_1^T w & a_w < 0 \end{cases}$ | $\begin{cases} w_{1:2} = w_{\text{sig}} + w_{\text{opp}} \\ w_\perp = w - w_{1:2} \end{cases}$ |
| $\gamma_\mu = f_\rho(\mu)y(\mu)$ | $\gamma_{\min} = \min_{\mu \in \{\pm\mu_1, \pm\mu_2\}} \gamma_\mu$ | $\gamma_{\max} = \max_{\mu \in \{\pm\mu_1, \pm\mu_2\}} \gamma_\mu$ |
| $g_\mu = |\ell'_\rho(\mu)|$ | $g_{\min} = \min_{\mu \in \{\pm\mu_1, \pm\mu_2\}} |\ell'_\rho(\mu)|$ | $g_{\max} = \max_{\mu \in \{\pm\mu_1, \pm\mu_2\}} |\ell'_\rho(\mu)|$ |

## 4 PROOF OVERVIEW

Throughout the following section, and in our proofs, we will use the following shorthand to refer to the components of a neurons $w$. We decompose $w = w_{1:2} + w_\perp$, where $w_{1:2}$ is the projection of $w$ in the direction spanned $e_1$ and $e_2$ (and equivalently by $\mu_1 = e_1 - e_2$ and $\mu_2 = e_1 + e_2$), and $w_\perp$ is the component of $w$ in the orthogonal subspace. We further decompose $w_{1:2} = w_{\text{sig}} + w_{\text{opp}}$ as follows:

$$w_{\text{sig}} = \begin{cases} \frac{1}{2}\mu_1\mu_1^T w & a_w \geq 0; \\ \frac{1}{2}\mu_2\mu_2^T w & a_w < 0. \end{cases} \qquad w_{\text{opp}} = \begin{cases} \frac{1}{2}\mu_2\mu_2^T w & a_w \geq 0; \\ \frac{1}{2}\mu_1\mu_1^T w & a_w < 0. \end{cases}$$

Intuitively, we want the neurons to grow in the $w_{\text{sig}}$ direction, but not the $w_{\text{opp}}$ direction; in a network achieving the maximum normalized margin, we will have $w = w_{\text{sig}}$ exactly, and $w_{\text{opp}} = w_\perp = 0$. We summarize this notation in Table 1, along with future shorthand we will introduce in this section.

The main idea of our proof is to break up the analysis of SGD into two main phases. In the first phase, the network is small, and thus we have (for most $x$) that the loss $\ell_\rho(x)$ is well approximated by a first order approximation of the loss at $f_\rho = 0$, namely

$$\ell_0(x; \rho) := -2\log(1/2) - y(x)f_\rho(x).$$

As long as this approximation holds, the neurons of the network evolve (approximately) independently, since $\ell'_0(x) := \frac{\partial \ell_0(x;\rho)}{\partial f_\rho(x)} = -y(x)$ does not depend on the full network $\rho$. We will show under this approximation that for many neurons, $\|w_{\text{sig}}\|$ grows exponentially fast. Thus we will run this first phase for $\Theta(\log(d)/\eta)$ iterations until for all four clusters $\mu \in \{\pm\mu_1, \pm\mu_2\}$, there exists a large set of neurons $S_\mu$ on which $w_{\text{sig}}^T \mu > 0$, and the "margin" from this set of neurons is large, i.e.

$$\tilde{\gamma}_\mu := \mathbb{E}_\rho[\mathbf{1}(w \in S_\mu)a_w\sigma(w^T\mu)] \gg \mathbb{E}_\rho\|w_\perp + w_{\text{opp}}\|^2. \tag{4.1}$$

In the Phase 2, we assume that Eq. 4.1 holds, and we leverage the dominance of the signal to show that (1) The signal components $w_{\text{sig}}$ grow faster that $w_{\text{opp}} + w_\perp$, and thus Eq. 4.1 continues to hold; and (2) SGD balances the signal components in the 4 cluster directions such that the margins $\tilde{\gamma}_\mu$ balance, and become sufficiently large to guarantee $o(1)$ loss.

We proceed to describe the analysis in the two phases in more detail. Full proofs are in the Appendix.

### 4.1 PHASE 1

In Phase 1, we approximate the evolution of the network at each gradient step by the gradient step that would occur for a network with output 0. The main building blocks of our analysis are estimates of the $L_0 := \mathbb{E}_x\ell_0(x; \rho)$ population gradients, and bounds on the difference $\nabla L_0 - \nabla L_\rho$.

$L_0$ **population gradients.** Since the primary objective of this phase is to grow the neurons in the signal direction, we sketch here the computation of the gradient $\nabla_{w_{1:2}}L_0$ in the subspace spanned by $\mu_1, \mu_2$. The remaining estimates of $\nabla L_0$ are simpler, and their main objective is to show that $\nabla_{w_\perp}L_0$ and $\nabla_{a_w}L_0$ are sufficiently small, such that $\|w_\perp\|$ doesn't change much throughout Phase 1, and $|a_w|$ stays approximately the same as $\|w\|$. For convenience, the reader may assume that $|a_w| = \|w\|$ exactly, which would hold if we took $\eta$ to 0 as in gradient flow.

For a data sample $x \sim \text{Rad}^d$, we denote $x = z + \xi$, where $z \in \text{Span}(\{\pm\mu_1, \pm\mu_2\})$, and $\xi \perp \text{Span}(\{\pm\mu_1, \pm\mu_2\})$. By leveraging the symmetry of the data distribution and the fact that $y(z) =$

$y(-z)$, we can compute

$$
\begin{aligned}
\nabla_{w_{1:2}} L_0 &= -a_w \mathbb{E}_{x=z+\xi} y(x) \sigma'(w^T x) z \\
&= -a_w \mathbb{E}_\xi \frac{1}{2} \mathbb{E}_z y(z) \left( \sigma'(w^T \xi + w^T z) - \sigma'(w^T \xi - w^T z) \right) z \\
&= -a_w \mathbb{E}_\xi \frac{1}{2} \mathbb{E}_z y(z) \mathbf{1}(|w^T z| \geq |w^T \xi|) \operatorname{sign}(w^T z) z \\
&= -\frac{1}{2} a_w \mathbb{E}_z y(z) \operatorname{sign}(w^T z) z \mathbb{P}_\xi[|w^T z| \geq |w^T \xi|] \\
&\approx -\frac{1}{2} a_w \mathbb{E}_z y(z) \operatorname{sign}(w^T z) z \mathbb{P}_{G \sim \mathcal{N}(0, \|w_\perp\|^2)}[G \leq |w^T z|] \\
&\approx -\frac{1}{2} a_w \mathbb{E}_z y(z) \operatorname{sign}(w^T z) z \sqrt{\frac{2}{\pi}} \frac{|w^T z|}{\|w\|}.
\end{aligned}
\tag{4.2}
$$

Here the two approximations come from the fact that $\xi$ has boolean coordinates and not Gaussian, and from an approximation of the Gaussian distribution, which holds whenever $\frac{|w^T z|}{\|w_\perp\|}$ is small. By taking the expectation over $z \in \{\pm\mu_1, \pm\mu_2\}$, the last line of Eq 4.2 can be shown to evaluate to

$$
-\frac{|a_w|}{\|w\|\sqrt{2\pi}} w_{\text{sig}} + \frac{|a_w|}{\|w\|\sqrt{2\pi}} w_{\text{opp}}.
\tag{4.3}
$$

Observe that near initialization, this gradient is quite small, since $\frac{\|w_{\text{sig}}\|}{\|w\|}$ is approximately $\frac{1}{\sqrt{d}}$ for a random initialization. Nevertheless, this gradient suggests that $w_{\text{sig}}$ will grow exponentially fast.

**Bounding the difference** $\nabla L_0 - \nabla L_\rho$. To bound $\|\nabla_w L_\rho - \nabla_w L_0\|_2$, first recall that

$$
\nabla_w L_0 - \nabla_w L_\rho = \mathbb{E}_x a_w (\ell_\rho'(x) - \ell_0'(x)) \sigma'(w^T x) x.
$$

Defining $\Delta_x := (\ell_\rho'(x) - \ell_0'(x)) \sigma'(w^T x)$, we can show using routine arguments (see Lemma D.2 for the details) that:

$$
\begin{aligned}
\|\nabla_w L_\rho - \nabla_w L_0\|_2 = |a_w| \|\mathbb{E}_x \Delta_x x\| &\leq |a_w| \sqrt{\mathbb{E}_x \Delta_x^2} \\
&\approx |a_w| \sqrt{\mathbb{E}_x f_\rho(x)^2} \\
&\lesssim |a_w| \mathbb{E}_\rho[\|a_w w\|] \approx \frac{|a_w|}{\text{polylog}(d)}.
\end{aligned}
\tag{4.4}
$$

While this deviation bound is useful for showing that $w_\perp$ doesn't move too much, this bound far exceeds the scale of the gradient in the $w_{\text{sig}}$, which is on the scale $\frac{|a_w|}{\sqrt{d}}$ near initialization. Fortunately, we can show in Lemma D.3 that the deviation is much smaller on the first two coordinates, namely,

$$
\|\nabla_{w_{1:2}} L_\rho - \nabla_{w_{1:2}} L_0\|_2 \leq |a_w| O(\log^2(d)) \left( \mathbb{E}_\rho[\|a_w w_{1:2}\|] + \mathbb{E}_\rho[\|a_w w\|] \frac{\|w_{1:2}\|}{\|w\|} \right)
\tag{4.5}
$$

Note that since near initialization $\|w_{1:2}\| \ll \|w\|$ for all neurons, this guarantee is much stronger than Eq. 4.4. In fact, since throughout this phase we can show that $a_w$ and $\|w\|$ change relatively little, staying at the scale $1/\text{polylog}(d)$, the approximation error in Eq. 4.5 is smaller than the gradient in the $w_{\text{sig}}$ direction (Eq. 4.3) whenever say $\|w_{\text{sig}}\| \geq 100 \mathbb{E}_\rho[\|a_w w_{1:2}\|]$, which occurs on a substantial fraction of the neurons.

Lemma D.3 is the most important lemma in our Phase 1 analysis. At a high level, it shows that the approximation error $\|\nabla_{w_{1:2}} L_\rho - \nabla_{w_{1:2}} L_0\|_2$ can be coupled with the growth of the signal, $-(\nabla_w L_0)^T \frac{w_{\text{sig}}}{\|w_{\text{sig}}\|}$. This is because we use a symmetrization trick with the pairs $z + \xi$ and $-z + \xi$ to show that both the error and the signal gradient only grow from samples $x = z + \xi$ where $|z^T w| \geq |\xi^T w|$.

In more detail, to prove Eq. 4.5, we also need to leverage the fact that for any $\xi \in \{\mu_1, \mu_2\}^\perp$ and $z \in \{\pm\mu_1, \pm\mu_2\}$, we have $|\ell_\rho'(\xi + z) - \ell_\rho'(\xi - z')| \leq 4p \mathbb{E}_\rho[\|a_w w_{1:2}\|]$, much smaller than we can

expect $|\ell'_\rho(x) - \ell'_0(x)|$ to be. Thus $|\Delta_{\xi+z} - \Delta_{\xi-z}| \leq 4p\mathbb{E}_\rho[\|a_w w_{1:2}\|]$ whenever $|\xi^T w| \geq |z^T w|$ (such that $\sigma'(w^T(\xi + z)) = \sigma'(w^T(\xi - z))$). Following the symmetrization trick in Eq. 4.2, we have

$$
\begin{aligned}
\left\|\frac{1}{a_w}\left(\nabla_{w_{1:2}}L_\rho - \nabla_{w_{1:2}}L_0\right)\right\| &= \|\mathbb{E}_x \Delta_x z\| \\
&= \|\mathbb{E}_\xi \mathbb{E}_z \Delta_{\xi+z} z\| \\
&= \frac{1}{2}\|\mathbb{E}_\xi \mathbb{E}_z (\Delta_{\xi+z} - \Delta_{\xi-z})z\| \\
&\leq 2\sqrt{2}\mathbb{E}_\rho[\|a_w w_{1:2}\|] + \sqrt{2}\mathbb{E}_\xi \mathbb{E}_z \mathbf{1}(|\xi^T w| \leq |z^T w|)|\Delta_x|.
\end{aligned}
$$

A careful computation comparing $w^T\xi$ to a Gaussian distribution then shows that

$$
\mathbb{E}_z \mathbf{1}(|\xi^T w| \leq |z^T w|)|\Delta_x| \approx \left(\mathbb{P}_x[|\xi^T w| \leq |z^T w|]\right)\left(\mathbb{E}_x|\Delta_x|\right) \lesssim \frac{\|w_{1:2}\|}{\|w\|}\mathbb{E}_\rho[\|a_w w\|].
$$

**Putting Phase 1 Together** The building blocks above, combined with standard concentration bounds on $\nabla \hat{L}_\rho$, suffice to show that a substantial mass of neurons will evolve according to Eq 4.3, leading to exponential growth in $w_{\text{sig}}$. After $\Theta(\log(d)/\eta)$ iterations, for these neurons, we can achieve $\|w_{\text{sig}}\| \gg \|w_\perp + w_{\text{opp}}\|$. Formally, we show the following for some $\zeta = 1/\text{polylog}(d)$:

**Lemma 4.1** (Output of Phase 1: Informal; See Lemma D.1 for formal version). *With high probability, for $\eta \leq \tilde{O}(1)$ and some $\zeta = 1/\text{polylog}(d)$, after some $T = \Theta(\log(d)/\eta)$ iterations of minibatch SGD, with $m = \tilde{\Theta}(d)$ samples in each minibatch, the network $\rho_T$ satisfies:*

1. $\mathbb{E}_{\rho_T}[\|w_\perp + w_{opp}\|^2] \leq \theta.$

2. *For each $\mu \in \{\pm\mu_1, \pm\mu_2\}$, on at least a $0.1$ fraction of all the neurons, we have $w_{sig}^T \mu > 0$, and $\|w_{sig}\|^2 \geq \zeta^{-1}\theta.$*

We remark that the analysis to prove Lemma 4.1 is somewhat subtle, since the tight approximation in Eq 4.2 breaks down when $\|w_{\text{sig}}\|$ approaches $\|w_\perp\|$. The details are given in Appendix D.

### 4.1.1 PHASE 2

The conclusion of Lemma 4.1 is a sufficient condition of the network to begin the second phase. In the second phase, we have that (for most $x$)

$$\ell'_\rho(x) \approx \ell'_\rho(z), \tag{4.6}$$

where we recall that $z$ is the component of $x$ in the space spanned by $\mu_1$ and $\mu_2$. We refer to this as the *clean* loss derivative, and our main tool will be analyzing the evolution of SGD under this clean surrogate for the loss derivative. Namely, we define:

$$\nabla_w^{\text{cl}} L_\rho := a_w \mathbb{E}_x \ell'_\rho(z)\sigma'(w^T x)x \quad \text{and} \quad \nabla_{a_w}^{\text{cl}} L_\rho := \mathbb{E}_x \ell'_\rho(z)\sigma(w^T x). \tag{4.7}$$

Before proceeding, we introduce the following definitions, which will be useful in Phase 2 (summarized in Table 1):

$$
\gamma_{\min} := \min_{\mu\in\{\pm\mu_1,\pm\mu_2\}} \gamma_\mu \qquad g_{\min} := \min_{\mu\in\{\pm\mu_1,\pm\mu_2\}} |\ell'_\rho(\mu)| = \frac{\exp(-\gamma_{\max})}{1+\exp(-\gamma_{\max})}
$$

$$
\gamma_{\max} := \max_{\mu\in\{\pm\mu_1,\pm\mu_2\}} \gamma_\mu \qquad g_{\max} := \max_{\mu\in\{\pm\mu_1,\pm\mu_2\}} |\ell'_\rho(\mu)| = \frac{\exp(-\gamma_{\min})}{1+\exp(-\gamma_{\min})}
$$

To ensure the approximation in Eq. 4.6 holds throughout the entire the second phase, we will maintain a certain inductive hypothesis, which ensures the the scale of the signal-direction components of the network continue to dominate the scale of the non-signal-direction components of the network. Formally, we consider the following condition.

**Definition 4.2** (Signal-Heavy Inductive Hypothesis). *For parameters $\zeta = o(1)$ and $H > 1$ with $\zeta \leq \exp(-10H)$, we say a network is $(\zeta, H)$-signal-heavy if there exists some set of heavy neurons $S$ on which $\exp(6H)\|w_\perp\| + \|w_{opp}\| \leq \zeta\|w_{sig}\|$, and*

$$\mathbb{E}_\rho \mathbf{1}(w \notin S)\|w\|^2 \leq \zeta\tilde{\gamma}_{min}.$$

*Here we have defined $\tilde{\gamma}_\mu := \mathbb{E}[\mathbf{1}(w \in S, w_{sig}^T \mu > 0) a_w \sigma(w^T \mu)]$ and $\tilde{\gamma}_{min} := \min_{\mu \in \{\pm\mu_1, \pm\mu_2\}} \tilde{\gamma}_\mu$. Further,*

$$\mathbb{E}_\rho[\|w\|^2] \leq \mathbb{E}_\rho[|a_w|^2] + \zeta H \leq 2H,$$

*and for all neurons, we have $|a_w| \leq \|w\|$.*

We show via a straightforward argument in Lemma E.4 that if the conclusion of Lemma 4.1 (from Phase 1) holds for some $\zeta$, then the network is $(\Theta(\zeta^{1/3}), H)$-signal-heavy, for $H = \Theta(\log\log(d))$.

Assuming that the network is $(\zeta, H)$-signal-heavy, using a similar approach to Eq. 4.4, we can show (see Lemma E.5 for the precise statement) that for any neuron $(w, a_w)$,

$$\frac{1}{|a_w|}\|\nabla_w L_\rho - \nabla_w^{cl} L_\rho\|_2 \lesssim \sqrt{\mathbb{E}_x(f_\rho(x) - f_\rho(z))^2} \lesssim \mathbb{E}_\rho[\|a_w w_\perp\|] \leq \zeta\gamma_{max},$$

and similarly $\|\nabla_{a_w} L_\rho - \nabla_{a_w}^{cl} L_\rho\|_2 \lesssim \|w\|\zeta\gamma_{max}$.

By working with the clean gradients, it is possible to approximately track (or bound) the evolution of $w_{sig}$, $w_\perp$, and $w_{opp}$ on neurons in $S$, the set of neurons for which $\|w_{sig}\| \gg \|w_\perp + w_{opp}\|$. In Lemmas E.6, E.7, and E.8 we show the following for any $w \in S$ (let $\mu$ be the direction of $w_{sig}$):

1. **The signal component $w_{\mathbf{sig}}$ grows quickly.** We have $-w_{sig}^T \nabla_w^{cl} L_\rho \approx |a_w \ell'_\rho(\mu)|\tau$, where $\tau := \frac{\sqrt{2}}{4}$. Also $a_w$ grows at a similar rate. This growth is due to the fact that points with $z = -\mu$ will almost never activate the ReLU, while points with $z = \mu$ almost always will.

2. **A linear combination of $\|w_\perp\|^2$ and $\|w_{\mathbf{opp}}\|^2$ decreases.** The argument here is more subtle, but the key idea is to argue that if $|w_\perp^T \xi| \geq |w_{opp}^T z|$ frequently, then $\|w_\perp\|^2$ will decrease. Meanwhile, if $|w_\perp^T \xi| \leq |w_{opp}^T z|$ frequently, then $w_{opp}$ will decrease (and there is a sizeable event on which they both decrease).

Since most of the mass of the network is in $S$, this shows that the signal will grow at the exponential rate $\tau|\ell'_\rho(\mu)|$ — or for the "weakest" cluster, that is, in the direction $\mu$ that maximizes $\tilde{\gamma}_\mu$, we will have $\tilde{\gamma}_{min}^{(t+1)} \gtrsim (1 + 2\eta\tau g_{max}) \tilde{\gamma}_{min}^{(t)}$.

On neurons outside of $S$, we show in Lemma E.11 that they grow *at most* as fast as the rate of the weakest clusters, meaning we can essentially ignore these neurons.

**Remark 4.3.** *If we did not train the second layer weights (and for instance they all had norm 1), then our tools would not suffice to maintain the signal-heavy hypothesis in Definition 4.2. Indeed, the neurons in $S$ would grow at a* linear *rate of $\tau|\ell'_\rho(\mu)|$, and at (up to) an equal linear rate outside of $S$. Thus the neurons outside of $S$ might eventually attain a non-negligible mass. However, because the layers are trained simultaneously, this leads to positive feedback between the growth of $\|w_{sig}\|$ and $|a_w|$, leading to exponential growth, maintaining the mass ratios between the neurons in and out of $S$.*

Combining the ideas above, we prove the following lemma, which shows that after one SGD step, the network stays signal-heavy (with a slightly worse parameter), the behavior of the weakest margin improves, and the network (measured by the size of the largest margin $\gamma_{max}$) doesn't become too big.

**Lemma 4.4** (Phase 2 Inductive Step: Informal; See Lemma E.3 for formal version)**.** *If a network $\rho_t$ is $(\zeta, H)$-signal heavy with heavy set $S$, then after one minibatch gradient step, with probability $1 - d^{-\omega(1)}$,*

1. *$\rho_{t+1}$ is $(\zeta(1 + 10\eta\zeta H), H)$-signal heavy.*

2. *$\tilde{\gamma}_{min}^{(t+1)} \geq (1 + 2\eta\tau(1 - o(1))g_{max}) \tilde{\gamma}_{min}^{(t)}$*

3. *$\tilde{\gamma}_{max}^{(t+1)} \leq (1 + 2\eta\tau(1 + o(1))g_{min}) \tilde{\gamma}_{max}^{(t)}$, where $\tilde{\gamma}_{max}^{(t)} := \max_{\mu \in \{\pm\mu_1, \pm\mu_2\}} \tilde{\gamma}_\mu^{(t)}$.*

Theorem 3.1 is proved by iterating this lemma $\Theta(\log\log(d)/\eta)$ times, yielding $\gamma_{min} \approx \tilde{\gamma}_{min} = \omega(1)$.

## 5 CONCLUSION

In this work, we showed that in $\tilde{O}(d)$ samples, it is possible to learn the XOR function on Boolean data on a 2-layer neural network. Our results shows that by a careful analysis that compares that

dynamics to the dyamincs under the surrogate $L_0$ loss, we can show that SGD find the signal features, and escape the region of the saddle where it was initialized. Then, after learning the feature direction, we show that SGD will enlarge and balance the signal components to learn well-classify points from all 4 clusters. We discuss some of the limits and possible extensions of our techniques in Section A.

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
