1. **The signal component $w_{\text{sig}}$ grows quickly.** We have $-w_{\text{sig}}^T \nabla^{cl}_w L_\rho \approx |a_w \ell'_\rho(\mu)|\tau$, where $\tau := \frac{\sqrt{2}}{4}$. Also $a_w$ grows at a similar rate. This growth is due to the fact that points with $z = -\mu$ will almost never activate the ReLU, while points with $z = \mu$ almost always will.

2. **A linear combination of $\|w_\perp\|^2$ and $\|w_{\text{opp}}\|^2$ decreases.** The argument here is more subtle, but the key idea is to argue that if $|w_\perp^T \xi| \geq |w_{\text{opp}}^T z|$ frequently, then $\|w_\perp\|^2$ will decrease. Meanwhile, if $|w_\perp^T \xi| \leq |w_{\text{opp}}^T z|$ frequently, then $w_{\text{opp}}$ will decrease (and there is a sizeable event on which they both decrease).

Since most of the mass of the network is in $S$, this shows that the signal will grow at the exponential rate $\tau|\ell'_\rho(\mu)|$ — or for the "weakest" cluster, that is, in the direction $\mu$ that maximizes $\tilde{\gamma}_\mu$, we will have $\tilde{\gamma}^{(t+1)}_{\min} \gtrsim (1 + 2\eta\tau g_{\max}) \tilde{\gamma}^{(t)}_{\min}$.

On neurons outside of $S$, we show in Lemma E.11 that they grow *at most* as fast as the rate of the weakest clusters, meaning we can essentially ignore these neurons.

**Remark 4.3.** *If we did not train the second layer weights (and for instance they all had norm 1), then our tools would not suffice to maintain the signal-heavy hypothesis in Definition 4.2. Indeed, the neurons in $S$ would grow at a* linear *rate of $\tau|\ell'_\rho(\mu)|$, and at (up to) an equal linear rate outside of $S$. Thus the neurons outside of $S$ might eventually attain a non-negligible mass. However, because the layers are trained simultaneously, this leads to positive feedback between the growth of $\|w_{sig}\|$ and $|a_w|$, leading to exponential growth, maintaining the mass ratios between the neurons in and out of $S$.*

Combining the ideas above, we prove the following lemma, which shows that after one SGD step, the network stays signal-heavy (with a slightly worse parameter), the behavior of the weakest margin improves, and the network (measured by the size of the largest margin $\gamma_{\max}$) doesn't become too big.

**Lemma 4.4** (Phase 2 Inductive Step: Informal; See Lemma E.3 for formal version)**.** *If a network $\rho_t$ is $(\zeta, H)$-signal heavy with heavy set $S$, then after one minibatch gradient step, with probability $1 - d^{-\omega(1)}$,*

1. *$\rho_{t+1}$ is $(\zeta(1 + 10\eta\zeta H), H)$-signal heavy.*

2. *$\tilde{\gamma}^{(t+1)}_{min} \geq (1 + 2\eta\tau(1 - o(1))g_{max}) \tilde{\gamma}^{(t)}_{min}$*

3. *$\tilde{\gamma}^{(t+1)}_{max} \leq (1 + 2\eta\tau(1 + o(1))g_{min}) \tilde{\gamma}^{(t)}_{max}$, where $\tilde{\gamma}^{(t)}_{max} := \max_{\mu \in \{\pm\mu_1, \pm\mu_2\}} \tilde{\gamma}^{(t)}_\mu$.*

Theorem 3.1 is proved by iterating this lemma $\Theta(\log\log(d)/\eta)$ times, yielding $\gamma_{\min} \approx \tilde{\gamma}_{\min} = \omega(1)$.

## 5 CONCLUSION

In this work, we showed that in $\tilde{O}(d)$ samples, it is possible to learn the XOR function on Boolean data on a 2-layer neural network. Our results shows that by a careful analysis that compares that dynamics to the dyamincs under the surrogate $L_0$ loss, we can show that SGD find the signal features, and escape the region of the saddle where it was initialized. Then, after learning the feature direction, we show that SGD will enlarge and balance the signal components to learn well-classify points from all 4 clusters. We discuss some of the limits and possible extensions of our techniques in Section A.

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

## LIST OF APPENDICES

## A    DISCUSSION

We now discuss some of the limits and possible extensions of our techniques.

**Minibatch SGD vs SGD vs GD.**    In this work, we study minibatch SGD, with a batch size of $m \geq d\text{polylog}(d)$. This affords us enough samples at each iteration to have strong enough convergence to the population loss. Extending our results to SGD with a batch size of $1$ is an interesting open question, and it is possible that this could be achieved using the drift-martingale techniques in Tan & Vershynin (2019); Arous et al. (2021); Abbe et al. (2023). Such methods allow larger fluctuations from the population loss at each step, but show that the fluctuations concentrate over time, even when SGD is run for $T = \omega(1/\eta)$ steps, enough time to escape a saddle.

We remark that in this problem, using minibatch SGD with fresh samples can achieve stronger sample complexities than that required to show uniform convergence of the empirical gradient to the population gradient (as in Ge et al. (2017); Mei et al. (2018a)), which in our setting, is $\Omega(d^2)$ samples. This means proving the convergence of GD on the empirical loss would require tools beyond uniform convergence.

**Boolean Data vs Gaussian Data.**    One limitation of this work is that our results only hold for boolean data, and not gaussian data $x \sim \mathcal{N}(0, I_d)$. As a matter of convenience, it is easier to compute the population gradients $\nabla_w L_0$ and $\nabla_w^{\text{cl}} L_\rho$ with Boolean data, and the gradient does not depend on interactions between $w_{\text{sig}}$ and $w_{\text{opp}}$. With some willingness to perform various Gaussian integrals, we believe the analysis in Phase 1 could be extended to the Gaussian setting. This would require changing Lemma D.17 to reflect the population gradients, and modifying the definition of "strong" neurons (Def. D.13) to be a more restrictive set that only includes neurons where $\|w_{\text{opp}}\| \ll \|w_{\text{sig}}\|$, such that $w_{\text{sig}}$ grows at the maximum possible rate. We do not know of any way to directly extend Phase 2 to the Gaussian case. This is because if the cluster margins $\gamma_u$ become very imbalanced, it is possible $w_{\text{sig}}$ could grow in the wrong direction.

**Classification vs Regression.**    In our classification setting, it suffices to show that the margin on each cluster grows large. We accomplish this in our Phase 2 analysis by showing that there is a large mass of neurons primarily in the $\mu$-direction for each $\mu \in \{\pm\mu_1, \pm\mu_2\}$. Adapting this strategy may be possible for XOR regression on Boolean data, but on Gaussian data, representing the ground truth function would require more specialization among the neurons. To see this, consider the following simpler example: to represent the single-index function $f^*(x) = (e_1^T x)^2$ on Gaussian data on a ReLU network without biases, the neurons cannot all be oriented in the $\pm e_1$ direction, otherwise the output would be $a\sigma(x_1) + b\sigma(-x_1)$ for scalars $a, b$. Studying the power of SGD to perform this specialization is an exciting open direction. We believe that our Phase 1 analysis may be a useful first step in this regard to show that the network can become signal-heavy. More powerful techniques would need to be developed to show specialization once the network contains sufficient signal.

## B  NOTATION AND ORGANIZATION

We use $\mathrm{Rad}^\ell$ to denote the uniform distribution on $\{\pm 1\}^\ell$. Any expectations or probabilities over $x$ are over the distribution $x \sim \mathrm{Rad}^d$. Whenever the variable $x$ is used, we use $z = z(x)$ to denote the projection of $x$ onto its first two coordinates (the space spanned by $\mu_1$ and $\mu_2$), and we use $\xi = \xi(x)$ to denote the projection of $x$ orthogonal to $z$. We use expectations over $z$ and $\xi$ to mean expectation over $z(x)$ and $\xi(x)$, where $x \sim \mathrm{Rad}^\ell$. For a vector $v \in \mathbb{R}^d$, we use the notation $v_{\setminus i}$ to denote the vector $v - e_i v_i$, which sets the $i$th coordinate of $v$ to 0.

Throughout, we often remove the superscript $(t)$ and use $w$ to denote $w^{(t)}$. Sometimes for emphasis, or when we are comparing $w^{(t+1)}$ and $w^{(t)}$, we will include the superscripts. However, the reader should not be alarmed if we use $w^{(t)}$ and $w$ interchangeably, even in such calculations. The same holds for all other neural network parameters.

Throughout the appendix, will use the notation $x \in a \pm b$ to mean that $|x - a| \le b$.

We recall that much helpful notation used throughout the appendix is summarized in Table 1 in the main body.

In Section C, we derive several auxiliary lemmas which will be used throughout the appendix. In Section D, we prove our results for Phase 1. In Section E, we prove our results for Phase 2. In Section F, we prove the main theorem. In Section G, we sketch why $\tilde{\Theta}(d)$ samples are needed for learning the XOR function with a rotationally invariant algorithm.

## C  AUXILIARY LEMMAS

### C.1  FROM BOOLEANS TO GAUSSIANS

The following section provides some lemmas to handle the fact that our data is drawn from the Boolean hypercube, and not Gaussian. At a high level, our goal is to show that for all neurons $w$, for any $a > 0$ we have that

$$\mathbb{P}_{\xi \sim \mathrm{Rad}^d}[|\xi^T w| \le a] \approx \mathbb{P}_{X \sim \mathcal{N}(0, I_d)}[|X^T w| \le a].$$

Of course, this is not true for all vectors $w$ (eg. if $w$ is sparse, these two probabilities may differ significantly.) Fortunately, the neurons $w$ we care about will be equal to their initialization (which is a random vector on the sphere) plus a small arbitrary perturbation. Thus we will define a notion of a "well-spread vector", claim that at initialization with high probability all neurons are well-spread (Lemma C.2), and then use Lemmas C.3 and C.4 to relate the Boolean probabilities to Gaussian probabilities, for all neurons $w$ that are near well-spread vectors. Later on in training, the neurons may not be close to their initialization, but we will be able to bound their $\ell_\infty$ norm, and thus use the well known Berry-Esseen Lemma to relate the Boolean and Gaussian probabilities.

**Definition C.1** (Well-Spread Vector). *We say a vector $v \in \mathbb{R}^d$ is $c$-well-spread if:*

1.  *$\|v\|_3^3 \le 20 \|v\|_2^3 d^{-1/2}$ and $\|v\|_\infty \le \frac{\log(d)}{\sqrt{d}} \|v\|_2$.*

2.  *Let $S$ index the set of $d/c^2$ coordinates of $v$ with smallest absolute value (break ties arbitrarily). Then $\sum_{i \in S} |v_i| \ge \frac{\|v\| \sqrt{d}}{c^5}$, and $\max_{i \in S} |v_i| \le \frac{\|v\|}{c\sqrt{d}}$.*

**Lemma C.2.** *For any constant $c$ large enough, the following holds. For any $r > 0$, if $w \sim \mathcal{S}^{d-1}(r)$, then with probability $1 - d^{-\omega(1)}$, $w$ is $c$-well-spread.*

*Proof.* Without loss of generality we may assume $r = 1$. We can write $w = u/X$, where $u \sim \mathcal{N}(0, \frac{1}{d} I_d)$, and $X = \|u\|$. With probability $1 - d^{-\omega(1)}$, the following events hold:

1.  We have $X \in [0.9, 1.1]$. This holds by Bernstein's concentration inequality for sub-exponential random variables.

2.  $\|u\|_\infty \le \frac{\log(d)}{2\sqrt{d}}$. This holds by a union bound over all $d$ coordinates, and using the Gaussian CDF.

3.  $\|u\|_3^3 \le 10 d^{-1/2}$. This holds by applying Lipshitz concentration of Gaussians to the function $f(u) := \|u\|_3$. Indeed, this function is 1-Lipshitz since $\|u\|_3 \le \|u\|_2$. We have $\mathbb{E}[\|u\|_3] = \mathbb{E}[(\sum_i |u_i^3|)^{1/3}] \le \left(\sum_i \mathbb{E}[\sum |u_i^3|]\right)^{1/3} \le \left(\sum_i \sqrt{\mathbb{E}[\sum u_i^6]}\right)^{1/3} = \left(d\sqrt{d^{-3} 15}\right)^{1/3} \le 2 d^{-1/6}$, so Lipshitz concentration yields that $\|u\|_3 \le 2.1 d^{-1/6}$ with probability $1 - d^{-\omega(1)}$, and such, that $\|u\|_3^3 \le 10 d^{-1/2}$.

4. For $c$ large enough, there are at least $d/c^2$ indices $i$ for which $|u_i| \leq \frac{1}{2c\sqrt{d}}$. This holds by a Chernoff bound for Bernoulli random variables, since for any $i$, $\mathbb{P}\left[|u_i| \leq \frac{1}{2c\sqrt{d}}\right] \geq \frac{1}{4c}$.

5. There are no more than $d/(2c^2)$ coordinates $i$ for which $|u_i| \leq \frac{10}{c^3\sqrt{d}}$. This again holds by a Chernoff bound for Bernoulli random variables, since for any $i$, $\mathbb{P}\left[|u_i| \leq \frac{10}{c^3\sqrt{d}}\right] \leq \frac{20}{c^3}$.

Combining the first, second, and third items above yields the first property of a well-spread vector for $w$. Combining the first, fourth, and fifth items above yields the second property of a well-spread vector for $w$. □

**Lemma C.3.** *There exists a universal constant $C_0$ such that the following holds for any $C$-well-spread vector $v$ for $C \geq C_0$. For any $\Delta \in \mathbb{R}^d$ with $\|\Delta\|_2 \leq \zeta\|v\|_2$ if $\zeta \leq \frac{1}{C^{10000}}$, we have for $d$ large enough:*

$$\mathbb{P}_\xi\left[|\xi^T(v+\Delta)| \leq \frac{\|v\|}{\sqrt{d}}\right] \geq \frac{1}{2}\exp(-100C^8)\frac{1}{\sqrt{d}}.$$

**Lemma C.4.** *There exists a universal constant $C_0$ such that the following holds for any $C$-well-spread vector $v$ for $C \geq C_0$. For any $\Delta \in \mathbb{R}^d$ with $\|\Delta\|_2 \leq \zeta\|v\|_2$ if $\zeta \leq \frac{1}{C^{10000}}$, we have for $d$ large enough,*

$$\left|\mathbb{P}_{\xi\sim\mathrm{Rad}^d}[\xi^T(v+\Delta) \in [a\|v\|, b\|v\|]] - P_{\frac{\lfloor b-a\rfloor}{2}}\right| \leq 2P_{\frac{\lfloor b-a\rfloor}{2}}(\sqrt{\zeta} + max(|a|,|b|)^2) + 200C_{BE}d^{-1/2},$$

*where*

$$P_c := \mathbb{P}_{G\sim\mathcal{N}(0,1)}[|G| \leq c] = \mathbb{P}_{X\sim\mathcal{N}(0,I_d)}[|X^Tv| \leq c\|v\|],$$

*and $C_{BE}$ is the constant from Theorem C.5.*

Our main tool in proving these two lemmas is the Berry-Esseen Inequality.

**Theorem C.5** (Berry-Esseen Inequality)**.** *There exists a universal constant $C_{BE}$ such that for independent mean $0$ random variables $X_1, \cdots X_n$, with $\mathbb{E}[X^2] = \sigma_i^2$, and $\mathbb{E}[|X_i|^3] = \rho_i$, we have*

$$\sup_x |\Phi(x) - F_n(x)| \leq C_{BE}\frac{\sum \rho_i}{(\sum \sigma_i^2)^{3/2}},$$

*where $\Phi$ is the CDF of $\mathcal{N}(0,1)$, and $F_n$ is the CDF of $\frac{\sum X_i}{\sqrt{\sum \sigma_i^2}}$. Thus if $u \in \mathbb{R}^\ell$, then*

$$\sup_x \left|\mathbb{P}_{\xi\sim\mathrm{Rad}^\ell}[u^T\xi \geq x] - \mathbb{P}_{G\sim\mathcal{N}(0,1)}[G \geq x/\|u\|]\right| \leq C_{BE}\frac{\|u\|^3}{\|u\|_2^3}.$$

We will use the following lemma which follows from Chebychev's inequality and Berry-Esseen.

**Lemma C.6.** *The following holds for any constant $C$ large enough. Suppose $u \in \mathbb{R}^\ell$ satisfies $\|u\|_\infty \leq 1$. Then for $\xi \sim \mathrm{Rad}^\ell$, we have*

$$\mathbb{P}[|\xi^Tu| \leq C] \geq \frac{1}{C\sqrt{\ell}}.$$

*Proof.* We need to do casework on the size of $\|u\|$. Applying Berry-Esseen, if $C \geq 8\sqrt{\pi}C_{BE}$ and $\|u\|_2 \geq 8\sqrt{\pi}C_{BE}$, we have

$$\mathbb{P}_\xi[|\xi^Tu| \leq C] \geq \mathbb{P}_{G\sim\mathcal{N}(0,1)}[|G| \leq C/\|u\|_2] - 2C_{BE}\frac{\|u\|_3^3}{\|u\|_2^3}$$

$$\geq \mathbb{P}_{G\sim\mathcal{N}(0,1)}[|G| \leq C/\|u\|_2] - 2C_{BE}\frac{\|u\|_2^2\|u\|_\infty}{\|u\|_2^3}$$

$$\geq \mathbb{P}_{G\sim\mathcal{N}(0,1)}[|G| \leq C/\|u\|_2] - \frac{2C_{BE}}{\|u\|_2}$$

$$\geq \sqrt{\frac{1}{\pi}}\frac{C}{C+\|u\|_2} - \frac{2C_{BE}}{\|u\|_2}$$

$$\geq \frac{1}{C\sqrt{\ell}}$$

Here in the second inequality we used Holder's inequality (with $p = 1$, $q = \infty$), and in the final inequality we used the bounds $8\sqrt{\pi}C_{\text{BE}} \leq \|u\|_2 \leq \sqrt{\ell}\|u\|_\infty \leq \sqrt{\ell}$ and $C \geq 8\sqrt{\pi}C_{\text{BE}}$.

If $\|u\| \leq 4\sqrt{\pi}C_{\text{BE}}$, then by Chebychev's inequality, we have for $C$ large enough:

$$\mathbb{P}_\xi[|\xi^T u| \geq C] \leq \frac{\mathbb{E}(\xi^T u)^2}{C^2} = \frac{\|u\|_2^2}{C^2} \leq 1 - \frac{1}{C\sqrt{\ell}}.$$

$\square$

*Proof of Lemma C.3.* Let $w := v + \Delta$. Let $B$ be the set of "bad" coordinates on which $|\Delta_i| \geq \frac{\|v\|}{3C^3\sqrt{d}}$. Thus since $\|\Delta\| \leq \zeta\|v\|$, we have $|B| \leq (3C^3)^2 d\zeta^2 \leq \frac{d}{3C^4}$ for $C$ large enough. Let $S$ be the set of $d/C^2$ coordinates of $v$ with smallest absolute value (as in the definition of well-spread). Thus letting $S' := S \setminus B$, we have:

**D1**

$$\sum_{i \in S'} |w_i| \geq \sum_{i \in S} |v_i| - |S'| \max_{i \notin B} |\Delta_i| - |B| \max_{i \in S} |v_i|$$

$$\geq \frac{\|v\|\sqrt{d}}{C^5} - \frac{d}{C^2}\left(\frac{\|v\|}{3C^3\sqrt{d}}\right) - \frac{d}{3C^4}\left(\frac{\|v\|}{C\sqrt{d}}\right)$$

$$\geq \frac{\|v\|\sqrt{d}}{3C^5}.$$

**D2** $\max_{i \in S'} |w_i| \leq \frac{\|v\|_2}{C\sqrt{d}} + \frac{\|v\|}{3C^3\sqrt{d}} \leq \frac{2\|v\|_2}{C\sqrt{d}}$.

Without loss of generality, assume the coordinates of $w$ are ordered with the with the indices not in $S'$ first, followed by the indices in $S'$.

Let $X_t$ for $t = 1 \ldots d$ be the random walk $X_t = \sum_{i=1}^t \xi_i w_i$. Let $\tau$ be the first time at which the random walk crosses zero after step $d - |S'|$, such that $|X_\tau| \leq |w_\tau|$. If it never crosses zero after this time, let $\tau := d + 1$.

Let $A$ be the event that $\tau \neq d + 1$, and let $B_t$ be the event that $\left|\sum_{i=t+1}^d \xi_i w_i\right| \leq \frac{\|v\|}{2\sqrt{d}}$. We proceed with a sequence of claims.

**Claim C.7.** *If $A$ and $B_\tau$ occur, then $\left|\sum_{i=1}^d \xi_i w_i\right| \leq \frac{\|v\|}{\sqrt{d}}$.*

*Proof.* If $A$ and $B_\tau$ occur, then

$$\left|\sum_{i=1}^d \xi_i w_i\right| \leq |X_\tau| + \left|\sum_{i=\tau+1}^d \xi_i w_i\right|$$

$$\leq |w_\tau| + \frac{\|v\|}{2\sqrt{d}}$$

$$\leq \max_{i \in S'} |w_i| + \frac{\|v\|}{2\sqrt{d}} \leq \frac{\|v\|}{\sqrt{d}},$$

for $C$ large enough. $\square$

**Claim C.8.**

$$\mathbb{P}_\xi\left[A \text{ and } B_\tau\right] \geq \mathbb{P}_\xi\left[A\right] \min_{t \in [d-|S'|, d]} \mathbb{P}_\xi\left[B_t\right]$$

*Proof.*

$$\mathbb{P}_\xi \left[ A \text{ and } B_\tau \right] = \sum_{t=d-|S'|}^{d} \mathbb{P}_\xi[\tau = t]\mathbb{P}_\xi[B_t|\tau = t]$$

$$= \sum_{t=d-|S'|}^{d} \mathbb{P}_\xi[\tau = t]\mathbb{P}_\xi[B_t]$$

$$\geq \left( \sum_{t=d-|S'|}^{d} \mathbb{P}_\xi[\tau = t] \right) \min_{t\in[d-|S'|,d]} \mathbb{P}_\xi[B_t]$$

$$= \mathbb{P}_\xi[A] \min_{t\in[d-|S'|,d]} \mathbb{P}_\xi[B_t].$$

$\square$

**Claim C.9.** *For any $t \in [d-|S'|,d]$, $\mathbb{P}_\xi[B_t] \geq \frac{1}{\sqrt{d}}$.*

*Proof.* We will apply Lemma C.6. Fix $t$ and let $u' \in \mathbb{R}^{d-t}$ be the vector with $u_i' = w_{i+t}$. Then $\|u'\|_\infty \leq \frac{2\|v\|_2}{C\sqrt{d}}$ by **D2**. Let $u := \frac{u'}{\|u'\|_\infty}$. Applying Lemma C.6 to $u$ yields (so long as $C \geq 4C_{C.6}$, where $C_{C.6}$ is the constant in Lemma C.6):

$$\mathbb{P}\left[ |\xi^T u'| \leq \frac{\|v\|}{2\sqrt{d}} \right] = \mathbb{P}\left[ |\xi^T u| \leq \frac{\|v\|}{2\sqrt{d}\|u\|_\infty} \right]$$

$$\geq \mathbb{P}\left[ |\xi^T u| \leq \frac{C}{4} \right]$$

$$\geq \frac{1}{C_{C.6}\sqrt{d-t}}$$

$$\geq \frac{1}{C_{C.6}\sqrt{|S'|}}$$

$$\geq \frac{1}{C_{C.6}\sqrt{d/C^2}} \geq \frac{1}{\sqrt{d}}.$$

Here the first inequality in the last line follows from the fact that $|S'| \leq |S| = \frac{d}{C^2}$. $\square$

**Claim C.10.**

$$\mathbb{P}_\xi[\tau \geq d - |S'|] \geq \frac{1}{2}\exp(-100C^8).$$

*Proof.* Because for any scalar $b$ and any set of coordinates $P$ we have $\mathbb{P}_\xi\left[ \sum_{i\in P} \xi_i w_i = b \right] = \mathbb{P}_\xi\left[ \sum_{i\in P} \xi_i w_i = -b \right]$, we have

$$\mathbb{P}_\xi[\tau \geq d - |S'|] \geq \mathbb{P}_\xi\left[ |X_{d-|S'|}| \leq 2\|v\| \right] \mathbb{P}\left[ \sum_{i=d-|S'|+1}^{d} \xi_i^T w_i \geq 2\|v\| \right]. \tag{C.1}$$

We will show that these two probabilities are sufficiently large via Berry-Esseen. For the second probability, let $u \in \mathbb{R}^{d-|S'|}$ be the vector with $u_i = w_{i+d-|S'|}$. Then we have by **D1**,

$$\|u\|_2 \geq \frac{1}{\sqrt{|S'|}}\|u\|_1 \geq \frac{\|v\|\sqrt{d}}{3C^5\sqrt{|S'|}} \geq \frac{\|v\|\sqrt{d}}{3C^5\sqrt{|S|}} = \frac{\|v\|}{3C^4},$$

while by Holder's inequality and **D2**, we have

$$\|u\|_3^3 \leq \|u\|_2^2\|u\|_\infty \leq \|u\|_2^2\frac{2\|v\|}{C\sqrt{d}}.$$

Thus by Berry-Esseen, we have

$$\mathbb{P}\left[\sum_{i=d-|S'|+1}^{d} \xi_i^T w_i \geq 2\|v\|\right] \geq \mathbb{P}_{G\sim\mathcal{N}(0,1)}[G \geq 2\|v\|/\|u\|] - C_{\mathrm{BE}}\frac{2\|v\|}{C\sqrt{d}\|u\|} \quad (\text{C.2})$$

$$\geq \mathbb{P}_{G\sim\mathcal{N}(0,1)}[G \geq 6C^4] - C_{\mathrm{BE}}\frac{6C^3}{\sqrt{d}}$$

$$\geq \exp(-100C^8),$$

for a constant $C$ large enough (and $d$ sufficiently large).

Now consider the probability $\mathbb{P}_\xi\left[|X_{d-|S'|}| \leq 2\|v\|\right]$. By Chebychev's inequality, we have

$$\mathbb{P}_\xi\left[|X_{d-|S'|}| \geq 2\|v\|\right] \leq \frac{\mathbb{E}[X_{d-|S'|}^2]}{4\|v\|^2}$$

$$= \frac{\mathbb{E}[\sum_{i=1}^{d-|S'|} w_i^2]}{4\|v\|^2}$$

$$\leq \frac{\|w\|^2}{4\|v\|^2} \leq \frac{1}{2}.$$

Combining this with Eq C.2 and Eq C.1 yields the claim. $\qquad\square$

Combining Claims C.7-C.10 yields

$$\mathbb{P}_\xi\left[|\xi^T w| \leq \frac{\|v\|}{\sqrt{d}}\right] \geq \frac{1}{2}\exp(-100C^8)\frac{1}{\sqrt{d}},$$

which proves the lemma. $\qquad\square$

*Proof of Lemma C.4.* We can without loss of generality assume $b \geq |a|$, since the variable $\xi$ is symmetric. Define $B := \{i : |\Delta_i| \geq \sqrt{\zeta}\frac{\|v\|}{\sqrt{d}}\}$. Observe that since $\|\Delta\|_2 \leq \zeta\|v\|$, we have $|B| \leq \zeta d$. Let $w = v + \Delta$. We can write

$$\mathbb{P}_\xi\left[\xi^T w \in [a,b]\|v\|\right] = \int_{x=-\infty}^{\infty} \mathbb{P}_\xi\left[\sum_{i\in B}\xi_i w_i = x\|v\|\right]\mathbb{P}_\xi\left[\sum_{i\in[d]\setminus B}\xi_i w_i \in [a-x, b-x]\|v\|\right] \tag{C.3}$$

We use the following claim to bound this integral.

**Claim C.11.**

$$\left|\mathbb{P}_\xi\left[\sum_{i\in[d]\setminus B}\xi_i w_i \in [a-x, b-x]\|v\|\right] - P_{\frac{b-a}{2}}\right| \leq P_{\frac{b-a}{2}}\left(\sqrt{\zeta} + (|x|+b)^2/2\right) + 200C_{BE}d^{-1/2}.$$

*Proof.* We use Berry-Esseen. Let $u \in \mathbb{R}^{d-|B|}$ with coordinates equal to the set $\{w_i\}_{i\in[d]\setminus S}$. Then

$$\left|\|u\|^2 - \|v\|^2\right| \leq |B|\max_{i\in B}|v_i|^2 + \|\Delta\|^2 + 2\|v\|\|\Delta\|$$

$$\leq \zeta\log(d)^2\|v\|^2 + 3\zeta\|v\|^2$$

$$\leq \sqrt{\zeta}\|v\|^2,$$

where the last line follows because $\zeta \leq 1/\log^5(d)$. Further

$$\|u\|_3^3 \leq \sum_{i\notin B} w_i^3 \leq \sum_{i\notin B} 4\left(v_i^3 + \Delta_i^3\right) \leq\leq 4\|v\|_3^3 + 4\|\Delta\|^2\sup_{i\notin B}|\Delta_i| \leq 4\|v\|_3^3 + 4\zeta^{3/2}\|v\|_2^3 d^{-1/2} \leq 100\|v\|_2^3 d^{-1/2}.$$

In both these computations we have used the fact that $v$ is well-spread to bound $\|v\|_\infty$ and $\|v\|_3^3$.

By Berry-Esseen (Theorem C.5), we have

$$\mathbb{P}_\xi \left[ \sum_{i \in [d] \setminus B} \xi_i w_i \in [a - x, b - x] \|v\| \right] \in \mathbb{P}_{G \sim \mathcal{N}(0,1)} \left[ G \in [a - x, b - x] \frac{\|v\|}{\|u\|} \right] \pm 2C_{\mathrm{BE}} \frac{\|u\|_3^3}{\|u\|_2^3}$$

(C.4)

Now we have

$$\mathbb{P}_{G \sim \mathcal{N}(0,1)} \left[ G \in [a - x, b - x] \frac{\|v\|}{\|u\|} \right]$$

$$\in \mathbb{P}_{G \sim \mathcal{N}(0,1)} \left[ G \in [a - x, b - x] \right] \left[ 1 - \sqrt{\zeta}, 1 + \sqrt{\zeta} \right]$$

$$\in \mathbb{P}_{G \sim \mathcal{N}(0,1)} \left[ G \in \left[ -\frac{b-a}{2}, \frac{b-a}{2} \right] \right] \left[ \frac{\phi(b + |x|)}{\phi(0)} - \sqrt{\zeta}, 1 + \sqrt{\zeta} \right].$$

where we have used the fact that $\frac{\|v\|}{\|u\|} \in \left[ \frac{1}{\sqrt{1 - \sqrt{\zeta}}}, \frac{1}{\sqrt{1 + \sqrt{\zeta}}} \right] \in [1 - \sqrt{\zeta}, 1 + \sqrt{\zeta}]$, and that $a \le |b|$.
Now

$$\frac{\phi(b + |x|)}{\phi(0)} = \exp(-(b + |x|)^2/2) \ge 1 - (b + |x|)^2/2,$$

so

$$\mathbb{P}_{G \sim \mathcal{N}(0,1)} \left[ G \in [a - x, b - x] \frac{\|v\|}{\|u\|} \right] \in P_{\frac{b-a}{2}} \left[ 1 - \sqrt{\zeta} - (b + |x|)^2/2, 1 + \sqrt{\zeta} \right]$$

Thus returning to Eq C.4, we have

$$\mathbb{P}_\xi \left[ \sum_{i \in [d] \setminus B} \xi_i w_i \in [a - x, b - x] \|v\| \right] \in P_{\frac{b-a}{2}} \left[ \left( 1 - (|x| + b)^2/2 - \sqrt{\zeta} \right), 1 + \sqrt{\zeta} \right] \pm 200 C_{\mathrm{BE}} d^{-1/2},$$

which yields the desired result. $\square$

Returning to Eq C.3, we have

$$\left| \mathbb{P}_\xi \left[ \xi^T w \in [a\|v\|, b\|v\|] \right] - P_{\frac{b-a}{2}} \right| \le P_{\frac{b-a}{2}} \sqrt{\zeta} + 200 C_{\mathrm{BE}} d^{-1/2} + \frac{1}{2} P_{\frac{b-a}{2}} \int_{x=-\infty}^{\infty} \mathbb{P}_\xi \left[ \sum_{i \in B} \xi_i w_i = x\|v\| \right] (|x| + b)^2$$

$$\le P_a(\sqrt{\zeta} + b^2) + 200 C_{\mathrm{BE}} d^{-1/2} + P_{\frac{b-a}{2}} \int_{x=-\infty}^{\infty} \mathbb{P}_\xi \left[ \sum_{i \in B} \xi_i w_i = x\|v\| \right] x^2$$

$$= P_a(\sqrt{\zeta} + b^2) + 200 C_{\mathrm{BE}} d^{-1/2} + P_{\frac{b-a}{2}} \frac{1}{\|v\|^2} \mathbb{E}_\xi \left[ \left( \sum_{i \in B} \xi_i w_i \right)^2 \right]$$

$$= P_a(\sqrt{\zeta} + b^2) + 200 C_{\mathrm{BE}} d^{-1/2} + P_{\frac{b-a}{2}} \frac{1}{\|v\|^2} \left( \sum_{i \in B} w_i^2 \right).$$

Now

$$\sum_{i \in B} w_i^2 \le \sum_{i \in B} 2(\Delta_i^2 + v_i^2) \le 2\|\Delta\|^2 + 2|B| \max_i v_i^2 \le \sqrt{\zeta} \|v\|^2.$$

Thus we have

$$\left| \mathbb{P}_\xi \left[ |\xi^T w| \in [a\|v\|, b\|v\|] \right] - P_{\frac{b-a}{2}} \right| \le 2 P_{\frac{b-a}{2}} (\sqrt{\zeta} + b^2) + 200 C_{\mathrm{BE}} d^{-1/2},$$

which proves the lemma. $\square$

## C.2 Simultaneous Training of Two-Layer ReLU Networks

We will make use of the following general-purpose lemma for training neural networks with ReLU activations and Lipshitz loss functions.

**Lemma C.12** (Empirical Concentration). *If we train with data from $P_d$ on any loss $L$ that is 2-Lipshitz in the output of the network, with a minibatch size of $m \geq d \log(d)^2$, we have for any neuron $(w, a_w)$ with probability $1 - d^{-\omega(1)}$,*

1. *$\|\nabla_w L - \nabla_w \hat{L}\|^2 \leq \frac{d \log^2(d)}{m} a_w^2$; and for any $i \in [d]$, $\|\nabla_{w_i} L - \nabla_{w_i} \hat{L}\|^2 \leq \frac{\log^2(d)}{m} a_w^2$;*
2. *$\|\nabla_{a_w} L - \nabla_{a_w} \hat{L}\|^2 \leq \frac{d \log^2(d)}{m} \|w\|^2$;*

*Proof.* We can use the Generalized Hoeffding's inequality to prove this.

For the first statement, each coordinate of $\ell'_\rho(x)\sigma'(w^T x)x$ minus its expectation a random variable bounded by a constant, and is thus subgaussian.

For the second statements, $\ell'_\rho(x)\sigma(w^T x)$ minus its expectation is subgaussian with parameter $\|w\|$. □

**Lemma C.13.** *If we train with data from $P_d$ on any loss $\ell$ that is 2-Lipshitz with respect the the output of the network, with a minibatch size of $m \geq d \log(d)^2$ and $\eta \leq 1/4$, we have for any neuron $(w, a_w)$ with probability $1 - d^{-\omega(1)}$,*

**S1** $\|\nabla_w L\| \leq 2|a_w|$;

**S2** $\|\nabla_a L\| \leq 2\|w\|$;

**S3** *If $|a_w^{(t)}| \leq \|w^{(t)}\|$, then $|a_w^{(t+1)}| \leq \|w^{(t+1)}\|$.*

**S4** $\|w^{(t+1)}\|^2 - |(a_w^{(t+1)})|^2 \leq 4\eta^2 |a_w^{(t)}|^2 + \|w^{(t)}\|^2 - |(a_w^{(t)})|^2$.

*Proof.* Consider the first statement first. We have

$$\frac{1}{|a_w|} \|\nabla_w L\| = \sup_{v:\|v\|=1} \mathbb{E}_x \ell'_\rho(x)\sigma'(w^T x)x^T v$$
$$\leq \mathbb{E}_x |\ell'_\rho(x)||x^T v|$$
$$\leq 2\mathbb{E}_x |x^T v|$$
$$\leq 2.$$

For the second statement, we have

$$\|\nabla_a L\| = \mathbb{E}_x \ell'_\rho(x)\sigma(w^T x)$$
$$\leq 2\|w\|.$$

For the third and fouth statements, we use the shorthand $a = a_w^{(t)}$, $w = w^{(t)}$. To prove the third statement, we can write

$$(a^{(t+1)})^2 = \left(a - \eta \nabla_a \hat{L}\right)^2$$
$$= (a)^2 - 2\eta a \nabla_a \hat{L} + \eta^2 (\nabla_a \hat{L})^2$$

and

$$\|w^{(t+1)}\|^2 = \|w - \eta \nabla_w \hat{L}\|^2$$
$$= \|w\|^2 - 2\eta w^T \nabla_w \hat{L} + \eta^2 \|\nabla_w \hat{L}\|^2.$$

Because we use the ReLu activation, we have

$$(w^{(t)})^T \nabla_w \hat{L} = a^{(t)} \nabla_a \hat{L}.$$

Thus

$$(a^{(t+1)})^2 - \|w^{(t+1)}\|^2 = (a^{(t)})^2 - \|w^{(t)}\|^2 + \eta^2 \left( (\nabla_a \hat{L})^2 - \|\nabla_w \hat{L}\|^2 \right)$$

$$\leq (a^{(t)})^2 - \|w^{(t)}\|^2 + \eta^2 \left( (\nabla_a \hat{L})^2 - \frac{1}{\|w\|^2} (w^T \nabla_w \hat{L})^2 \right)$$

$$= (a^{(t)})^2 - \|w^{(t)}\|^2 + \eta^2 \left( (\nabla_a \hat{L})^2 - \frac{a^2}{\|w\|^2} (\nabla_a \hat{L})^2 \right)$$

$$= (a^{(t)})^2 - \|w^{(t)}\|^2 + \frac{\eta^2 (\nabla_a \hat{L})^2}{\|w\|^2} \left( \|w\|^2 - a^2 \right)$$

$$= \left( (a^{(t)})^2 - \|w^{(t)}\|^2 \right) \left( 1 - \frac{\eta^2 (\nabla_a \hat{L})^2}{\|w\|^2} \right)$$

By the previous conclusions of the lemma and Lemma C.12, we have with probability $1 - d^{-\omega(1)}$,

$$(\nabla_a \hat{L})^2 \leq 2(\nabla_a L)^2 + 2\zeta \|w\|^2 \leq 2\|w\|^2 + 2\zeta \|w\|^2,$$

where $\zeta = \log(d)^2 \frac{d}{m}$. Assuming $\eta \leq 1/4$, this yields the desired statement.

For the fourth result, we have with probability $1 - d^{-\omega(1)}$,

$$\|w^{(t+1)}\|^2 - (a^{(t+1)})^2 - \|w^{(t)}\|^2 + (a^{(t)})^2 = \eta^2 \left( \|\nabla_w \hat{L}\|^2 - (\nabla_a \hat{L})^2 \right)$$

$$\leq 2\eta^2 \left( \|\nabla_w L\|^2 + \|\nabla_w \hat{L} - \nabla_w L\|^2 \right)$$

$$\leq 4\eta^2 a_w^2.$$

$\square$

## D  PHASE 1

In this section, we prove the following lemma.

**Lemma D.1** (Output of Phase 1; Formal). *For any constants $c$ sufficiently large, and $C$ sufficiently large in terms of $c$, for any $d$ large enough, the following holds. Let $\theta := 1/\log(d)^C$.*

*Suppose we train a 2-layer neural network with minibatch SGD as in Section 2.2 with a minibatch size of $m \geq d/\theta^2$, width $1/\theta \leq p \leq d^C$, and step size $\eta \leq \theta$, and initialization scale $\theta$. Then with probability at least $1 - \theta$, after some $T_1 = \Theta(\log(d)/\eta)$ steps of minibatch SGD, the network $\rho_{T_1}$ satisfies:*

1. *$\mathbb{E}_{\rho_{T_1}}[\|a_w w\|] \leq 1$;*
2. *$\mathbb{E}_{\rho_{T_1}}[\|w_\perp + w_{opp}\|^2] \leq 4\theta^2$;*
3. *For all $\mu \in \{\pm\mu_1, \pm\mu_2\}$, on at least a $0.1$ fraction of the neurons, we have $\|w_{sig}\| \geq \log(d)^c \theta$ and $w_{sig}^T \mu > 0$.*

*Additionally,*

$$\mathbb{E}_{\rho_{T_1}}[\|w\|^2] \leq \mathbb{E}_{\rho_{T_1}}[|a_w|^2] + \sqrt{\eta},$$

*and for all neurons, we have $|a_w| \leq \|w\|$.*

### D.1  PHASE 1 GRADIENT BOUNDS

The core ingredients of Phase 1 are the following three lemmas, which relate the gradient $\nabla L_\rho$ to $\nabla L_0$ and compute several properties of the $L_0$ population gradient. Recall that we have

$$\nabla_w L_0 = \mathbb{E}_x \frac{\partial}{\partial w} \ell_0(x; \rho) = -\mathbb{E}_x a_w y(x) \sigma'(w^T x) x;$$

$$\nabla_{a_w} L_0 = \mathbb{E}_x \frac{\partial}{\partial a_w} \ell_0(x; \rho) = -\mathbb{E}_x y(x) \sigma(w^T x),$$

which is independent of the full distribution $\rho$.

**Lemma D.2.** *For any neuron* $(w, a_w)$,

**G1** $|\nabla_{a_w} L_0 - \nabla_{a_w} L_\rho| \leq 2\|w\|\mathbb{E}_\rho[\|a_w w\|]$.

**G2** $\|\nabla_w L_0 - \nabla_w L_\rho\| \leq 2|a_w|\mathbb{E}_\rho[\|a_w w\|]$.

**Lemma D.3.** *Suppose* $\mathbb{E}_\rho[\|a_w w\|] \leq d^{O(1)}$. *For any neuron* $(w, a_w)$, *for any* $i \in [d]$,

$$\|\nabla_{w_i} L_\rho - \nabla_{w_i} L_0\|$$
$$\leq |a_w| \left( 4(\mathbb{E}_\rho[\|a_w w_i\|]) + 2\log(d)\mathbb{E}_\rho[\|a_w w\|]\mathbb{E}_x \mathbf{1}(|x_{\backslash i}^T w| \leq |w_i|) + d^{-\omega(1)} \right).$$

**Lemma D.4** (Signal Subspace $L_0$ Population Gradients). *For any neuron* $(w, a_w)$, *we have*

**B1** $-w_{sig}^T \nabla_w L_0 = \frac{\sqrt{2}}{4}|a_w|\mathbb{E}_\xi \mathbf{1}(|w^T\xi| \leq \sqrt{2}\|w_{sig}\|)\|w_{sig}\|$

**B2** $-w_{opp}^T \nabla_w L_0 = -\frac{\sqrt{2}}{4}|a_w|\mathbb{E}_\xi \mathbf{1}(|w^T\xi| \leq \sqrt{2}\|w_{opp}\|)\|w_{opp}\|$

**B3** $-w_\perp^T \nabla_w L_0 \leq \begin{cases} \frac{|a_w|}{4}\mathbb{E}_\xi[\mathbf{1}(|w^T\xi| \in [\sqrt{2}\|w_{sig}\|, \sqrt{2}\|w_{opp}\|])|w^T\xi|] & \|w_{opp}\| > \|w_{sig}\|; \\ -\frac{|a_w|}{4}\mathbb{E}_\xi[\mathbf{1}(|w^T\xi| \in [\sqrt{2}\|w_{opp}\|, \sqrt{2}\|w_{sig}\|])|w^T\xi|] & \|w_{opp}\| \leq \|w_{sig}\|. \end{cases}$

**B4** *For* $i \in [3, d]$, *with* $X = (\xi - e_i \xi_i)^T w$, *we have*

$$-w_i^T \nabla_{w_i} L_0 =$$
$$\frac{|a_w||w_i|}{4} \left( \mathbb{P}\left[ X \in [\sqrt{2}\|w_{opp}\| - |w_i|, \sqrt{2}\|w_{opp}\| + |w_i|] \right] - \mathbb{P}\left[ X \in [\sqrt{2}\|w_{sig}\| - |w_i|, \sqrt{2}\|w_{sig}\| + |w_i|] \right] \right).$$

*Proof of Lemma D.4.* First consider **B1**. By symmetrizing over the pair $(z + \xi, -z + \xi)$, we have

$$-w_{sig}^T \nabla_w L_0 = a_w \mathbb{E}_x y(x)\sigma'(w^T x)z^T w_{sig}$$
$$= \frac{1}{2} a_w \mathbb{E}_\xi y(z)(\sigma'(w^T\xi + w_{sig}^T z) - \sigma'(w^T\xi - w_{sig}^T z))z^T w_{sig}$$
$$= \frac{1}{2} a_w \mathbb{E}_\xi y(z)\mathbf{1}(|w^T\xi| \leq |w_{sig}^T z|)\,\text{sign}(w_{sig}^T z)z^T w_{sig}$$
$$= \frac{\sqrt{2}}{4}|a_w|\mathbb{E}_\xi \mathbf{1}(|w^T\xi| \leq \sqrt{2}\|w_{sig}\|)\|w_{sig}\|$$

since $y(z)a_w > 0$ if $z \in \text{span}(w_{sig})$.

Next consider **B2**. By a similar calculation via symmetrization, but using the fact that $y(z)a_w < 0$ if $z \in \text{span}(w_{opp})$, we have

$$-w_{opp}^T \nabla_w L_0 = \frac{1}{2} a_w \mathbb{E}_\xi y(z)\mathbf{1}(|w^T\xi| \leq |w_{opp}^T z|)\,\text{sign}(w_{opp}^T z)z^T w_{opp}$$
$$= \frac{\sqrt{2}}{4}|a_w|\mathbb{E}_\xi \mathbf{1}(|w^T\xi| \leq \sqrt{2}\|w_{opp}\|)\|w_{opp}\|.$$

Next consider **B3**. Symmetrizing over the pair $(z + \xi, z - \xi)$, we have

$$-w_\perp^T \nabla_{w_\perp} L_0 = a_w \mathbb{E}_x y(x)\sigma'(w^T x)w_\perp^T \xi$$
$$= a_w \frac{1}{2}\mathbb{E}_x y(z)\left(\sigma'(w^T z + w^T \xi) - \sigma'(w^T z - w^T \xi)\right)w_\perp^T \xi$$
$$= a_w \frac{1}{2}\mathbb{E}_x y(z)\mathbf{1}(|w^T\xi| \geq |w^T z|)|w_\perp^T \xi|$$
$$= \begin{cases} |a_w|\frac{1}{4}\mathbb{E}_\xi \mathbf{1}(|w^T\xi| \in [\sqrt{2}\|w_{opp}\|, \sqrt{2}\|w_{sig}\|])|w^T\xi| & \|w_{opp}\| \leq \|w_{sig}\| \\ -|a_w|\frac{1}{4}\mathbb{E}_\xi \mathbf{1}(|w^T\xi| \in [\sqrt{2}\|w_{sig}\|, \sqrt{2}\|w_{opp}\|])|w^T\xi| & \|w_{sig}\| \leq \|w_{opp}\| \end{cases}$$

Finally consider **B4**. Recall that $\xi_{\backslash i}$ denotes $\xi - e_i \xi_i$. Then symmetrizing over the pair $(z + \xi_{\backslash i} + e_i \xi_i, z + \xi_{\backslash i} - e_i \xi_i)$, we have

$$-w_i^T \nabla_{w_i} L_0 = -a_w \mathbb{E}_x y(x)\sigma'(w^T x)w_i \xi_i$$
$$= -a_w \frac{1}{2}\mathbb{E}_x y(z)\left(\sigma'(w^T z + w^T\xi_{\backslash i} + w_i \xi_i) - \sigma'(w^T z + w^T\xi_{\backslash i} - w_i\xi_i)\right)w_i\xi_i$$
$$= -a_w \frac{1}{2}\mathbb{E}_x y(z)\mathbf{1}(|w^T z + w^T\xi_{\backslash i}| \leq |w_i|)|w_i|$$

Now explicitly evaluating the expectation over $z$ and noting that the variable $\xi_{\backslash i}$ is symmetric, we have

$$
a_w \mathbb{E}_x y(z) \mathbf{1}(|w^T z + w^T \xi_{\backslash i}| \le |w_i|)
$$
$$
= |a_w| \frac{1}{2} \mathbb{P}_{\xi_{\backslash i}} \left[ w^T \xi_{\backslash i} \in \left[ \sqrt{2} \|w_{\text{sig}}\| - |w_i|, \sqrt{2} \|w_{\text{sig}}\| - |w_i| \right] \right]
$$
$$
- |a_w| \frac{1}{2} \mathbb{P}_{\xi_{\backslash i}} \left[ w^T \xi_{\backslash i} \in \left[ \sqrt{2} \|w_{\text{opp}}\| - |w_i|, \sqrt{2} \|w_{\text{opp}}\| - |w_i| \right] \right].
$$

Thus with $X = w^T \xi_{\backslash i}$, we have

$$
-w_i^T \nabla_{w_i} L_0 =
$$
$$
\frac{|a_w||w_i|}{4} \left( \mathbb{P}_X \left[ X \in [\sqrt{2}\|w_{\text{opp}}\| - |w_i|, \sqrt{2}\|w_{\text{opp}}\| + |w_i|] \right] - \mathbb{P}_X \left[ X \in [\sqrt{2}\|w_{\text{sig}}\| - |w_i|, \sqrt{2}\|w_{\text{sig}}\| + |w_i|] \right] \right).
$$

$\square$

To prove Lemma D.2 and D.3, we will need the following lemma.

**Lemma D.5.** *Suppose $x \sim \text{Rad}^d$. Then*

$$
\mathbb{E}_x(\ell'_\rho(x) - \ell'_0(x))^2 \le 4(\mathbb{E}_\rho[\|a_w w\|])^2.
$$

*Further for any $x$ on the boolean hypercube and $i \in [d]$,*

$$
(\ell'_\rho(x_{\backslash i} + e_i x_i) - \ell'_\rho(x_{\backslash i} - e_i x_i))^2 \le 16(\mathbb{E}_\rho[\|a_w w_i\|])^2.
$$

*Proof.* Recall that $\ell_\rho(x) = -2 \log \left( \frac{1}{1 + \exp(-y(x) f_\rho(x))} \right)$, and so $\ell'_\rho(x) = -\frac{2y(x) \exp(-y(x) f_\rho(x))}{1 + \exp(-y(x) f_\rho(x))}$. Observe that $\ell'_\rho(x)$ is 2-Lipshitz with respect to $f_\rho(x)$. Thus for the first statement, using Jensen's inequality, we have

$$
\mathbb{E}_x(\ell'_\rho(x) - \ell'_0(x))^2 \le 4\mathbb{E}_x f_\rho(x)^2
$$
$$
= 4\mathbb{E}_x \left( \mathbb{E}_\rho a_w \sigma(w^T x) \right)^2
$$
$$
= 4 \sup_{v: \|v\| = 1} \mathbb{E}_x \left( \mathbb{E}_\rho \|a_w w\| |v^T x| \right)^2
$$
$$
= 4(\mathbb{E}_\rho \|a_w w\|)^2
$$

For the second statement, by the 2-Lipshitzness of $\ell'$, we have

$$
(\ell'_\rho(x_{\backslash i} + e_i x_i) - \ell'_\rho(x_{\backslash i} - e_i x_i))^2 \le 4(f_\rho(x_{\backslash i} + e_i x_i) - f_\rho(x_{\backslash i} - e_i x_i))^2
$$
$$
= 4 \left( \mathbb{E}_\rho a_w (\sigma(w^T x_{\backslash i} + w_i x_i) - \sigma(w^T x_{\backslash i} - w_i x_i)) \right)^2
$$
$$
\le 4 \left( \mathbb{E}_\rho 2|a_w||w_i| \right)^2
$$
$$
= 16(\mathbb{E}_\rho[\|a_w w_i\|])^2.
$$

$\square$

*Proof of Lemma D.2.* For convenience, define $\Delta_x := (\ell'_\rho(x) - \ell'_0(x))\sigma'(w^T x)$. We consider item **G1** first. By Cauchy Schwartz, we have

$$
|\nabla L_0(a_w) - \nabla L_\rho(a_w)| = \left| \mathbb{E}_x(\ell'_\rho(x) - \ell'_0(x))\sigma(w^T x) \right|
$$
$$
= \left| \mathbb{E}_x \Delta_x w^T x \right|
$$
$$
\le \sqrt{\mathbb{E}_x[\Delta_x^2]} \sqrt{\mathbb{E}_x(w^T x)^2}
$$
$$
\le \sqrt{\mathbb{E}_x[(\ell'_\rho(x) - \ell'_0(x))^2]} \|w\|_2.
$$

Now by Lemma D.5, we have $\mathbb{E}_x[(\ell'_\rho(x) - \ell'_0(x))^2] \le 4p^2 (\mathbb{E}_\rho[\|a_w w\|])^2$. Plugging this yields **G1** and **G2**.

For item **G2**, we similarly have

$$
\begin{aligned}
\|\nabla_w L_\rho - \nabla_w L_0\|_2 &= |a_w| \|\mathbb{E}_x \Delta_x x\| \\
&= |a_w| \sup_{v:\|v\|=1} \mathbb{E}_x \Delta_x \langle v, x \rangle \\
&\leq |a_w| \sup_{v:\|v\|=1} \sqrt{\mathbb{E}_x \Delta_x^2} \sqrt{\mathbb{E}_x \langle v, x \rangle^2} \\
&\leq |a_w| \sqrt{\mathbb{E}_x \Delta_x^2} \\
&\leq |a_w| 2 \mathbb{E}_\rho [\|a_w w\|].
\end{aligned}
$$

$\square$

*Proof of Lemma D.3.* Define $\Delta_x := (\ell_\rho'(x) - \ell_0'(x))\sigma'(w^T x)$. Using the symmetry of the data for pairs $(x_{\setminus i} + e_i x_i, x_{\setminus i} - e_i x_i)$, we have

$$
\begin{aligned}
&\left\| \frac{1}{a_w} (\nabla_{w_i} L_\rho - \nabla_{w_i} L_0) \right\| \\
&= \|\mathbb{E}_x \Delta_x x_i\| \\
&= \frac{1}{2} \|\mathbb{E}_x (\Delta_{x_{\setminus i} + e_i x_i} - \Delta_{x_{\setminus i} - e_i x_i}) x_i\| \\
&\leq \frac{1}{2} \|\mathbb{E}_x \mathbf{1}(|x_{\setminus i}^T w| \geq |w_i|)(\Delta_{x_{\setminus i} + e_i x_i} - \Delta_{x_{\setminus i} - e_i x_i}) x_i\| + \frac{1}{2} \|\mathbb{E}_x \mathbf{1}(|x_{\setminus i}^T w| \leq |w_i|)(\Delta_{x_{\setminus i} + e_i x_i} - \Delta_{x_{\setminus i} - e_i x_i}) x_i\| \\
&\leq \frac{1}{2} \sup_x |\ell_\rho'(x_{\setminus i} + e_i x_i) - \ell_\rho'(x_{\setminus i} - e_i x_i)| + \mathbb{E}_x \mathbf{1}(|x_{\setminus i}^T w| \leq |w_i|)|\Delta_x|.
\end{aligned}
$$

Here the last line follows from the fact that whenever $|x_{\setminus i}^T w| \geq |w_i|$, we have $\sigma'(w^T(x_{\setminus i} + e_i x_i)) = \sigma'(w^T(x_{\setminus i} - e_i x_i))$, and thus $|\Delta_{x_{\setminus i} + e_i x_i} - \Delta_{x_{\setminus i} - e_i x_i}| \leq |\ell_\rho'(x_{\setminus i} + e_i x_i) - \ell_\rho'(x_{\setminus i} - e_i x_i)|$. Note that the sup is over $x$ on the boolean hypercube.

Now by Lemma D.5, we have $\sup_x |\ell_\rho'(x_{\setminus i} + e_i x_i) - \ell_\rho'(x_{\setminus i} - e_i x_i)| \leq 4(\mathbb{E}_\rho[\|a_w w_i\|])$. Further, by the 2-Lipshitzness of $\ell_\rho'$ with respect to $f_\rho(x)$, (see the proof of Lemma D.5), we have

$$
\begin{aligned}
&\mathbb{E}_x \mathbf{1}(|x^T(w - w_i x_i)| \leq |w_i|)|\Delta_x| \\
&\leq 2\mathbb{E}_x \mathbf{1}(|x^T(w - w_i x_i)| \leq |w_i|)|f_\rho(x)| \\
&\leq 2\mathbb{E}_x \mathbf{1}(|x^T(w - w_i x_i)| \leq |w_i|)\log(d)\mathbb{E}_\rho[\|a_w w\|] + 2\left(\sqrt{d}\mathbb{E}_\rho[\|a_w w\|]\right) \mathbb{P}_x[|f_\rho(x)| \geq \log(d)\mathbb{E}_\rho[\|a_w w\|]] \\
&\leq 2\mathbb{E}_x \mathbf{1}(|x^T(w - w_i x_i)| \leq |w_i|)\log(d)\mathbb{E}_\rho[\|a_w w\|] + d^{-\omega(1)},
\end{aligned}
$$

where the last line follows from McDiarmid's inequality of bounded differences. Thus putting these pieces together,

$$
\begin{aligned}
&\|\nabla_{w_i} L_\rho - \nabla_{w_i} L_0\| \\
&\leq |a_w| \left( 4\mathbb{E}_\rho[\|a_w w_i\|] + 2\log(d)\mathbb{E}_\rho[\|a_w w\|]\mathbb{E}_x \mathbf{1}(|x_{\setminus i}^T w| \leq |w_i|) + d^{-\omega(1)} \right),
\end{aligned}
$$

which yields the lemma. $\square$

We additionally state and prove the following helper lemma that will be used in Phase 1.

**Lemma D.6** (Helper Lemma). *Suppose for some vector $u_t$ and reals $0 \leq Q_t \leq B_t < 1$, we have $\|u_t\|^2 \leq Q_t^2 + \theta B_t^2$. Also suppose that for some vectors $G$ and $\hat{G}$ and some $\chi > \theta^{1/2}$:*

1. *$-u_t^T G \leq \chi \left( Q_t^2 + \theta B_t^2 \right)$*
2. *$\|G - \hat{G}\| \leq \theta B_t$*
3. *$\|G\| \leq O(B_t)$.*
4. *$\eta \leq \theta^2 = o(1)$.*

*Then with $u_{t+1} := u_t - \eta \hat{G}$, we have*

$$
\|u_{t+1}\|^2 \leq (Q_t^2 + \theta B_t^2)(1 + 5\eta\chi).
$$

*Proof.* Define $W_t^2 := Q_t^2 + \theta^{1/2} B_t^2$, such that $B_t \leq W_t \theta^{-1/2}$.

$$
\begin{aligned}
\|u_{t+1}\|^2 &= \|u_t - \eta \hat{G}\|^2 \\
&\leq \|u_t\|^2 - 2\eta u_t^T G + 2\eta \|u_t\| \|G - \hat{G}\| + 2\eta^2 \|G - \hat{G}\|^2 + 2\eta^2 \|G\|^2 \\
&\leq W_t^2 (1 + 2\eta\chi) + 2\eta \|u_t\| \theta B_t + O(\eta^2 B_t^2) \\
&\leq W_t^2 (1 + 2\eta\chi) + 3\eta \|W_t\| \theta B_t \\
&\leq W_t^2 \left(1 + 2\eta\chi + 3\eta\theta^{1/2}\right) \\
&\leq W_t^2 \left(1 + 5\eta\chi\right).
\end{aligned}
$$

$\square$

### D.2 Inductive Lemmas, and Proof of Lemma D.1 assuming Inductive Lemmass

We now give a short sketch of the analysis in Phase 1 used to prove Lemma D.1. Let $\zeta = 1/\log^c(d)$ and $\theta = 1/\log^C(d)$, where $c$ and $C$ are sufficiently large constants. While we will omit stating it explicitly, in all the lemmas henceforth in Section D, it is assumed that first $c$ is chosen to be a sufficiently large constant, and then $C$ is chosen to be sufficiently large in terms of $c$.

Phase 1 will be broken down into two sub-phases, 1a and 1b. The analysis in both sub-phases is quite similar, but our approximation of the gradients will be courser in Phase 1b than in 1a. Phase 1a will last for most of the time (some $T_{1a} = \Theta(\log(d))$ iterations), and and the end of the phase, we will guarantee the existence of a substantial set of "strong" neurons (see Definition D.13) for which $\zeta^{1.5} \|w\| \leq \|w_{\text{sig}}\| \leq \|w\|$. Note that this is a very meaningful guarantee, since at initialization we have $\|w_{\text{sig}}\| \approx \frac{1}{\sqrt{d}} \|w\|$, and $\zeta$ is $1/\text{polylog}(d)$. Phase 1b will last for only some $T_{1b} = \Theta(\log\log(d))$ iterations, enough to guarantee that on some set of strong neurons, we have $\|w_{\text{sig}}\| \geq \|w_\perp + w_{\text{opp}}\|/\zeta$. This will suffice to prove Lemma D.1.

To formalize this, we state three definitions will will be the basis of our inductive analysis for Phase 1. Our goal will to be to show that all neurons are "controlled" or "weakly controlled", meaning $w_{\text{opp}}$ and $w_\perp$ don't grow too large, while a substantial fraction of neurons are "strong", and in these neurons, $w_{\text{sig}}$ grows quickly.

In what follows, we define the rate parameter $\tau := \frac{1}{\sqrt{2\pi}}$ to be the approximate rate at which the neurons near initialization would grow at under the $L_0$ population loss if $|a_w| = \|w\|$. To see this, observe that from Lemma D.4, we have

$$
\begin{aligned}
\frac{-w_{\text{sig}}^T \nabla_{w_{\text{sig}}} L_0}{\|w_{\text{sig}}\|^2} &= \frac{|a_w|}{\|w_{\text{sig}}\|} \frac{\sqrt{2}}{4} \mathbb{P}_\xi \mathbf{1}(|w^T \xi| \leq \sqrt{2} \|w_{\text{sig}}\|) \\
&\approx \frac{|a_w|}{\|w_{\text{sig}}\|} \frac{\sqrt{2}}{4} \frac{2\sqrt{2} \|w_{\text{sig}}\|}{\sqrt{2\pi} \|w\|} \\
&= \frac{1}{\sqrt{2\pi}}.
\end{aligned}
$$

Note that the "$\approx$" approximation step will hold under the conditions that $\|w_{\text{sig}}\| \ll \|w_\perp\| \approx |a_w|$, and that the vector $w_\perp$ is well-spread among its coordinates – that is, none of its coordinates in the standard basis are too large, which could preclude the central limit theorem convergence of $w^T \xi \to \mathcal{N}(0, \|w_\perp\|)$ in distribution. The details of the comparison of the probability over the boolean vector to the analogous probability over a Gaussian vector is fleshed out in Section C.1.

This calculation gives some intuition for two conditions that we will maintain in our definition of controlled neurons for Phase 1a: we should have $|a_w| \approx \|w_\perp\|$, and $w_\perp$ should be well-spread in some sense, which we will enforce by requiring that $w_\perp^{(t)} - w_\perp^{(0)}$ and $\|w_\perp\|_\infty$ are small. We define the following control parameters.

**Definition D.7** (Control Parameters). *Let $T_{1a}$ and $T_{1b}$ be as defined in Definition D.8. Define*

$$
B_t^2 := \begin{cases} \frac{\log^3(d)\theta^2}{d} (1 + 2\eta\tau(1 + 1/\log(d)))^t & t \leq T_{1a}; \\ \frac{\log^3(d)\theta^2}{d} (1 + 2\eta\tau(1 + 1/\log(d)))^{T_{1a}} (1 + 4\eta)^{t - T_{1a}} \zeta^{-2} & T_{1a} \leq t \leq T_{1b}. \end{cases} \tag{D.1}
$$

$$
Q_t^2 := \frac{\log^3(d)\theta^2}{d} \left(1 + \frac{50\eta}{\log(d)}\right)^t.
$$

*Let $C_{WS}$ be the universal constant which is the maximum of the constants in Lemmas C.2, C.3, and C.4.*

**Definition D.8** (Phase 1 Length). *Let $T_{1a}$ be the last time at which we have $B_t^2 \leq \theta^2\zeta^2$, that is,*

$$T_{1a} := \left\lfloor \frac{\log(d) + 2\log(\zeta) - 3\log(\log(d))}{\log(1 + 2\eta\tau(1 + 1/\log(d)))} \right\rfloor = \frac{1}{\eta}\Theta(\log(d)).$$

*Let $T_{1b}$ to be the last time at which we have $B_t^2 \leq \theta^2\zeta^{-600}$, that is,*

$$T_{1b} := T_{1a} + \left\lfloor \frac{\log(\theta^2\zeta^{-598}/B_{T_{1a}}^2)}{\log(1 + 4\eta)} \right\rfloor = T_{1a} + \frac{1}{\eta}\Theta(\log\log(d)).$$

**Definition D.9** (Controlled Neurons). *We say a neuron $(w, a_w)$ is controlled at iteration $t \leq T_{1b}$ if:*

**C1** $\|w_{sig}\|^2 \leq \min(B_t^2, \theta^2\zeta^2)$.

**C2** $\|w_{opp}\|^2 \leq Q_t^2 + \theta B_t^2$

**C3** $|a_w| \in \theta(1 \pm t\eta\zeta)$, and $|a_w| \leq \|w\|$.

**C4** $\|w_\perp - w_\perp^{(0)}\| \leq \theta\zeta^{1/4}\eta t$, and $w_\perp^{(0)}$ is $C_{WS}$-well spread (see Definition C.1).

**C5** $\|w_\perp\|_\infty^2 \leq Q_t^2 + \theta B_t^2$.

In Phase 1b, we will need to consider the case where $\|w_{\text{sig}}\|$ grows larger than $\theta\zeta$ for some neurons. Thus we introduce the following definition of "weakly controlled" neurons.

**Definition D.10** (Weakly Controlled Neurons). *We say a neuron $(w, a_w)$ is weakly controlled at iteration $t \in [T_{1a}, T_{1b}]$ if:*

**W1** $\theta^2\zeta^2 \leq \|w_{sig}\|^2 \leq B_t^2 \leq \theta^2\zeta^{-600}$.

**W2** $\|w_{opp}\|^2 \leq 2\theta B_t^2(1 + 3\eta\zeta)^t \leq 4\theta^2\zeta^2$.

**W3** $\|w\|^2 \geq |a_w|^2 \geq \|w\|^2 - \zeta^{1/2}\theta^2 - 8\eta^2(t - T_{1a})\theta^2\zeta^{-600}$.

**W4** $\|w_\perp\|^2 \leq 2\theta^2(1 + 3\eta\zeta)^t \leq 3\theta^2$.

**W5** *Either we have $\|w_\perp\| \leq \|w_{sig}\|$, or $\|w_\perp\|_\infty \leq \zeta^{C_{BE}10000}\theta(1 + 21C_{BE}\eta)^{t-T_{1a}} \leq \zeta^{1000}\|w_\perp\|_2$, where $C_{BE}$ is the universal constant from Theorem C.5.*

We note the following simple claims which can be verified by plugging in the values $T_{1a}$ and $T_{1b}$, and recalling that $\zeta = \log^{-c}(d)$ and $\theta = \log^{-C}(d)$ for $c$ and $C$ sufficiently large.

**Claim D.11.** *For any $t \leq T_{1b}$, conditions **C2**-**C5** of Definition D.9 imply conditions **W2**-**W5** of Definition D.10.*

**Claim D.12.** *If all neurons are controlled or weakly controlled at time $t$, then*

1. $\mathbb{E}_{\rho_t}[\|a_w w\|] \leq 2\max\left(\theta^2, B_t^2\right) \leq 2\theta^2\zeta^{-600}$.

2. *For any $i \in [d]$, $\mathbb{E}_{\rho_t}[\|a_w w_i\|] \leq 3\max\left(\theta B_t, B_t^2\right)$.*

We now define strong neurons, which is the set of neurons on which $\|w_{\text{sig}}\|$ grows quickly.

**Definition D.13** (Strong Neurons). *We say a neuron $(w, a_w)$ is strong at iteration $t$ if it is controlled or weakly controlled, $w_{sig}^T w_{sig}^{(0)} > 0$, and*

$$\|w_{sig}\|^2 \geq S_t^2 := \frac{\theta^2}{d}\prod_{s\leq t}\left(1 + 2\eta\tau(1 - \epsilon_s)\right)^s,$$

*where*

$$\epsilon_s := \begin{cases} 1 - \frac{1}{C_S} & s \leq C_S\log(800C_{BE})/(\eta\tau); \\ 5\zeta^{1/10} + \frac{200C_{BE}\sqrt{\pi}}{(1+2\eta\tau/C_S)^{s/2}} & C_S\log(800C_{BE})/(\eta\tau) < s \leq T_{1a}; \\ 1 - \frac{1}{20} & T_{1a} \leq s \leq T_{1b}, \end{cases}$$

*and we have defined the universal constant $C_S := \frac{6400}{\sqrt{\pi}}\exp(100C_{WS}^8)$.*

While the definition of a strong neuron is technical, the meaning is that $\|w_{\text{sig}}\|^2$ grows roughly at the rate of $(1 + 2\eta\tau)^t$. Indeed, this is the case when $\epsilon_s$ is small, which is true in the middle range of values $s$ above, which covers most of the iterations. (The fact that $\epsilon_s$ is constant for small $s$ comes from some errors derived in comparing probabilities of events on Boolean vectors to their Gaussian counterparts; see Section C.1).

We have the following implication of the definition of a strong neuron.

**Lemma D.14.** *For $t \leq T_{1a}$, we have $S_t^2 \geq B_t^2 / \log^4(d)$. Thus for a strong neuron, after $T_{1a}$ steps, we have $\|w_{sig}^{(T_{1a})}\| \geq \zeta^{1.5}\theta$. Further, after $T_{1b}$ steps, we have $\|w_{sig}^{(T_{1b})}\| \geq \zeta^{-1}\theta$.*

*Proof.* We check the first statement first.

$$\frac{B_t^2}{S_t^2} \leq \log^3(d) \prod_{s=1}^{t} \frac{1 + 2\eta\tau\left((1 + 1/\log(d))\right)}{1 + 2\eta\tau(1 - \epsilon_s)} \qquad \text{(Defs. D.13, D.7)}$$

$$\leq \log^3(d) \prod_{s=1}^{t} \left(1 + 2\eta\tau\left((1 + 1/\log(d))\right)\right)\left(1 - 2\eta\tau(1 - \epsilon_s) + 4\eta^2\tau^2\right)$$

$$\left(\tfrac{1}{1+q} \leq 1 - q + q^2 \quad \forall q > 0\right)$$

$$\leq \log^3(d) \prod_{s=1}^{t} \left(1 + 4\eta\tau\left(1/\log(d) + \epsilon_s\right)\right) \qquad (\eta = o(1))$$

$$\leq \log^3(d) \exp\left(4\eta\tau \sum_{s=1}^{t} 1/\log(d) + \epsilon_s\right) \qquad (1 + q \leq e^q \quad \forall q > 0)$$

$$= \Theta\left(\log^3(d)\right) \leq \log^4(d).$$

Here the last line follows from the fact that $\epsilon_s$ is constant for $\Theta(1/\eta)$ iterations, and then it is exponentially decaying down to a minimum of $5\zeta^{1/10}$. Thus since $t \leq T_{1a} = \Theta(\log(d)/\eta)$, the sum is $\Theta(1/\eta)$.

Thus after $T_{1a}$ steps, since $\zeta = o(\log^{-4}(d))$, we have

$$\|w_{\text{sig}}^{(T_{1a})}\| \geq \frac{B_t}{\log^2(d)} \geq \frac{\zeta\theta}{2\log^2(d)} \geq \zeta^{1.5}\theta.$$

For the second statement, we consider $\|w_{\text{sig}}^{(T_{1b})}\|$. Observe that

$$\frac{B_{T_{1b}}^2}{S_{T_{1b}}^2} = \frac{B_{T_{1a}}^2}{S_{T_{1a}}^2}\zeta^{-2} \prod_{s=T_{1a}+1}^{T_{1b}} \frac{1 + 4\eta}{1 + \eta\tau/10} \qquad \text{(Defs. D.13, D.7)}$$

$$\leq \frac{B_{T_{1a}}^2}{S_{T_{1a}}^2}\zeta^{-2} \prod_{s=T_{1a}+1}^{T_{1b}} (1 + 4\eta)^{0.99}$$

$$= \frac{B_{T_{1b}}^2}{S_{T_{1a}}^2}\left(\frac{B_{T_{1b}}^2}{B_{T_{1a}}^2}\right)^{-0.01}\zeta^{-0.02} \qquad \text{(Def. D.7)}$$

$$\leq \frac{B_{T_{1b}}^2}{S_{T_{1a}}^2}\zeta^{5.5}, \qquad \text{(Def. D.8)}$$

and thus $S_{T_{1b}}^2 \geq S_{T_{1a}}^2 \zeta^{-5.5} \geq (\theta^2\zeta^3)\zeta^{-5.5} \geq \theta^2\zeta^{-2}$. It follows that $\|w_{\text{sig}}^{(T_{1b})}\| \geq \theta\zeta^{-1}$. $\qquad \square$

The first main lemma for Phase 1 is the following inductive step.

**Lemma D.15** (Controlled Neurons Inductive Step). *Suppose for some $t \leq T_{1b}$, all neurons are controlled or weakly controlled. Then with probability at least $1 - d^{-\omega(1)}$, for any neuron $(w^{(t)}, a_w^{(t)})$ which is controlled, at step $t + 1$:*

1. *The neuron $(w^{(t+1)}, a_w^{(t+1)})$ is either controlled or weakly controlled.*

2. *If $(w^{(t)}, a_w^{(t)})$ is strong, then $(w^{(t+1)}, a_w^{(t+1)})$ is strong.*

The following is our main inductive step for the second half of Phase 1.

**Lemma D.16** (Weakly Controlled Neurons Inductive Step). *Suppose for some $T_{1a} \leq t \leq T_{1b}$, all neurons are controlled weakly controlled. Then with probability at least $1 - d^{-\omega(1)}$, at step $t + 1$, all neurons are controlled or weakly controlled, and any strong neuron remains strong.*

We defer the proofs of Lemmas D.15 and D.16 to the following subsection. Assuming Lemma D.16 we can now prove Lemma D.1, which we restate here for the readers convenience.

**Lemma D.1** (Output of Phase 1; Formal). *For any constants $c$ sufficiently large, and $C$ sufficiently large in terms of $c$, for any $d$ large enough, the following holds. Let $\theta := 1/\log(d)^C$.*

*Suppose we train a 2-layer neural network with minibatch SGD as in Section 2.2 with a minibatch size of $m \geq d/\theta^2$, width $1/\theta \leq p \leq d^C$, and step size $\eta \leq \theta$, and initialization scale $\theta$. Then with probability at least $1 - \theta$, after some $T_1 = \Theta(\log(d)/\eta)$ steps of minibatch SGD, the network $\rho_{T_1}$ satisfies:*

1. *$\mathbb{E}_{\rho_{T_1}}[\|a_w w\|] \leq 1$;*
2. *$\mathbb{E}_{\rho_{T_1}}[\|w_\perp + w_{opp}\|^2] \leq 4\theta^2$;*
3. *For all $\mu \in \{\pm\mu_1, \pm\mu_2\}$, on at least a $0.1$ fraction of the neurons, we have $\|w_{sig}\| \geq \log(d)^c\theta$ and $w_{sig}^T\mu > 0$.*

*Additionally,*
$$\mathbb{E}_{\rho_{T_1}}[\|w\|^2] \leq \mathbb{E}_{\rho_{T_1}}[|a_w|^2] + \sqrt{\eta},$$
*and for all neurons, we have $|a_w| \leq \|w\|$.*

*Proof of Lemma D.1.* Choose $T_1 = T_{1a} + T_{1b}$. First we confirm that at initialization, all neurons are controlled (Definition D.9), and at least a $0.1$ fraction of neurons are strong (Definition D.13). Indeed for any neuron $w$, by the Guassian CDF, with probability $1 - d^{-\omega(1)}$, we have $|w_i| \leq \theta\frac{\log(d)^2}{\sqrt{d}}$. Taking a union bound over all $i$ and over all at most polynomially many neurons yields properties **C1**, **C2**, and **C5** for all neurons. Property **C3** holds for all neurons since $a_w$ is initialized to have norm 1. Finally, with probability $1 - d^{-\omega(1)}$, by Lemma C.2, all neurons are well-spread at initialization, yielding **C4**. Now we can applying Chernoff's bound to bound the number of neurons for which $\|w_{\text{sig}}^{(0)}\| \geq \frac{\theta}{\sqrt{d}}$, yielding a $0.1$ fraction of neurons with $\|w_{\text{sig}}^{(0)}\| \geq \frac{\theta}{\sqrt{d}}$ and $w_{\text{sig}}^T\mu > 0$ for each $\mu \in \{\pm\mu_1, \pm\mu_2\}$.

Now with probability $1 - d^{-\omega(1)}$, after $T_{1a}$ steps, Lemma D.15 guarantees that we have a network for which all of the neurons are controlled or weakly controlled (D.10), and for each $\mu \in \{\pm\mu_1, \pm\mu_2\}$, at least a $0.1$ fraction of the neurons are strong and $w_{\text{sig}}^T\mu > 0$.

Now applying Lemma D.16 $T_{1b} - T_{1a}$ times yields that with probability $1 - d^{-\omega(1)}$, after $T_{1b}$ steps, for each $\mu$, we have at least a $0.1$ fraction of the neurons are strong and $w_{\text{sig}}^T\mu > 0$, and all neurons are controlled or weakly controlled.

We can now conclude the first item of the lemma, which bounds $\mathbb{E}_{\rho_{T_1}}[\|a_w w\|]$ from properties **W1**, **W2**, and **W3**, **W4**. The second item, which bounds $\mathbb{E}_{\rho_{T_1}}[\|w_\perp + w_{\text{opp}}\|^2]$ follows from properties **W2** and **W4**. Note, if any of the neurons are controlled (instead of weakly controlled), the same bounds hold from the respective properties of controlled neurons. The third item follows by considering the set of strong neurons, and observing that on strong neurons, by Lemma D.14, we have
$$\|w_{\text{sig}}\| \geq \zeta^{-1}\theta \geq \log(d)^c\theta.$$

Finally, the additional clause follows from properties **C3** and **W3**. $\qquad\square$

### D.3 PROOF OF INDUCTIVE LEMMAS (LEMMAS D.15 AND D.16)

To prove Lemma D.15, we will need to compute several bounds on the $L_0$ population gradients on controlled neurons. Recall that most of the gradients have been computed already in Lemma D.4; the following lemma just gives some additional bounds that hold for controlled neurons.

**Lemma D.17** (Phase 1a $L_0$ Population Gradients Bounds). *If all neurons in the network are controlled or weakly controlled at some step $t \leq T_{1b}$, then for any controlled neuron $(w, a_w)$,*

**A1** $\|\nabla_{w_\perp} L_0\| \leq 3\min\left(\sqrt{\theta B_t}, \theta\zeta^{1/2}\right)$.

**A2** $|\nabla_{a_w} L_0| \leq 4\theta\min\left(\sqrt{\theta B_t}, \theta\zeta^{1/2}\right)$.

**A3** *For any $i \in [d]$, $\|\nabla_{w_i} L_0\| \leq \frac{1}{2}\min\left(B_t, \theta\zeta\right)$.*

**A4** *For any $i \in [d]$, $\|\nabla_{w_i} L_0 - \nabla_{w_i} L_\rho\| \leq \frac{1}{2}\theta B_t$.*

*Proof.* We begin with **A1**. Recalling that $x = z + \xi$ for $z = x_{1:2}$, we have

$$\frac{1}{a_w} \nabla_{w_\perp} L_0 = \mathbb{E}_x - y(x)\sigma'(w^T x)\xi$$

$$= -\mathbb{E}_x y(z)\sigma'(w^T \xi)\xi + \mathbb{E}_x y(z)(\sigma'(w^T x) - \sigma'(w^T \xi))\xi$$

$$= \mathbb{E}_x y(x)(\sigma'(w^T x) - \sigma'(w^T \xi))\xi,$$

since $y(z)$ is independent of $\xi$.

Now consider the norm of $\mathbb{E}_x y(x)(\sigma'(w^T x) - \sigma'(w^T \xi))\xi$. We have

$$\|\mathbb{E}_x y(x)(\sigma'(w^T x) - \sigma'(w^T \xi))\xi\| = \sup_{v:\|v\|=1} \mathbb{E}_x y(x)(\sigma'(w^T x) - \sigma'(w^T \xi))\xi^T v$$

$$\leq \sqrt{\mathbb{E}_x(\sigma'(w^T x) - \sigma'(w^T \xi))^2}\sqrt{\mathbb{E}_\xi(v^T \xi)^2}$$

$$= \sqrt{\mathbb{E}_x \mathbf{1}(|\xi^T w| \leq |z^T w|)}$$

$$\leq \sqrt{\mathbb{P}_\xi[|\xi^T w| \leq \sqrt{2}\|w_{1:2}\|]}.$$

Now since $(w, a_w)$ is controlled, the property **C4** and Lemma C.4 (plugging in $v = w_\perp^{(0)}$, $\Delta = w_\perp^{(t)} - w_\perp^{(0)}$, and $a = \frac{\sqrt{2}\|w_{1:2}\|}{\|v\|} \ll 1$) guarantees that

$$\mathbb{P}_\xi[|\xi^T w| \leq \sqrt{2}\|w_{1:2}\|] \leq 2\mathbb{P}_{G \sim \mathcal{N}(0,1)}[|G| \leq a] + 200 C_{\text{BE}} d^{-1/2}$$

$$\leq \frac{\|w_{1:2}\|}{\theta} + 200 C_{\text{BE}} d^{-1/2}.$$

Thus we have

$$\|\nabla_{w_\perp} L_0\| \leq 2\sqrt{\theta\|w_{1:2}\|} + 15|a_w|\sqrt{C_{\text{BE}}} d^{-1/4} \leq 3\min\left(\sqrt{\theta B_t}, \theta \zeta^{1/2}\right),$$

where we have used the fact that since the neuron is controlled, $|a_w| \leq 2\theta$.

Next consider **A3**. By Lemma D.4, we have

$$\|\nabla_{w_i} L_0\| \leq \frac{1}{4}|a_w|\left(\mathbb{P}_x \mathbf{1}(|w^T x_{\backslash i}| \leq |w_i|)\right)$$

$$\leq \frac{|a_w|}{4}\left(\frac{|w_i|}{\|w^{(0)}\|} + 200 C_{\text{BE}} d^{-1/2}\right)$$

$$\leq \frac{1}{3}|w_i| + 50 C_{\text{BE}} d^{-1/2}\theta$$

$$\leq \frac{1}{2}\min(B_t, \theta\zeta),$$

where the third to last line follows from Lemma C.4 and plugging in the Gaussian density $P_a := \mathbb{P}_{G \sim \mathcal{N}(0,1)}[|G| \leq a] \leq \sqrt{\frac{2}{\pi}}a$. The final inequality follows from **C1**, **C2**, and **C5**.

Next consider **A2**. Combining **A3** and **A1**, we have

$$|\nabla_{a_w} L_0| = |w^T \nabla_w L_0|$$

$$\leq |w_\perp^T \nabla_{w_\perp} L_0| + |w_{1:2} \nabla_{w_\perp} L_0|$$

$$\leq \|w\|\|\nabla_{w_\perp} L_0\| + \|w_{1:2}\|\|\nabla_{w_{1:2}} L_0\|$$

$$\leq 3\|w\|\min\left(\sqrt{\theta B_t}, \theta\zeta^{1/2}\right) + \|w\|\min(B_t, \theta\zeta)$$

$$\leq 4\theta\min\left(\sqrt{\theta B_t}, \theta\zeta^{1/2}\right).$$

Finally consider **A4**. Applying Lemma D.3 and then Claim D.12 yields

$$\|\nabla_{w_i} L_\rho - \nabla_{w_i} L_0\| \tag{D.2}$$

$$\leq |a_w|\left(4(\mathbb{E}_\rho[\|a_w w_i\|]) + 2\log(d)\mathbb{E}_\rho[\|a_w w\|]\mathbb{E}_\xi \mathbb{E}_z \mathbf{1}(|x_{\backslash i}^T w| \leq |w_i|) + d^{-\omega(1)}\right)$$

$$\leq 2\theta\left(12 B_t \max(\theta, B_t) + 4\log(d)\max(\theta^2, B_t^2)\mathbb{E}_\xi \mathbb{E}_z \mathbf{1}(|x_{\backslash i}^T w| \leq |w_i|) + d^{-\omega(1)}\right).$$

Now $w_\perp^{(0)}$ is well-spread, and $\|w_\perp^{(t)} - w_\perp^{(0)}\| \le o(1)\|w_\perp^{(0)}\|$ by the definition of controlled. Letting $v = w_\perp^{(0)}$ and $\Delta = w_\perp^{(t)} - w_\perp^{(0)}$, plugging in Lemma C.4 we obtain

$$
\begin{aligned}
\mathbb{E}_x \mathbb{E}_z \mathbf{1}(|x_{\backslash i}^T w| \le |w_i|) &\le \mathbb{P}_{G \sim \mathcal{N}(0,1)}\left[|G| \le \frac{|w_i|}{\|w - e_i w_i\|}\right](1 + o(1)) + 200 C_{\mathrm{BE}} d^{-1/2} \\
&\le \frac{|w_i|}{\|w\|} + 200 C_{\mathrm{BE}} d^{-1/2} \\
&\le \frac{2\min(B_t, \theta\zeta)}{\theta} \\
&\le \frac{2B_t}{\max(\theta, B_t)}
\end{aligned}
$$

where the second to last line follows from the definition of controlled and of $B_t$ (eq. D.1). Thus returning to Equation D.2, and using Claim D.12, we have

$$
\|\nabla_{w_i} L_\rho - \nabla_{w_i} L_0\| \le \Theta(\log(d))\left(\theta B_t \max(\theta, B_t) + \theta B_t \max(\theta, B_t)\right) \le \theta B_t/2,
$$

since $\max(\theta, B_t) = o(1/\log(d))$.

$\square$

We are now ready to prove the inductive step, Lemma D.15.

*Proof of Lemma D.15.* Suppose that $(w^{(t)}, a_w^{(t)})$ is controlled. Our first goal will be to show that for $(w^{(t+1)}, a_w^{(t+1)})$, items **C3**, **C4**, **C2**, and **C5** hold. We will handle $\|w_{\mathrm{sig}}\|$ at the end.

Next we prove that **C3**. To prove that $|a_w| \in \theta(1 \pm t\eta\zeta)$ holds at the $(t+1)$th step, we have with probability $1 - d^{-\omega(1)}$

$$
\begin{aligned}
|a_w^{(t+1)} - a_w^{(t)}| &= |\eta\nabla\hat{L}_\rho a_w^{(t)}| \\
&\le \eta|\nabla_{a_w} L_0| + \eta|\nabla_{a_w} L_\rho - \nabla_{a_w}\hat{L}_\rho| + \eta|\nabla_{a_w} L_0 - \nabla_{a_w} L_\rho| \\
&\le \eta\left(4\theta^2\zeta^{1/2} + \|w\|\sqrt{\frac{d\log(d)^2}{m}} + 4\|w\|\mathbb{E}_\rho[\|a_w w\|]\right) \\
&\qquad\qquad \text{(Lemma D.17 (item \textbf{A2}), C.12, and D.2 respectively)} \\
&\le \eta\theta\zeta. \qquad\qquad\qquad\qquad\qquad\qquad (\tfrac{d}{m} \le \theta^2, \text{Claim D.12})
\end{aligned}
$$

Here the final inequality we also used the definition of controlled to bound $\|w\|$. With probability $1 - d^{-\omega(1)}$, by Lemma C.13 **S3**, we have $|a_w^{(t+1)}| \le \|w^{(t+1)}\|$.

To prove that **C4**, which states that $\|w_\perp - w_\perp^{(0)}\| \le \theta\zeta^{1/4}\eta t$, continues to hold, we similarly have with probability $1 - d^{-\omega(1)}$,

$$
\begin{aligned}
\|w_\perp^{(t+1)} - w_\perp^{(t)}\| &= \|\eta\nabla_{w_\perp}\hat{L}_\rho\| \\
&\le \eta\|\nabla_{w_\perp} L_0\| + \eta\|\nabla_w L_\rho - \nabla_w\hat{L}_\rho\| + \eta\|\nabla_w L_0 - \nabla_w L_\rho\| \\
&\le \eta\left(3\theta\zeta^{1/2} + |a_w|\sqrt{\frac{d\log(d)^2}{m}} + 2|a_w|\mathbb{E}_\rho[\|a_w w\|]\right) \\
&\qquad\qquad \text{(Lemma D.17 (item \textbf{A1}), C.12, and D.2 respectively)} \\
&\le \eta\theta\zeta^{1/4}. \qquad\qquad\qquad\qquad (\textbf{C3}, \tfrac{d}{m} \le \theta^2, \text{Claim D.12})
\end{aligned}
$$

Next we check that item **C2**, which states that $\|w_{\mathrm{opp}}\|^2 \le Q_t^2 + \theta B_t^2$, continues to hold. We will use the helper lemma, Lemma D.6. Observe that we have:

1. $-w_{\mathrm{opp}}^T \nabla_w L_0 \le 0$ by Lemma D.4.

2. $\|\nabla_{w_{1:2}} L_0 - \nabla_{w_{1:2}}\hat{L}_\rho\| \le \theta B_t$ with probability $1 - d^{-\omega(1)}$. This follows from combining Lemma D.17 item **A4** (applied twice to $i = 1, 2$) with Lemma C.12.

3. $\|\nabla_{w_{\mathrm{opp}}} L_0\| \le B_t$ by Lemma D.17 item **A3**.

Then calling Lemma D.6 with $u_t = w_{\text{opp}}$, $G = \nabla_{w_{1:2}} L_0$, $\hat{G} = \nabla_{w_{1:2}} \hat{L}_\rho$, and $Q_t$ and $B_t$ as in Definition D.7, we achieve

$$\|w_{\text{opp}}^{(t+1)}\|^2 \le (Q_t^2 + \theta B_t^2)(1 + 5\eta\theta^{1/2}) \le Q_{t+1}^2 + \theta B_{t+1}^2,$$

as desired.

Finally we check that **C5**, which states that $\|w_\perp\|_\infty^2 \le Q_t^2 + \theta B_t^2$, continues to hold by showing that for any $i \in [3, d]$, $w_i^2$ cannot grow too quickly. From Lemma D.4 item **B4**, with $X = (\xi - e_i\xi_i)^T w$, we have that

$$-w_i^T \nabla_{w_i} L_0 = \frac{|a_w||w_i|}{4} \left( \mathbb{P}_\xi \left[ X \in [\sqrt{2}\|w_{\text{opp}}\| - |w_i|, \sqrt{2}\|w_{\text{opp}}\| + |w_i|] \right] - \mathbb{P}_\xi \left[ X \in [\sqrt{2}\|w_{\text{sig}}\| - |w_i|, \sqrt{2}\|w_{\text{sig}}\| + |w_i|] \right] \right)$$

Now employing Lemma C.4 twice with $v = w_\perp^{(0)}$, $\Delta = w_\perp^{(t)} - w_\perp^{(0)} - e_i w_i$ both times, and $[a\|v\|, b\|v\|]$ as the intervals $\left[\sqrt{2}\|w_{\text{sig}}\| - |w_i|, \sqrt{2}\|w_{\text{sig}}\| - |w_i|\right]$ and $\left[\sqrt{2}\|w_{\text{opp}}\| - |w_i|, \sqrt{2}\|w_{\text{opp}}\| - |w_i|\right]$, we obtain that

$$-w_i^T \nabla_{w_i} L_0 \le |a_w||w_i| P_{\frac{|w_i|}{\|w_\perp^{(0)}\|}} \left( \sqrt{2\eta t \zeta^{1/4} + \frac{\log(d)^2}{\sqrt{d}}} + 2|a_w| \frac{\|w_{1:2}\|^2 + \|w_\perp\|_\infty^2}{\|w_\perp^{(0)}\|} \right) + 200 C_{\text{BE}} d^{-1/2} |a_w||w_i|,$$

where $P_c := \mathbb{P}_{G \sim \mathcal{N}(0,1)}[|G| \le c]$. Here we have used the fact that $\frac{\|w_\perp^{(t)} - w_\perp^{(0)} - e_i w_i\|}{\|w_\perp^{(0)}\|} \le 2\eta t \zeta^{1/4} + \frac{\log(d)^2}{\sqrt{d}}$ by **C4** and **C5**, which serves as the role of $\zeta$ in Lemma C.4.

Simplifying this expression, and observing that the second term in the parenthesis becomes insignificant by **C1**-**C5**, we have that

$$-w_i^T \nabla_{w_i} L_0 \le -\frac{|a_w||w_i|^2}{\|w_\perp\|} \zeta^{1/9} + 200 C_{\text{BE}} d^{-1/2} |a_w||w_i|.$$

Thus since $|a_w| \le \|w\| \le 2\theta$, and $\|w_\perp\| \ge \theta/2$ (by **C3** and **C4**), we have

$$w_i^T \nabla_{w_i} L_0 \le 2|w_i|^2 \zeta^{1/9} + 200 C_{\text{BE}} d^{-1/2} \theta |w_i| \le 2\zeta^{1/9}|w_i|^2 + \frac{Q_0}{\log(d)} |w_i|.$$

Now we proceed via Lemma D.6. Observe that for any $i \in [3, d]$, we have:

1. $-w_i^T \nabla_w L_0 \le (Q_t^2 + \theta B_t^2)\left(2\zeta^{1/9} + \frac{1}{\log(d)}\right)$ by the calculation above and the assumption that $w$ is controlled.
2. $\|\nabla_{w_i} L_0 - \nabla_{w_i} \hat{L}_\rho\| \le \theta B_t$ with probability $1 - d^{-\omega(1)}$. This follows from combining Lemma D.17 item **A4** with Lemma C.12.
3. $\|\nabla_{w_i} L_0\| \le B_t/2$ by Lemma D.17 item **A3**.

Then calling Lemma D.6 with $u_t = w_{\text{opp}}$, $G = \nabla_{w_{1:2}} L_0$, $\hat{G} = \nabla_{w_{1:2}} \hat{L}_\rho$, and $Q_t$ and $B_t$ as in Definition D.7, we achieve with probability $1 - d^{-\omega(1)}$,

$$\|w_i^{(t+1)}\|^2 \le (Q_0^2 + \theta B_t^2) \left(1 + 5\eta\left(2\zeta^{1/9} + \frac{1}{\log(d)}\right)\right) \le Q_{t+1}^2 + \theta B_{t+1}^2,$$

as desired.

Finally we check the growth of $\|w_{\text{sig}}\|$, which we will use to show that either **C1** or **W1** holds at step $t + 1$, and that a strong neuron stays strong. We have by Lemma D.17 item **B1** that

$$\|w_{\text{sig}}^{(t+1)}\|^2 = \|w_{\text{sig}}^{(t)} - \eta\mu\mu^T \nabla\hat{L}_\rho(w^{(t)})\|^2 \tag{D.3}$$

$$= \|w_{\text{sig}}\|^2 - 2\eta w_{\text{sig}}^T \nabla_w \hat{L}_\rho + \eta^2 \|\mu\mu^T \nabla_w \hat{L}_\rho\|^2$$

$$= \|w_{\text{sig}}\|^2 - 2\eta w_{\text{sig}}^T \nabla_w L_0 - 2\eta w_{\text{sig}}^T (\nabla_w \hat{L}_\rho - \nabla_w L_0) + \eta^2 \|\mu\mu^T \nabla_w \hat{L}_\rho\|^2$$

$$= \|w_{\text{sig}}\|^2 + 2\eta \frac{\sqrt{2}}{4} |a_w| \|w_{\text{sig}}\| X - 2\eta w_{\text{sig}}^T (\nabla_w \hat{L}_\rho - \nabla_w L_0) + \eta^2 \|\mu\mu^T \nabla_w \hat{L}_\rho\|^2,$$

where $X := \mathbb{E}_\xi \mathbf{1}(|w^T \xi| \le \sqrt{2}\|w_{\text{sig}}\|)$.

Now with probability $1 - d^{-\omega(1)}$, we can control the last two terms as follows.

$$| - 2\eta w_{\text{sig}}^T (\nabla_w \hat{L}_\rho - \nabla_w L_0) + \eta^2 \|\mu\mu^T \nabla_w \hat{L}_\rho\|^2| \leq 2\eta \|w_{\text{sig}}\| \|\nabla_{w_{1:2}} \hat{L}_\rho - \nabla_{w_{1:2}} L_0\| + \eta^2 \|\mu\mu^T \nabla_w \hat{L}_\rho\|^2 \quad (\text{D.4})$$
$$\leq 2\eta \|w_{\text{sig}}\| \|\nabla_{w_{1:2}} \hat{L}_\rho - \nabla_{w_{1:2}} L_0\|$$
$$+ 2\eta^2 \|\nabla_{w_{1:2}} \hat{L}_\rho - \nabla_{w_{1:2}} L_0\|^2 + 2\eta^2 \|\nabla_{w_{1:2}} L_0\|^2.$$

Now we have

$$\|\nabla_{w_{1:2}} \hat{L}_\rho - \nabla_{w_{1:2}} L_0\| \leq \|\nabla_{w_{1:2}} \hat{L}_\rho - \nabla_{w_{1:2}} L_\rho\| + \|\nabla_{w_{1:2}} L_\rho - \nabla_{w_{1:2}} L_0\|$$
$$\leq \theta B_t / 2 + \theta B_t / 2 \leq \theta B_t,$$

by Lemma C.12 and Lemma D.4 **A4**. Second, we have $\|\nabla_{w_{1:2}} L_0\|^2 \leq B_t^2$ by Lemma D.4 **A3**. Plugging these two bounds back into Eq. D.4 yields

$$| - 2\eta w_{\text{sig}}^T (\nabla_w \hat{L}_\rho - \nabla_w L_0) + \eta^2 \|\mu\mu^T \nabla_w \hat{L}_\rho\|^2| \leq 2\eta \|w_{\text{sig}}\| \theta B_t + 4\eta^2 B_t^2 \leq 6\eta\theta B_t^2, \quad (\text{D.5})$$

where we have used the fact that $\|w_{\text{sig}}\| \leq B_t$ (property **C1**) and that $\eta \leq \theta$. Now by Lemma C.4, we have

$$X \leq (1 + 3\zeta^{1/10})\mathbb{P}_{G \sim \mathcal{N}(0,1)} \left[ |G| \leq \frac{\sqrt{2}\|w_{\text{sig}}\|}{\|w_\perp^{(0)}\|} \right] + 200 C_{\text{BE}} d^{-1/2} \quad (\text{D.6})$$

$$\leq (1 + 3\zeta^{1/10}) \frac{2\sqrt{2}}{\sqrt{2\pi}} \frac{\|w_{\text{sig}}\|}{\|w_\perp^{(0)}\|} + 200 C_{\text{BE}} d^{-1/2}$$

(Upper bound Gaussian density by density at 0.)

$$\leq \frac{B_t}{\theta} \frac{2}{\sqrt{\pi}} \left( 1 + 3\zeta^{1/10} + \frac{1}{\log^{1.1}(d)} \right) \qquad \text{(Definition D.7 of $B_t$ and C1)}$$

$$\leq \frac{B_t}{\theta} \frac{2}{\sqrt{\pi}} \left( 1 + \frac{2}{\log^{1.1}(d)} \right) \qquad \text{($\zeta = \log^{-c}(d)$ for $c$ large enough.)}$$

Here we have used Lemma C.4 with $v = w_\perp^{(0)}$, $\Delta = w_\perp - w_\perp^{(0)}$, and $\zeta = \frac{w_\perp - w_\perp^{(0)}}{w_\perp^{(0)}} \leq \zeta^{1/5}$ (by **C4**), and $b = -a = \frac{\|w_{\text{sig}}\|\sqrt{2}}{\|w_\perp^{(0)}\|}$.

To check that either **C1** in the definition of controlled continues to hold at time $t + 1$, so that $w^{(t+1)}$ is controlled (Def. D.9) *or* that item **W1** in the definition of weakly controlled (Def. D.10), it suffices to check that $\|w_{\text{sig}}^{(t+1)}\|^2 \leq B_{t+1}^2$. Indeed from combining Eqs. D.3, D.5, D.6 we have with probability $1 - d^{-\omega(1)}$,

$$\|w_{\text{sig}}^{(t+1)}\|^2 \leq \|w_{\text{sig}}\|^2 + \frac{1}{\sqrt{2}} \eta \|w_{\text{sig}}\| |a_w| X + 6\eta\theta B_t^2$$

$$\leq B_t^2 + 2\eta\tau B_t^2 \left( 1 + \frac{2}{\log^{1.1}(d)} \right) + 6\eta\theta B_t^2$$

$$\leq B_t^2 \left( 1 + 2\eta\tau \left( 1 + \frac{1}{\log(d)} \right) \right)$$

$$\leq B_{t+1}^2.$$

Here the second inequality we used the fact that $\frac{|a_w|}{\theta} \leq 1 + \Theta(\log(d)\zeta)$ by **C3**.

Now we check that if the neuron is strong, it stays strong. To do this, we need to lower bound $X = \mathbb{E}_\xi \mathbf{1}(|w^T \xi| \leq \sqrt{2}\|w_{\text{sig}}\|)$. For the case that $\frac{\|w_{\text{sig}}\|}{\|w_\perp\|}$ is small (ie., $t$ is small), we use Lemma C.3, which yields that

$$X \geq \frac{1}{2\exp(100 C_{\text{WS}}^8)\sqrt{d}}.$$

Indeed, we can apply Lemma C.3 wit $v = w_\perp^{(0)}$ and $\Delta = w_\perp - w_\perp^{(0)}$, which by **C4** yields $\frac{\|\Delta\|}{\|v\|} \leq \zeta^{1/5}$.

Recall the definition of $S_t$ from Definition D.13. For $t \leq C_{\mathrm{S}} \log(800C_{\mathrm{BE}})/(\eta\tau)$, we have $S_t = \frac{\theta}{\sqrt{d}}\left(1 + 2\tau\eta/C_{\mathrm{S}}\right)^{t/2} \leq \frac{\theta}{\sqrt{d}} e^{t\tau\eta/C_{\mathrm{S}}} = \frac{\theta}{\sqrt{d}} 800C_{\mathrm{BE}}$, and thus

$$X \geq \frac{1}{2\exp(100C_{\mathrm{WS}}^8)} \frac{1}{800C_{\mathrm{BE}}} \frac{S_t}{\theta} = \frac{2}{C_{\mathrm{S}}}\left(\frac{S_t}{\theta}\frac{2}{\sqrt{\pi}}\right),$$

where we recall that $C_{\mathrm{S}} = \frac{6400}{\sqrt{\pi}}\exp(100C_{\mathrm{WS}}^8)$. Further, from Lemma C.4, applied as in Eq. D.6, we have

$$X \geq (1 - 3\zeta^{1/10})\mathbb{P}_{G\sim\mathcal{N}(0,1)}\left[|G| \leq \frac{\sqrt{2}\|w_{\mathrm{sig}}\|}{\|w_\perp^{(0)}\|}\right] - \frac{200C_{\mathrm{BE}}}{\sqrt{d}}$$

$$\geq (1 - 3\zeta^{1/10})\left(\frac{2}{\sqrt{\pi}}\frac{\|w_{\mathrm{sig}}\|}{\|w_\perp^{(0)}\|}\right)e^{-\frac{2\|w_{\mathrm{sig}}\|^2}{2\|w_\perp^{(0)}\|^2}} - \frac{200C_{\mathrm{BE}}}{\sqrt{d}}$$
$$\text{(Lower bound Gaussian density by endpoints of interval)}$$

$$\geq (1 - 3\zeta^{1/10})\left(\frac{2}{\sqrt{\pi}}\frac{\|w_{\mathrm{sig}}\|}{\|w_\perp^{(0)}\|}\right)\left(1 - \frac{\|w_{\mathrm{sig}}\|^2}{\|w_\perp^{(0)}\|^2}\right) - \frac{200C_{\mathrm{BE}}}{\sqrt{d}} \qquad (e^{-x} \geq 1 - x)$$

$$\geq (1 - 4\zeta^{1/10})\left(\frac{2}{\sqrt{\pi}}\frac{\|w_{\mathrm{sig}}\|}{\|w_\perp^{(0)}\|}\right) - \frac{200C_{\mathrm{BE}}}{\sqrt{d}}. \qquad (\frac{\|w_{\mathrm{sig}}\|^2}{\|w_\perp^{(0)}\|^2} \leq \zeta^2 \text{ by } \mathbf{C1.})$$

Thus from the definition of strong neurons, since $\epsilon_s$ is decreasing for $s \leq T_{1a}$ (see Def. D.13), we have $\frac{\|w_{\mathrm{sig}}\|^2}{d\|w_\perp\|^2} \geq \frac{\|w_{\mathrm{sig}}\|^2}{2d\theta^2} \geq \frac{1}{2}(1 + 2\eta\tau/C_{\mathrm{S}})^t$, and so

$$X \geq \left(1 - 4\zeta^{1/10} - \frac{200C_{\mathrm{BE}}}{\sqrt{d}}\frac{\sqrt{\pi}\|w_\perp\|}{2\|w_{\mathrm{sig}}\|}\right)\left(\frac{2}{\sqrt{\pi}}\frac{\|w_{\mathrm{sig}}\|}{\|w_\perp\|}\right)$$

$$\geq \left(1 - 4\zeta^{1/10} - \frac{200C_{\mathrm{BE}}\sqrt{\pi}}{(1 + 2\eta\tau/C_{\mathrm{S}})^{t/2}}\right)\left(\frac{2}{\sqrt{\pi}}\frac{\|w_{\mathrm{sig}}\|}{\|w_\perp\|}\right).$$

Returning to Eq. D.3, we have that

$$\|w_{\mathrm{sig}}^{(t+1)}\|^2 \geq \|w_{\mathrm{sig}}\|^2 + 2\eta\frac{\sqrt{2}}{4}|a_w|\|w_{\mathrm{sig}}\|X - 6\eta\theta B_t^2.$$

To complete this computation, observe that for $t \leq T_{1a}$, by Lemma D.14, we have $B_t^2 \leq S_t^2 \log^3(d) \leq S_t^2 \theta^{-1/2}$. Thus since $\|w_{\mathrm{sig}}\| \geq S_t$, using the previous lower bounds on $X$, we have

$$\|w_{\mathrm{sig}}^{(t+1)}\|^2 \geq S_t^2\left(1 - 6\eta\theta^{1/2}\right) + 2\eta\frac{\sqrt{2}}{4}|a_w|\|w_{\mathrm{sig}}\|X$$

$$\geq \begin{cases} S_t^2\left(1 - 6\eta\theta^{1/2} + 2\eta\tau\frac{2}{C_{\mathrm{S}}}\frac{|a_w|}{\theta}\right) & t \leq C_{\mathrm{S}}\log(800C_{\mathrm{BE}})/(\eta\tau); \\ S_t^2\left(1 - 6\eta\theta^{1/2} + 2\eta\tau\left(1 - 4\zeta^{1/10} - \frac{200C_{\mathrm{BE}}\sqrt{\pi}}{(1+2\eta\tau/C_{\mathrm{S}})^{t/2}}\right)\frac{|a_w|}{\|w_\perp\|}\right) & t > C_{\mathrm{S}}\log(800C_{\mathrm{BE}})/(\eta\tau); \end{cases}$$

$$\geq \begin{cases} S_t^2\left(1 + \frac{2\eta\tau}{C_{\mathrm{S}}}\right) & t \leq C_{\mathrm{S}}\log(800C_{\mathrm{BE}})/(\eta\tau); \\ S_t^2\left(1 + 2\eta\tau\left(1 - 5\zeta^{1/10} - \frac{200C_{\mathrm{BE}}\sqrt{\pi}}{(1+2\eta\tau/C_{\mathrm{S}})^{t/2}}\right)\right) & t > C_{\mathrm{S}}\log(800C_{\mathrm{BE}})/(\eta\tau); \end{cases}$$

$$= S_t^2(1 + 2\eta\tau(1 - \epsilon_t)),$$

which means that the neuron stays strong.

$\square$

We are now ready to prove Lemma D.16.

*Proof of Lemma D.16.* By Lemma D.15, it suffices to prove that with probability $1 - d^{-\omega(1)}$, all weakly controlled neurons stay weakly controlled, and that all weakly controlled and strong neurons stay strong at iteration $t + 1$.

Suppose $(w^{(t)}, a_w^{(t)})$ is weakly controlled. We begin by showing that with high probability, the five properties **W1**, **W2**, **W3**, **W4**, **W5** of controlled neurons hold at iteration $t + 1$. We will omit the proof that **W2** continues to hold, as this proof is nearly identical to the proof for **C2** in Lemma D.15, which proceeds by applying Lemma D.6

We begin with **W3**. We have with probability $1 - d^{-\omega(1)}$,

$$
\begin{aligned}
|a_w^{(t+1)}|^2 - \|w^{(t+1)}\|^2 &\geq |a_w|^2 - \|w\|^2 - 4\eta^2|a_w|^2 \\
&\geq |a_w|^2 - \|w\|^2 - 4\eta^2\|w\|^2 \\
&\geq |a_w|^2 - \|w\|^2 - 8\eta^2\theta^2\zeta^{-600},
\end{aligned}
$$

Here the first and second inequalities are from Lemma C.13 item **S4** and **C3** respectively, and the third is from combining **W1**, **W2**, and **W4** to bound $\|w\|$. This, in addition to Lemma!C.13 **S3**, yield the desired conclusion since the gap between $a_w^2$ and $\|w\|^2$ cannot grow by more than $8\eta^2\theta^2\zeta^{-600}$ at each step.

Next we show **W4** holds at step $t + 1$ with high probability. Observe that we have:

1. $-w_\perp^T \nabla_w L_0 \leq 0$ by Lemma D.4, since by the definition of weakly controlled, $\|w_{\text{sig}}\| \geq \|w_{\text{opp}}\|$.
2. $\|\nabla_w L_0 - \nabla_w \hat{L}_\rho\| \leq \theta\zeta$ with probability $1 - d^{-\omega(1)}$. This follows from combining Lemmas D.2 and Claim D.12 (which together yield $\|\nabla_w L_0 - \nabla_w L_\rho\| \leq \theta\zeta$ ) with Lemma C.12 (which yields $\|\nabla_w L_\rho - \nabla_w \hat{L}_\rho\| \leq \zeta\theta/2$).
3. $\|\nabla_w L_\rho\| \leq 3B_t$ with probability $1 - d^{-\omega(1)}$ by Lemma C.13 (**S1**) and Lemma D.2.

Note the all the above approximations are very loose. Thus we have

$$
\begin{aligned}
\|w_\perp^{(t+1)}\|^2 &\leq \|w_\perp^{(t)}\|^2 + 2\eta\|w_\perp^{(t)}\|\|\nabla_w L_0 - \nabla_w \hat{L}_\rho\| + \eta^2\|\nabla_w \hat{L}_\rho\|^2 \\
&\leq 2\theta^2(1 + 3\eta\zeta)^t + 2\eta(2\theta)\theta\zeta + 9\eta^2 B_t^2 \\
&\leq 2\theta^2(1 + 3\eta\zeta)^t (1 + 3\eta\zeta) \leq 2\theta^2(1 + 3\eta\zeta)^{t+1},
\end{aligned}
$$

as desired.

We can carry out an almost identical computation to bound $\|w_{\text{opp}}^{(t+1)}\|$ to prove that **W2** continues to hold, so we omit the details.

Next we show that property **W1** holds at step $t + 1$ with high probability. Let $\mu$ be the direction of $w_{\text{sig}} := w_{\text{sig}}^{(t)}$. Following the same steps Lemma D.15, we have

$$
\begin{aligned}
\|w_{\text{sig}}^{(t+1)}\|^2 &= \|w_{\text{sig}} - \eta\mu\mu^T \nabla_w \hat{L}_\rho\|^2 \\
&= \|w_{\text{sig}}\|^2 - 2\eta w_{\text{sig}}^T \nabla_w \hat{L}_\rho + \eta^2\|\mu\mu^T \nabla_w \hat{L}_\rho\|^2 \\
&\in \|w_{\text{sig}}\|^2 - 2\eta w_{\text{sig}}^T \nabla_w L_0 \pm 2\eta\|w_{\text{sig}}\|\|\nabla_w \hat{L}_\rho - \nabla_w L_0\| \pm \eta^2\|\mu\mu^T \nabla_w \hat{L}_\rho\|^2 \\
&\in \|w_{\text{sig}}\|^2 - 2\eta w_{\text{sig}}^T \nabla_w L_0 \pm 6\eta\theta B_t^2.
\end{aligned}
$$

Here the final inequality follows from Eq. D.5 from the proof of Lemma D.15; the difference in assumption that the neurons are weakly controlled and not controlled does not affect this computation.

Now plugging in the $L_0$ population gradient from Lemma D.4, we have

$$
\|w_{\text{sig}}^{(t+1)}\|^2 \in \|w_{\text{sig}}\|^2 - 2\eta w_{\text{sig}}^T \nabla_w L_0 \pm \eta 6\eta\theta B_t^2 \tag{D.7}
$$

$$
\in \|w_{\text{sig}}^{(t)}\|^2 \left(1 + 2\eta\frac{\sqrt{2}}{4}\frac{|a_w|}{\|w_{\text{sig}}\|}\mathbb{P}_\xi[|w^T\xi| \leq \sqrt{2}\|w_{\text{sig}}\|]\right) \pm 6\eta\theta B_t^2.
$$

Now to prove that **W1** holds at time $t + 1$, we upper bound $\mathbb{P}_\xi[|w^T\xi| \leq \sqrt{2}\|w_{\text{sig}}\|]$ by 1 and consider two cases:

1. Case 1: $\|w_{\text{sig}}^{(t)}\| \leq B_t/2$. In this case, since $|a_w| \leq \|w\| \leq 2B_t$ (since the neuron is weakly controlled), we have $\|w_{\text{sig}}^{(t+1)}\|^2 \leq B_t^2 \leq B_{t+1}^2$.

2. Case 2: $\|w_{\text{sig}}^{(t)}\| \geq B_t/2$. In this case, since $t \geq T_{1a}$, we have $B_t \geq \theta(1 - o(1))$ (see Definition D.8), and so $\|w_{\text{sig}}\| \geq |a_w|/4$. Thus we have

$$
\|w_{\text{sig}}^{(t+1)}\|^2 \in \|w_{\text{sig}}^{(t)}\|^2 \left(1 + 2\eta\frac{\sqrt{2}}{4}4\right) \pm 6\eta\theta B_t^2 \leq \|w_{\text{sig}}^{(t)}\|^2(1 + 4\eta) \leq Q_{t+1}^2.
$$

We now show that the if the neurons have the additional strong property at step $t$ (Definition D.13), they continue to have this property at step $t + 1$. To do this, we need to lower bound the probability $\mathbb{P}_\xi[|w^T\xi| \leq \sqrt{2}\|w_{sig}\|]$.

**Claim D.18.** *If $(w, a_w)$ is weakly controlled and strong, then*

$$\mathbb{P}_\xi[|w^T\xi| \leq \sqrt{2}\|w_{sig}\|] \geq \frac{\|w_{sig}\|}{5\|w_\perp + w_{sig}\|}.$$

*Proof.* We will use the Berry-Esseen Inequality (stated in Theorem C.5). We have

$$\mathbb{P}_\xi[|w^T\xi| \leq \sqrt{2}\|w_{sig}\|] \geq \mathbb{P}_{G\sim\mathcal{N}(0,1)}\left[|G| \leq \frac{\sqrt{2}\|w_{sig}\|}{\|w_\perp\|}\right] - C_{BE}\frac{\|w_\perp\|_3^3}{\|w_\perp\|_2^3} \tag{D.8}$$

$$\geq \mathbb{P}_{G\sim\mathcal{N}(0,1)}\left[|G| \leq \frac{\sqrt{2}\|w_{sig}\|}{\|w_\perp\|}\right] - C_{BE}\frac{\|w_\perp\|_\infty}{\|w_\perp\|_2}$$

$$\geq \frac{\|w_{sig}\|}{4\|w_{sig} + w_\perp\|} - C_{BE}\frac{\|w_\perp\|_\infty}{\|w_\perp\|_2}.$$

Now by **W5**, either we have $\|w_\perp\| \leq \|w_{sig}\|$, or $\frac{\|w_\perp\|_\infty}{\|w_\perp\|_2} \leq \zeta^{1000}$.

We first consider the latter case when $\frac{\|w_\perp\|_\infty}{\|w_\perp\|_2} \leq \zeta^3$. By definition of a strong neuron and Lemma D.14, we have

$$\|w_{sig}\| \geq S_t \geq \theta\zeta^{700}.$$

Thus we have from Eq. D.8

$$\mathbb{P}_\xi[|w^T\xi| \leq \sqrt{2}\|w_{sig}\|] \geq \frac{\|w_{sig}\|}{5\|w_{sig} + w_\perp\|}.$$

Now if $\|w_\perp\| \leq \|w_{sig}\|$, then we have by Chebychev's inequality,

$$\mathbb{P}_\xi[|w^T\xi| \leq \sqrt{2}\|w_{sig}\|] \geq 1 - \mathbb{P}_\xi[|w^T\xi| \geq \sqrt{2}\|w_{sig}\|]$$

$$\geq 1 - \frac{\mathbb{E}_\xi[(w_\perp^T\xi)^2]}{2\|w_{sig}\|^2}$$

$$= 1 - \frac{\|w_\perp\|^2}{2\|w_{sig}\|^2} \geq \frac{1}{2} \geq \frac{\|w_{sig}\|}{5\|w_{sig} + w_\perp\|}.$$

$\square$

It follows from Claim D.18, Eq. D.7, and the definition of weakly controlled that

$$\|w_{sig}^{(t+1)}\|^2 \geq \|w_{sig}^{(t)}\|^2\left(1 + 2\eta\frac{\sqrt{2}}{4}\frac{|a_w|}{\|w_{sig}\|}\frac{\|w_{sig}\|}{5\|w_{sig} + w_\perp\|}\right) - 6\eta\theta B_t^2$$

$$\geq \|w_{sig}^{(t)}\|^2\left(1 + 2\eta\frac{\sqrt{2}}{4}\frac{\|w_{sig} + w_\perp\|/2}{5\|w_{sig} + w_\perp\|}\right) - 6\eta\theta B_t^2 \qquad (|a_w| \geq \|w\|/2 \text{ by } \mathbf{W3})$$

$$\geq \|w_{sig}^{(t)}\|^2\left(1 + \eta\frac{\sqrt{2}}{20}\right) - \eta\theta^{1/2}\|w_{sig}^{(t)}\|^2$$

$$(\|w_{sig}\| \geq S_t \geq B_t\zeta^{\Theta(1)} \text{ by Lemma D.14, and } C \text{ is large enough in terms of } c.)$$

$$\geq \|w_{sig}^{(t)}\|^2\left(1 + \frac{\eta}{20}\right),$$

which means that the neuron stays strong.

We now check that **W5** continues to hold. If $\|w_\perp^{(t)}\| \leq \|w_{sig}^{(t)}\|$, then it is a routine calculation to verify from the above computation and Lemma D.4 (and the associated [high probability] bounds

on the gradients in Lemmas C.13, C.12 that with probability $1 - d^{-\omega(1)}$, $\|w_\perp^{(t)}\|$ grows at a slower rate than $\|w_{\text{sig}}^{(t)}\|$, and thus $\|w_\perp^{(t+1)}\| \leq \|w_{\text{sig}}^{(t+1)}\|$. We omit the details of the calculation.

If $\|w_\perp^{(t)}\| \geq \|w_{\text{sig}}^{(t)}\|$, then we must consider the growth of $\|w_\perp\|_\infty$. Fix any $i \geq 3$. $M_t := \chi^{C_{\text{BE}}10000}\theta(1 + 21C_{\text{BE}}\eta)^{t-T_{1a}}$ is the bound guaranteed on on $\|w_\perp\|_\infty$ in Definition D.10, item **W5**. We will show that $w_i^2$ cannot grow too quickly. Recall that $\xi_{\backslash i}$ denotes $\xi - e_i\xi_i$. Then symmetrizing over the pair $(z + \xi_{\backslash i} + e_i\xi_i, z + \xi_{\backslash i} - e_i\xi_i)$, we have

$$-w_i^T\nabla_{w_i}L_0 = a_w\mathbb{E}_xy(x)\sigma'(w^Tx)\xi_i \tag{D.9}$$

$$= a_w\frac{1}{2}\mathbb{E}_xy(z)\left(\sigma'(w^Tz + w^T\xi_{\backslash i} + w_i\xi_i) - \sigma'(w^Tz + w^T\xi_{\backslash i} - w_i\xi_i)\right)w_i\xi_i$$

$$= a_w\frac{1}{2}\mathbb{E}_xy(z)\mathbf{1}(|w^Tz + w^T\xi_{\backslash i}| \leq |w_i|)|w_i|$$

Now explicitly evaluating the expectation over $z$, we have

$$a_w\mathbb{E}_xy(z)\mathbf{1}(|w^Tz + w^T\xi_{\backslash i}| \leq |w_i|)$$

$$= |a_w|\frac{1}{4}\mathbb{E}_{\xi_{\backslash i}}\left[\mathbf{1}(|\sqrt{2}\|w_{\text{sig}}\| + w^T\xi_{\backslash i}| \leq |w_i|) + \mathbf{1}(|-\sqrt{2}\|w_{\text{sig}}\| + w^T\xi_{\backslash i}| \leq |w_i|)\right]$$

$$- |a_w|\frac{1}{4}\mathbb{E}_{\xi_{\backslash i}}\left[\mathbf{1}(|\sqrt{2}\|w_{\text{opp}}\| + w^T\xi_{\backslash i}| \leq |w_i|) + \mathbf{1}(|-\sqrt{2}\|w_{\text{opp}}\| + w^T\xi_{\backslash i}| \leq |w_i|)\right]$$

Since the neuron is weakly controlled and thus $\|w_{\text{sig}}\| \geq \|w_{\text{opp}}\|$, this equals

$$-|a_w|\frac{1}{4}\mathbb{P}_{\xi_{\backslash i}}\left[|w^T\xi_{\backslash i}| \in |w_i| - \sqrt{2}\|w_{\text{sig}}\|, |w_i| - \sqrt{2}\|w_{\text{opp}}\|\right] + |a_w|\frac{1}{4}\mathbb{P}_{\xi_{\backslash i}}\left[|w^T\xi_{\backslash i}| \in |w_i| + \sqrt{2}\|w_{\text{opp}}\|, |w_i| + \sqrt{2}\|w_{\text{sig}}\|\right],$$

By Berry-Esseen (Theorem C.5), we have

$$-\mathbb{P}_{\xi_{\backslash i}}\left[|w^T\xi_{\backslash i}| \in |w_i| - \sqrt{2}\|w_{\text{sig}}\|, |w_i| - \sqrt{2}\|w_{\text{opp}}\|\right] + \mathbb{P}_{\xi_{\backslash i}}\left[|w^T\xi_{\backslash i}| \in |w_i| + \sqrt{2}\|w_{\text{opp}}\|, |w_i| + \sqrt{2}\|w_{\text{sig}}\|\right]$$

$$\leq 4C_{\text{BE}}\frac{\|w_\perp\|_3^3}{\|w_\perp - e_iw_i\|_2^3}$$

$$\leq 4C_{\text{BE}}\frac{\|w_\perp\|_\infty}{\|w_\perp - e_iw_i\|_2}$$

$$\leq 5C_{\text{BE}}\frac{M_t}{\|w_\perp\|_2}.$$

Here the the first inequality following because the Guassian analog of the first probability will be greater than the Gaussian analog of the second probability, since the intervals in question are of the same length, but the first one is closer to $0$. The last inequality follows from **W5**, since the neuron is weakly controlled.

Now returning to Equation D.9, we have

$$-w_i^T\nabla_{w_i}L_0 \leq 5C_{\text{BE}}|a_w|\frac{M_t}{\|w_\perp\|}|w_i| \leq 5C_{\text{BE}}|a_w|\frac{M_t^2}{\|w_\perp\|}.$$

Since we are in the case that $\|w_\perp\| \geq \|w_{\text{sig}}\|$ (which is also at least $\|w_{\text{opp}}\|$), we have $\|w_\perp\| \geq \|w\|/\sqrt{3}$, and thus since $|a_w| \leq \|w\|$ (recall Lemma C.13 **S3**), we have

$$-w_i^T\nabla_{w_i}L_0 \leq 10C_{\text{BE}}M_t^2.$$

We can show via the same calculation performed in this lemma for $w_{\text{sig}}$, that with probability $1 - d^{-\omega(1)}$, the approximation error due to using the population gradient $\nabla L_0$ instead of $\nabla\hat{L}_\rho$ and due to the second order term in $\eta$ are sufficiently small. We omit the details; as before, this uses Lemmas C.12 and C.13. Thus with probability $1 - d^{-\omega(1)}$,

$$(w_i^{(t+1)})^2 \leq M_t^2(1 + \eta21C_{\text{BE}}) \leq M_{t+1}^2,$$

as desired.

$\square$

## E   PHASE 2

Throughout Phase 2, we will show that we can maintain the following invariant. Recall that we have defined the following notation (summarized in Table 1):

$$\gamma_\mu := f_\rho(\mu)y(\mu)$$

$$\gamma_{\min} := \min_{\mu \in \{\pm\mu_1, \pm\mu_2\}} \gamma_\mu \qquad g_{\min} := \min_{\mu \in \{\pm\mu_1, \pm\mu_2\}} |\ell_\rho'(\mu)| = \frac{\exp(-\gamma_{\max})}{1 + \exp(-\gamma_{\max})}$$

$$\gamma_{\max} := \max_{\mu \in \{\pm\mu_1, \pm\mu_2\}} \gamma_\mu \qquad g_{\max} := \max_{\mu \in \{\pm\mu_1, \pm\mu_2\}} |\ell_\rho'(\mu)| = \frac{\exp(-\gamma_{\min})}{1 + \exp(-\gamma_{\min})}$$

**Definition 4.2** (Signal-Heavy Inductive Hypothesis). *For parameters $\zeta = o(1)$ and $H > 1$ with $\zeta \leq \exp(-10H)$, we say a network is $(\zeta, H)$-signal-heavy if there exists some set of* heavy *neurons $S$ on which $\exp(6H)\|w_\perp\| + \|w_{opp}\| \leq \zeta\|w_{sig}\|$, and*

$$\mathbb{E}_\rho \mathbf{1}(w \notin S)\|w\|^2 \leq \zeta\tilde{\gamma}_{min}.$$

*Here we have defined $\tilde{\gamma}_\mu := \mathbb{E}[\mathbf{1}(w \in S, w_{sig}^T\mu > 0)a_w\sigma(w^T\mu)]$ and $\tilde{\gamma}_{min} := \min_{\mu \in \{\pm\mu_1, \pm\mu_2\}} \tilde{\gamma}_\mu$. Further,*

$$\mathbb{E}_\rho[\|w\|^2] \leq \mathbb{E}_\rho[|a_w|^2] + \zeta H \leq 2H,$$

*and for all neurons, we have $|a_w| \leq \|w\|$.*

We will additionally use/recall the following definitions.

**Definition E.1.** *For a $(\zeta, H)$-signal heavy network $\rho$ with heavy set $S$, we define the* heavy-margin:

$$\tilde{\gamma}_\mu := \mathbb{E}[\mathbf{1}(w \in S_\mu)a_w\sigma(w^T\mu)],$$

*and $S_\mu := S \cap \{w : w_{sig}^T\mu > 0\}$. Let $\tilde{\gamma}_{min} := \min_{\mu \in \{\pm\mu_1, \pm\mu_2\}} \tilde{\gamma}_\mu$, and $\tilde{\gamma}_{max} := \max_{\mu \in \{\pm\mu_1, \pm\mu_2\}} \tilde{\gamma}_\mu$.*

Throughout this section, we define the rate parameter

$$\tau := \frac{\sqrt{2}}{4}.$$

We additionally define the following quantities.

**Definition E.2** (Parameters for Phase 2). *Define $\zeta_{T_1} := \log^{-c/3}(d)$, where $c$ is the constant in Lemma D.1, and $H := -\log(\zeta_{T_1})/20$.*

Our main inductive lemma for this phase is as follows:

**Lemma E.3** (Phase 2 Inductive Lemma; Formal). *Suppose $t \leq T_1 + \frac{\zeta_{T_1}^{-1/160}}{\eta}$. If a network $\rho_t$ is $(\zeta, H)$-signal heavy with heavy set $S$ and $\zeta \leq \zeta_{T_1}(1 + 10\eta\zeta H)^{t-T_1}$, then after one minibatch gradient step with step size $\eta \leq \zeta^3$, with probability $1 - d^{-\omega(1)}$,*

1. *$\rho_{t+1}$ is $(\zeta(1 + 10\eta\zeta H), H)$-signal heavy.*
2. *$\tilde{\gamma}_{min}^{(t+1)} \geq (1 + 2\eta\tau(1 - o(1))g_{max})\tilde{\gamma}_{min}^{(t)}$*
3. *$\tilde{\gamma}_{max}^{(t+1)} \leq (1 + 2\eta\tau(1 + o(1))g_{min})\tilde{\gamma}_{max}^{(t)}.$*

*Here $\zeta_{T_1}$ and $H$ are defined in Definition E.2, and $\tau = \frac{\sqrt{2}}{4}$.*

**Lemma E.4** (Base Case from Phase 1). *Assume the conclusion of Lemma D.1 holds for the network $\rho_{T_1}$ after $T_1$ steps. Then $\rho_{T_1}$ is $(\zeta_{T_1}, H)$ signal-heavy for the parameters $\zeta_{T_1}$ and $H$ define in Definition E.2.*

*Further, we have $\tilde{\gamma}_{min}^{(T_1)} \geq \log^{-\Theta(1)}(d)$, and $\tilde{\gamma}_{max}^{(T_1)} \leq 1$.*

*Proof.* Let $\rho := \rho_{T_1}$, and we will likewise drop the superscript $T_1$ on all other variables. First observe that $\tilde{\gamma}_{\max} \leq 1$ since for any $\mu \in \{\pm\mu_1, \pm\mu_2\}$, we have $\tilde{\gamma}_\mu \leq \mathbb{E}_\rho[\|a_ww_{1:2}\|] \leq 1$.
For $\mu \in \{\pm\mu_1, \pm\mu_2\}$, define

$$S_\mu := \{w : \zeta_{T_1}w^T\mu \geq \exp(6H)\|w_\perp\| + \|w_{opp}\|, a_wy(\mu) > 0\},$$

and let $S := S_{\mu_1} \cup S_{-\mu_1} \cup S_{\mu_2} \cup S_{-\mu_2}$.

First observe that for any $\mu \in \{\pm\mu_1, \pm\mu_2\}$, by the third item of Lemma D.1, we have

$$\tilde{\gamma}_\mu = \mathbb{E}_\rho[\mathbf{1}(w \in S_\mu)\|a_w w_{\text{sig}}\|]$$

$$\geq \frac{1}{20}\log^c(d)\theta^2$$

$$\geq \log^{-\Theta(1)}(d).$$

fFinally, we have

$$\mathbb{E}_\rho[\mathbf{1}(w \notin S)\|w\|^2] = \mathbb{E}_\rho\left[\mathbf{1}(w \notin S)\left(\|w_{\text{ns}}\|^2 + \|w_{\text{sig}}\|^2\right)\right]$$

$$\leq \mathbb{E}_\rho\left[\mathbf{1}(w \notin S)\|w_{\text{ns}}\|^2\left(1 + (\exp(6H)\zeta_{T_1}^{-1})^2\right)\right]$$

$$\leq \mathbb{E}_\rho\left[\|w_{\text{ns}}\|^2\right]\left(1 + (\zeta_{T_1}^{-2/3})^2\right)$$

$$\leq 4\theta^2(\zeta_{T_1}^{-4/3} + 1)$$

$$\leq 8\zeta_{T_1}^{-4/3}(20\log^{-c}\tilde{\gamma}_{\min})$$

$$\leq 160\zeta_{T_1}^{-4/3}\zeta_{T_1}^3\tilde{\gamma}_{\min}$$

$$\leq \zeta_{T_1}\tilde{\gamma}_{\min}.$$

Finally, observe that the last clause of Lemma D.1 yields the final condition of Definition 4.2 abound $\mathbb{E}[\|w\|^2]$. This yields the lemma. $\qquad\square$

We recall that in Phase 2, our main analysis tool is to compare to the "clean" gradients $\nabla^{\text{cl}}$, which are defined in Equation 4.7 as

$$\nabla_w^{\text{cl}}L_\rho := a_w\mathbb{E}_x\ell_\rho'(z)\sigma'(w^Tx)x \quad \text{and} \quad \nabla_{a_w}^{\text{cl}}L_\rho := \mathbb{E}_x\ell_\rho'(z)\sigma(w^Tx).$$

Similarly to Phase 1, the main building blocks of Phase 2 are computations of the gradients $\nabla^{\text{cl}}L_\rho$, and bounds on the distance $\|\nabla_w^{\text{cl}}L_\rho - \nabla L_\rho\|$. We state these main lemmas here.

Throughout, we define

$$H_\rho := \mathbb{E}_\rho[\|a_w w\|],$$

such that we have

$$\exp(-H_\rho) \leq g_{\min} \leq g_{\max} \leq 2. \tag{E.1}$$

### E.1 BOUNDING DISTANCE TO CLEAN GRADIENTS.

**Lemma E.5.** *For any neuron $w$, we have*

$$\|\nabla_w^{cl}L_\rho - \nabla_w L_\rho\|_2 \leq 4|a_w|\zeta H_\rho;$$

$$\|\nabla_{a_w}^{cl}L_\rho - \nabla_{a_w}L_\rho\|_2 \leq 4\|w\|\zeta H_\rho.$$

*Proof.* Letting $\Delta_x := (\ell_\rho'(x) - \ell_\rho'(z))\sigma'(w^Tx)$, we have

$$\|\nabla_w^{\text{cl}}L_\rho - \nabla_w L_\rho\|_2 = |a_w|\|\mathbb{E}_x\Delta_x x\|$$

$$= |a_w|\sup_{v:\|v\|=1}\mathbb{E}_x\Delta_x\langle v, x\rangle$$

$$\leq |a_w|\sup_{v:\|v\|=1}\sqrt{\mathbb{E}_x\Delta_x^2}\sqrt{\mathbb{E}_x\langle v, x\rangle^2}$$

$$= |a_w|\sqrt{\mathbb{E}_x\Delta_x^2}.$$

Now

$$\begin{aligned}
\mathbb{E}_x[\Delta_x^2] &\leq \mathbb{E}_x[(\ell_\rho'(x) - \ell_\rho'(z))^2] \\
&= \mathbb{E}_z(\ell_\rho'(z))^2 \mathbb{E}_\xi \left(\frac{\ell_\rho'(x)}{\ell_\rho'(z)} - 1\right)^2 \\
&\leq \mathbb{E}_z(\ell_\rho'(z))^2 \mathbb{E}_\xi \left(\exp(|f_\rho(z) - f_\rho(x)|) - 1\right)^2 \\
&\leq g_{\max}^2 \mathbb{E}_\xi \left(\exp(\mathbb{E}_\rho|a_w w^T \xi|) - 1\right)^2 \\
&\leq g_{\max}^2 \mathbb{E}_\xi \left(\exp(\mathbb{E}_\rho[\|a_w w_\perp\|]|v^T \xi|) - 1\right)^2,
\end{aligned}$$

where $v$ is any unit vector. Here the last line holds because the expression will be maximized when all neurons are in the same direction (up to sign).

Then since $v^T \xi$ is subgaussian, defining $N := \mathbb{E}_\rho[\|a_w w_\perp\|] \leq \zeta H_\rho$, we have

$$\mathbb{E}_\xi \left(\exp(|v^T \xi|) - 1\right)^2 \leq (\exp(2N + 2N^2) - 1)^2 \leq (\exp(\zeta H_\rho + \zeta^2 H_\rho^2) - 1)^2 \leq 4\zeta^2 H_\rho^2.$$

(Explicitly, we can verify this by upper bounding the moments of $|v^T \xi|$ by moments of a Gaussian, and then using the moment generating function of a half-Gaussian distribution.)

Thus plugging this back in and recalling that $g_{\max} \leq 2$ always, we achieve

$$\|\nabla_w^{cl} L_\rho - \nabla_w L_\rho\|_2 \leq 4|a_w|\zeta H_\rho < 1,$$

as desired. Similarly, we have

$$\|\nabla_{a_w}^{cl} L_\rho - \nabla_{a_w} L_\rho\|_2 = \mathbb{E}_x[\|\Delta_x w^T x\|] \leq \|w\|\sqrt{\mathbb{E}_x \Delta_x^2},$$

and thus

$$\|\nabla_{a_w}^{cl} L_\rho - \nabla_{a_w} L_\rho\|_2 \leq 4\|w\|\zeta H_\rho.$$

$\square$

## E.2 Clean Gradients

### E.2.1 Neurons in $S$

**Claim E.6** (Clean Gradients in Signal Direction). *If $w \in S$ and $\mu^T w_{sig} > 0$, then*

$$\mu^T \nabla_w^{cl} L_\rho = -|a_w|\tau g_\mu(1 \pm o(1)),$$

*and*

$$-y(\mu)\nabla_{a_w}^{cl} L_\rho = (1 \pm o(1))\|w_{sig}\|\tau g_\mu.$$

*Proof.* First, we compute

$$\begin{aligned}
\mu^T \nabla_w^{cl} L_\rho &= a_w \mathbb{E}_z \ell_\rho'(z)\sigma'(w^T z)z^T \mu - \mathbb{E}_x \ell_\rho'(z)(\sigma'(w^T x) - \sigma'(w^T z))z^t \mu \\
&\in -\tau a_w y(\mu)g_\mu \pm \sqrt{2}\mathbb{E}_x|\ell_\rho'(z)|\mathbf{1}(|w^T \xi| \geq |w^T z|) \\
&\in -\tau|a_w|g_\mu \pm \sqrt{2}|a_w|g_{\max}\frac{\|w_\perp\|^2}{\|w_{\text{sig}}^2\|} && \text{(Chebychev's inequality)} \\
&\in -\tau|a_w|g_\mu \pm \sqrt{2}|a_w|g_{\max}\zeta^2 && (w \in S) \\
&= -|a_w|\tau g_\mu(1 \pm o(1)).
\end{aligned}$$

Second, we compute

$$\begin{aligned}
\nabla_{a_w}^{cl} L_\rho &= \mathbb{E}_z \ell_\rho'(z)\sigma(w^T x) \\
&= -\tau y(\mu)\left(g_\mu\|w_{\text{sig}}\| \pm \Theta(g_{\max}\|w_{\text{ns}}\|)\right) \\
&= -\tau y(\mu)\left(g_\mu\|w_{\text{sig}}\| \pm \Theta(g_{\max}\zeta\|w_{\text{sig}}\|)\right) \\
&= -(1 \pm o(1))\frac{\tau y(\mu)}{g_\mu}\|w_{\text{sig}}\|.
\end{aligned}$$

$\square$

In the following lemma, we lower bound the size of the gradient in the $w_\perp$ and $w_{\text{opp}}$ directions.

**Lemma E.7.** *For a neuron $(w, a_w)$, let $X := \mathbb{P}_\xi[|w_\perp^T \xi| \geq \sqrt{2}\|w_{opp}\|]$. Then*

$$\frac{1}{|a_w|} w_\perp^T \nabla_w^{cl} L_\rho \geq \frac{1}{8} g_{min} X \|w_\perp\| - \zeta \|w_\perp\|.$$

*and*

$$\frac{1}{|a_w|} w_{opp}^T \nabla_w^{cl} L_\rho \geq \frac{1}{8} g_{min} \sqrt{2}\|w_{opp}\| - \frac{g_{max}\sqrt{2}}{4} X \|w_{opp}\|.$$

*Proof of Lemma E.7.* We compute

$$\text{sign}(a_w) \mathbb{E}_x \ell_\rho'(z) \sigma'(w^T x) x^T w_\perp = \mathbb{E}_\xi \mathbb{E}_z \ell_\rho'(z) \sigma'(w^T \xi + w^T z) \xi^T w_\perp$$

$$= \frac{1}{2} \mathbb{E}_\xi \mathbb{E}_z \ell_\rho'(z) \left( \sigma'(w^T \xi + w^T z) - \sigma'(-w^T \xi + w^T z) \right) \xi^T w_\perp$$

$$= \frac{1}{2} \mathbb{E}_z \ell_\rho'(z) \mathbb{E}_\xi \mathbf{1}(|w^T \xi| \geq |w^T z|) |\xi^T w_\perp|$$

$$\geq -\frac{1}{4} g_{max} \mathbb{E}_\xi \mathbf{1}(|w^T \xi| \geq \sqrt{2}\|w_{\text{sig}}\|) |\xi^T w_\perp| + \frac{1}{4} g_{min} \mathbb{E}_\xi \mathbf{1}(|w^T \xi| \geq \sqrt{2}\|w_{\text{opp}}\|) |\xi^T w_\perp|$$

$$\geq -\frac{1}{4} g_{max} \sqrt{\mathbb{E}_\xi [\mathbf{1}(|w^T \xi| \geq \sqrt{2}\|w_{\text{sig}}\|)]} \sqrt{\mathbb{E}_\xi [\xi^T w_\perp|^2]} + \frac{1}{4} g_{min} \mathbb{E}_\xi \mathbf{1}(|w^T \xi| \geq \sqrt{2}\|w_{\text{opp}}\|) \mathbb{E}_\xi |\xi^T w_\perp|$$

$$\geq -g_{max} \sqrt{\mathbb{P}[|w^T \xi| \geq \sqrt{2}\|w_{\text{sig}}\|]} \|w_\perp\| + \frac{1}{4} g_{min} X \|w_\perp\|$$

$$\geq -\zeta \|w_\perp\| + \frac{1}{4} g_{min} X \|w_\perp\|.$$

where here in the last line, we used the inductive hypothesis that $\|w_\perp\| \leq \|w_{\text{ns}}\| \leq \zeta \|w_{\text{sig}}\|$. We also compute the gradient in the $w_{\text{opp}}$ direction:

$$\text{sign}(a_w) \mathbb{E}_x \ell_\rho'(z) \sigma'(w^T x) x^T w_{\text{opp}} \geq -\frac{1}{4} \mathbb{E}_\xi g_{max} \mathbf{1}(|w^T \xi| \geq \sqrt{2}\|w_{\text{opp}}\|) \sqrt{2}\|w_{\text{opp}}\| + \frac{1}{8} g_{min} \sqrt{2}\|w_{\text{opp}}\|$$

$$= -\frac{g_{max}\sqrt{2}}{4} X \|w_{\text{opp}}\| + \frac{1}{8} g_{min} \sqrt{2}\|w_{\text{opp}}\|.$$

$\square$

We derive the following corollary of Lemma E.7, which will be used to show that a weighted average of $\|w_\perp\|^2$ and $\|w_{\text{opp}}\|^2$ decreases under the clean gradients.

**Corollary E.8** (Clean Population Gradients for $w_{\text{ns}}$)**.** *If $w \in S$, then*

$$-\nabla_w^{cl} L_\rho^T w_{opp} - \exp(6H) \nabla_w^{cl} L_\rho^T w_\perp \leq -\zeta^{2/3} \left( \|w_{opp}\| + \exp(6H) \|w_\perp\| \right) |a_w|$$

*Proof.* The gist of the proof of this claim is to show that if $w_{\text{opp}}$ is large relative to $w_\perp$, then the gradient will be large in the $w_{\text{opp}}$ direction, thereby decreasing $w_{\text{opp}}$. Conversely, if $w_\perp$ is large relative to $w_{\text{opp}}$, then the gradient will be large in the $w_\perp$ direction. Because however we are working on the Boolean hypercube, where $\xi$ is not rotationally invariant, the exact condition of "$w_{\text{opp}}$ being large relative to $w_\perp$" is slightly nuanced.
Let

$$X := \text{Pr}[|w_\perp^T \xi| \geq \sqrt{2}\|w_{\text{opp}}\|].$$

If $X$ is large, then we will show that $w_\perp^T \nabla_w^{\text{cl}} L_\rho$ is sufficiently large to yield the desired result. If $X$ is small, we will show that $w_{\text{opp}}^T \nabla_w^{\text{cl}} L_\rho$ is sufficiently large to yield the desired result.
Now

$$\frac{1}{|a_w|} \left( w_{\text{opp}}^T \nabla_w^{\text{cl}} L_\rho + \exp(6H) w_\perp^T \nabla_w^{\text{cl}} \right)$$

$$\geq \left( \frac{1}{8} g_{\min} \sqrt{2}\|w_{\text{opp}}\| - \frac{g_{\max}\sqrt{2}}{4} X \|w_{\text{opp}}\| + \frac{\exp(6H)}{8} g_{\min} X \|w_\perp\| - \zeta \exp(6H) \|w_\perp\| \right)$$

$$\geq \frac{1}{8} g_{\min} \sqrt{2}\|w_{\text{opp}}\| - \frac{\exp(H) g_{\min} \sqrt{2}}{4} X \|w_{\text{opp}}\| + \frac{\exp(6H)}{8} g_{\min} X \|w_\perp\| - \zeta g_{\min} \exp(7H) \|w_\perp\|$$

First we will consider the case that $\|w_\perp\|$ is large relative to $\|w_{opp}\|$. We will need the following claim.

**Claim E.9.** *If $\|w_\perp\| \geq 10\|w_{opp}\|$, then $X \geq \frac{1}{2}$.*

*Proof.* If $\|w_\perp\|_\infty \geq \sqrt{2}\|w_{opp}\|$, then with $i := \arg\max |(w_\perp)_i|$, with probability $1/2$ condtional on $\{\xi_j\}_{j \neq i}$, $\xi_i$ is such that $|w_\perp^T \xi| \geq |(w_\perp)_i|$, and thus

$$X = \mathbb{P}[|w_\perp^T \xi| \geq \sqrt{2}\|w_{opp}\|] \geq \frac{1}{2}.$$

Otherwise, by Berry-Essen (Theorem C.5), we have

$$\mathbb{P}[|w_\perp^T \xi| \geq \sqrt{2}\|w_{opp}\|] \geq \mathbb{P}_{G \sim N(0,1)}\left[|G| \geq \frac{\sqrt{2}\|w_{opp}\|}{\|w_\perp\|}\right] - \frac{\|w_\perp\|_\infty}{\|w_\perp\|}$$

$$\geq \mathbb{P}_{G \sim N(0,1)}\left[|G| \geq \frac{\sqrt{2}\|w_{opp}\|}{\|w_\perp\|}\right] - \frac{\sqrt{2}\|w_{opp}\|}{\|w_\perp\|}$$

$$\geq \mathbb{P}_{G \sim N(0,1)}\left[|G| \geq \frac{\sqrt{2}}{10}\right] - \frac{\sqrt{2}}{10}$$

$$\geq \frac{1}{2}.$$

$\square$

Thus if $\|w_\perp\| \geq 10\|w_{opp}\|$, by bounding $\frac{1}{2} \leq X \leq 1$, we have

$$\frac{1}{|a_w|}\left(w_{opp}^T \nabla_w^{cl} L_\rho + \exp(6H) w_\perp^T \nabla_w^{cl} L_\rho\right)$$

$$\geq \left(\frac{1}{8} g_{min}\sqrt{2}\|w_{opp}\| - \frac{\exp(H)g_{min}\sqrt{2}}{4}\|w_{opp}\| + \frac{\exp(6H)}{16}g_{min}\|w_\perp\| - \zeta g_{min}\exp(7H)\|w_\perp\|\right)$$

$$\geq \frac{\exp(6H)}{16}g_{min}\|w_\perp\| - \frac{\exp(H)g_{min}\sqrt{2}}{40}\|w_\perp\| - \zeta g_{min}\exp(7H)\|w_\perp\|$$
$$\hspace{8cm}(\|w_\perp\| \geq 10\|w_{opp}\|)$$

$$\geq \frac{\exp(6H)}{20}g_{min}\|w_\perp\| \hspace{4cm}(\zeta \leq \exp(-10H))$$

$$\geq \frac{g_{min}}{30}\left(\|w_{opp}\| + \exp(6H)\|w_\perp\|\right). \hspace{2.5cm}(\|w_\perp\| \geq 10\|w_{opp}\|)$$

Now if $\|w_\perp\| \leq 10\|w_{opp}\|$, we have

$$\frac{1}{|a_w|}\left(w_{opp}^T \nabla_w^{cl} L_\rho + \exp(6H) w_\perp^T \nabla_w^{cl} L_\rho\right)$$

$$\geq \left(\frac{1}{8} g_{min}\sqrt{2}\|w_{opp}\| - \frac{\exp(H)g_{min}\sqrt{2}}{4}X\|w_{opp}\| + \frac{\exp(6H)}{8}g_{min}X\|w_\perp\| - \zeta\exp(6H)\|w_\perp\|\right)$$

$$\geq \left(\frac{1}{10} g_{min}\sqrt{2}\|w_{opp}\| - \frac{\exp(H)g_{min}\sqrt{2}}{4}X\|w_{opp}\| + \frac{\exp(6H)}{8}g_{min}X\|w_\perp\|\right)$$
$$\hspace{6cm}(\zeta \leq \exp(-10H), \|w_\perp\| \leq 10\|w_{opp}\|)$$

$$\geq \frac{1}{10}g_{min}\sqrt{2}\|w_{opp}\| \hspace{4cm}(2\sqrt{2}\exp(-5H)\|w_{opp}\| \leq \|w_\perp\|)$$

$$\geq g_{min}\frac{\sqrt{2}}{200}\exp(-6H)\left(\|w_{opp}\| + \exp(6H)\|w_\perp\|\right) \hspace{1.5cm}(\|w_\perp\| \leq 10\|w_{opp}\|)$$

$$\geq \zeta^{2/3}g_{min}\left(\|w_{opp}\| + \exp(6H)\|w_\perp\|\right). \hspace{2.5cm}(\zeta \leq \exp(-10H))$$

Here the third inequality follows from the fact that if $\|w_\perp\| \geq 2\sqrt{2}\exp(-5H)\|w_{opp}\|$, then the positive term with an $X$ exceeds the negative term with an $X$. Alternatively, if $\|w_\perp\| \leq$

$2\sqrt{2}\exp(-5H)\|w_{\text{opp}}\|$, then by Chebychev's inequality that $X \leq \frac{\|w_\perp\|^2}{2\|w_{\text{opp}}\|^2} \leq 4\exp(-10H)$, so we can bound the negative term with an $X$.

These two cases prove the lemma.

$\square$

### E.2.2 NEURONS NOT IN $S$

Finally, we need to show that the neurons not in $S$ don't grow too large. To do this, we use the following claim, which states that no neuron can grow more at the rate $\tau g_{\max}$, which is the rate of growth of the neuron in the direction $\arg\min_\mu \mu^T$.

**Lemma E.10** (Clean Gradient Bound for all Neurons). *For any neuron,*

$$|\nabla_{a_w}^{cl} L_\rho| = |w^T \nabla_w^{cl} L_\rho| \leq \tau g_{max}\|w\|.$$

*Proof.* First recall that

$$|w^T \nabla_w^{\text{cl}} L_\rho| = |\mathbb{E}_z \ell'_\rho(z) \mathbb{E}_\xi \sigma(w^T z + w^T \xi)| = |\nabla_{a_w}^{\text{cl}} L_\rho|.$$

We compute

$$|\mathbb{E}_z \ell'_\rho(z) \mathbb{E}_\xi \sigma(w^T z + w^T \xi)|$$

$$\leq \sup_\mu \frac{1}{4}\left(g_{\max}\mathbb{E}_\xi\sigma(w^T\mu + w^T\xi) + g_{\min}\mathbb{E}_\xi\sigma(-w^T\mu + w^T\xi)\right)$$

$$\leq \sup_\mu \frac{1}{4}g_{\max}\left(\mathbb{E}_\xi\sigma(w^T\mu + w^T\xi) + \sigma(-w^T\mu + w^T\xi)\right)$$

$$= \sup_\mu \frac{1}{8}g_{\max}\left(\mathbb{E}_\xi\sigma(w^T\mu + w^T\xi) + \sigma(-w^T\mu + w^T\xi) + \sigma(w^T\mu - w^T\xi) + \sigma(-w^T\mu - w^T\xi)\right)$$

$$\leq \frac{\sqrt{2}}{4}\|w\|g_{\max} = \tau g_{\max}\|w\|.$$

Indeed, the last inequality follows from the fact that the expression is maximized when $w$ is in the direction of $\mu$. $\square$

We additionally use the following lemma.

**Lemma E.11.** *For any neuron $w$, with high probability*

$$\|w^{(t+1)}\|^2 \leq \|w^{(t)}\|^2\left(1 + 2\eta(1 + 2\zeta H)\tau g_{max}\right).$$

*Proof.* With high probability,

$$\|w^{(t+1)}\|^2 = \|w^{(t)}\|^2 - 2\eta w^T \nabla_w \hat{L} + \eta^2\|\nabla_w \hat{L}\|^2$$

$$\leq \|w^{(t)}\|^2 - 2\eta w^T \nabla_w^{\text{cl}} L_\rho + 2\eta\|w\|\|\nabla_w^{\text{cl}} L_\rho - \nabla_w \hat{L}\| + \eta^2\|\nabla_w \hat{L}\|^2$$

$$\leq \|w^{(t)}\|^2 - 2\eta w^T \nabla_w^{\text{cl}} L_\rho + 2\eta\|w\|^2\zeta H g_{\max} + \eta^2\|\nabla_w \hat{L}\|^2$$

$$\leq \|w^{(t)}\|^2 - 2\eta w^T \nabla_w^{\text{cl}} L_\rho + 4\eta H\zeta g_{\max}\|w\|^2,$$

where in the second inequality we used Claim E.5. Plugging in Lemma E.10 yields the claim. $\square$

### E.3 PROOF OF INDUCTIVE LEMMA

We break the proof of Lemma E.3 up into three main lemmas. The first lemma shows the growth of $\tilde{\gamma}_{\min}$. The second ensures that $\tilde{\gamma}_{\max}$ doesn't grow too fast. The third lemma ensures that the network stays signal-heavy.

**Lemma E.12** ($\tilde{\gamma}_{\min}$). *Suppose $\rho_t$ is $(\zeta, H)$-signal heavy for some signal-heavy set $S$. Then if $\eta \leq \zeta^3$, with probability $1 - d^{-\omega(1)}$,*

$$\tilde{\gamma}_{min}^{(t+1)} \geq \left(1 + 2\eta\tau(1 - o(1))g_{max}\right)\tilde{\gamma}_{min}^{(t)}.$$

*Further, for any neuron for which $\|w_{sig}^{(t)}\| \leq \exp(6H)\|w_\perp^{(t)}\| + \|w_{opp}^{(t)}\|$, we have $\|w_{sig}^{(t+1)}\| \leq \exp(6H)\|w_\perp^{(t+1)}\| + \|w_{opp}^{(t+1)}\|$.*

**Lemma E.13** ($\tilde{\gamma}_{\max}$). *Suppose $\rho_t$ is $(\zeta, H)$-signal heavy for some signal-heavy set $S$. Then if $\eta \leq \zeta^3$, with probability $1 - d^{-\omega(1)}$,*

$$\tilde{\gamma}_{max}^{(t+1)} \leq (1 + 2\eta\tau(1 + o(1))g_{min})\,\tilde{\gamma}_{max}^{(t)}.$$

To prove these two lemmas, we will also need the following lemma which states the the network doesn't change too much at each iteration.

**Lemma E.14.** *If $\rho_t$ satisfies Definition 4.2 and $\eta \leq \zeta^2$, then with probability $1 - d^{-\omega(1)}$, we have*

$$|\tilde{\gamma}_{\mu}^{(t+1)} - \tilde{\gamma}_{\mu}^{(t)}| \leq \sqrt{\eta}.$$

*Further for any $\mu$,*

$$|\gamma_{\mu} - \tilde{\gamma}_{\mu}| \leq 2\zeta H. \tag{E.2}$$

*Proof.* Using Lemma C.13, and C.12 we have with probability $1 - d^{-\omega(1)}$

$$
\begin{aligned}
|\tilde{\gamma}_{\mu}^{(t+1)} - \tilde{\gamma}_{\mu}^{(t)}| &\leq \eta\mathbb{E}_{\rho}[\|\nabla_{a_w}\hat{L}_{\rho}\|\|w\| + \|\nabla_w\hat{L}_{\rho}\|\|a_w\|] \\
&\leq 2\eta\mathbb{E}_{\rho}[\|w\|^2 + \|a_w\|^2] &&\text{(Lemma C.13 and C.12 )} \\
&\leq 8\eta H &&\text{(Definition 4.2)} \\
&\leq \sqrt{\eta}. &&(H \leq \log(\zeta^{-1})/10 \leq \log(\eta^{-1/2})/10 \leq \eta^{-1/2}/8)
\end{aligned}
$$

For the second statement, we have

$$
\begin{aligned}
|\gamma_{\mu} - \tilde{\gamma}_{\mu}| &\leq \mathbb{E}_{\rho}[\mathbf{1}(w \notin S)\|a_w w\|] + \mathbb{E}_{\rho}[\mathbf{1}(w \in S)\|a_w w_{\text{opp}}\|] \\
&\leq \mathbb{E}_{\rho}[\mathbf{1}(w \notin S)\|w\|^2] + \zeta\mathbb{E}_{\rho}[\mathbf{1}(w \in S)\|a_w w\|] \\
&\leq 2\zeta H.
\end{aligned}
$$

$\square$

*Proof of Lemma E.12.* Our approach here will be to show that for any $\mu \in \{\pm\mu_1, \pm\mu_2\}$ from which $\tilde{\gamma}_{\mu} \leq \tilde{\gamma}_{\min} + 2\sqrt{\eta}$, we have

$$\tilde{\gamma}_{\mu}^{(t+1)} \geq \left(1 + \eta\frac{\sqrt{2}}{2}(1 + o(1))g_{\max}\right)\tilde{\gamma}_{\min}^{(t)}. \tag{E.3}$$

Then by Lemma E.14 for any $\mu$ for which $\tilde{\gamma}_{\mu}^{(t)} \geq \tilde{\gamma}_{\min}^{(t)} + 2\sqrt{\eta}$, we have

$$\tilde{\gamma}_{\mu}^{(t+1)} \geq \tilde{\gamma}_{\mu}^{(t)} - \sqrt{\eta} \geq \tilde{\gamma}_{\min}^{(t)} + \sqrt{\eta} \geq (1 + \eta)\,\tilde{\gamma}_{\min}^{(t)} \geq \left(1 + \frac{\sqrt{2}}{2}\eta g_{\max}^{(t)}\right)\tilde{\gamma}_{\min}^{(t)}.$$

Let us prove Equation E.3. Fix any $\mu$ with $\tilde{\gamma}_{\mu} \leq \tilde{\gamma}_{\min} + 2\sqrt{\eta}$. We first define a set of neurons on which the growth of signal is large. Let

$$S_{\mu} = \{w : \zeta w^T\mu \geq \|w_{\perp} + w_{\text{opp}}\|, a_w y(\mu) > 0\},$$

that is, $S_{\mu} = S \cap \{w : w_{\text{sig}}^T\mu > 0\}$, where $S$ is the signal-heavy set from Definition 4.2.

**Claim E.15.** *For any $w \in S_{\mu}$,*

$$\|a_w^{(t+1)}w_{sig}^{(t+1)}\| \geq \|a_w^{(t)}w_{sig}^{(t)}\|\left(1 + 2\eta\tau(1 - o(1))g_{\mu}^{(t)}\right)$$

*Proof of Claim E.15.* Observe that (with $y = y(\mu)$ and $X := \tau g_\mu^{(t)}$, we have

$$ya_w^{(t+1)}\|w_{\text{sig}}^{(t+1)}\| = y(a_w^{(t)} - \eta\nabla_{a_w}\hat{L}_\rho)(\|w_{\text{sig}}^{(t)}\| - \eta\mu^T\nabla_w\hat{L}_\rho)$$

$$\geq y(a_w - \eta\nabla_{a_w}^{\text{cl}}L_\rho)(\|w_{\text{sig}}\| - \eta\mu^T\nabla_w^{\text{cl}}L_\rho) - O\left(\eta|a_w|\|\nabla_w^{\text{cl}}L_\rho - \nabla_w\hat{L}_\rho\| + \eta\|w_{\text{sig}}\|\|\nabla_{a_w}^{\text{cl}}L_\rho - \nabla_{a_w}\hat{L}_\rho\|\right) - O(\eta^2\|a_w w\|)$$
$$\text{(Lemma C.13)}$$

$$\geq y(a_w - \eta\nabla_{a_w}^{\text{cl}}L_\rho)(\|w_{\text{sig}}\| - \eta\mu^T\nabla_w^{\text{cl}}L_\rho) - O\left(\eta\zeta H_\rho(a_w^2 + \|w_{\text{sig}}\|^2)\right) - O(\eta^2\|a_w w\|)$$
$$\text{(Lemma E.5)}$$

$$\geq ya_w\|w_{\text{sig}}\| + \eta(1 - o(1))\left(X\|w_{\text{sig}}\|^2 + Xa_w^2\right) - O\left(\eta\zeta H_\rho(a_w^2 + \|w\|^2)\right) - O(\eta^2\|a_w w\|)$$
$$\text{(Lemma E.6)}$$

$$\geq ya_w\|w_{\text{sig}}\| + 2X\eta(1 - o(1))ya_w\|w_{\text{sig}}\| + O(\eta^2\|a_w w\|)$$
$$\text{(AM-GM, and } \zeta H_\rho = o(g_{\min}) \text{ by Eq. E.1, } \zeta \leq \exp(-10H))$$

$$\geq ya_w^{(t)}\|w_{\text{sig}}^{(t)}\|(1 + 2\eta(1 - o(1))X),$$

as desired. $\qquad\square$

Now we have

$$\frac{g_\mu}{g_{\max}} = \frac{\exp(-\gamma_\mu + \gamma_{\min})(1 + \exp(-\gamma_{\min}))}{1 + \exp(-\gamma_\mu)}$$

$$\geq \exp(-\gamma_\mu + \gamma_{\min}) \qquad\qquad (\gamma_\mu \geq \gamma_{\min})$$
$$\geq \exp(-\tilde{\gamma}_\mu + \tilde{\gamma}_{\min} - 4\zeta H) \qquad\qquad \text{(Eq E.2)}$$
$$\geq \exp(-2\sqrt{\eta} - 4\zeta H) \qquad\qquad \text{(Lemma E.14)}$$
$$\geq 1 - o(1).$$

Plugging this in to the previous equation yields

$$ya_w^{(t+1)}\|w_{\text{sig}}^{(t+1)}\| \geq ya_w^{(t)}\|w_{\text{sig}}^{(t)}\|(1 + 2\tau\eta(1 - o(1))g_{\max})$$

Now it remains to check that if a neuron is in $S_\mu$ at step $t$, then that neuron still satisfies $\zeta\|w_{\text{sig}}^{(t+1)}\| \geq \exp(6H)\|w_\perp^{(t+1)}\| + \|w_{\text{opp}}^{(t+1)}\|$ at time $t + 1$. Observe that for every neuron in $S$, we have:

1. $\|w_{\text{sig}}^{(t+1)}\| \geq \|w_{\text{sig}}^{(t)}\|$. This is easy to show (as in the calculation above) by plugging in the lower bound on $-w_{\text{sig}}^T\nabla_w^{\text{cl}}L_\rho$, and the upper bound on $\|\nabla_w^{\text{cl}}L_\rho - \nabla\hat{L}_\rho\|$.

2. Since $\eta$ is small enough (relative to $\zeta$), if $\exp(6H)\|w_\perp^{(t)}\| + \|w_{\text{opp}}^{(t)}\| \leq \zeta\|w_{\text{sig}}^{(t)}\|/2$, then $\exp(6H)\|w_\perp^{(t+1)}\| + \|w_{\text{opp}}^{(t+1)}\| \leq \zeta\|w_{\text{sig}}^{(t)}\| \leq \zeta\|w_{\text{sig}}^{(t+1)}\|$.

3. If $\exp(6H)\|w_\perp^{(t)}\| + \|w_{\text{opp}}^{(t)}\| \geq \zeta\|w_{\text{sig}}^{(t)}\|/2$, then

$$\exp(6H)\|w_\perp^{(t+1)}\|^2 + \|w_{\text{opp}}^{(t+1)}\|^2 - \left(\exp(6H)\|w_\perp^{(t)}\|^2 + \|w_{\text{opp}}^{(t)}\|^2\right)$$

$$\leq -2\eta\exp(6H)(w_\perp^{(t)})^T\nabla_w\hat{L}_\rho - 2\eta(w_{\text{opp}}^{(t)})^T\nabla_w\hat{L}_\rho + \exp(6H)\eta^2\|\nabla_w\hat{L}_\rho\|^2$$

$$\leq -2\eta\left(\exp(6H)w_\perp^T\nabla_w^{\text{cl}}L_\rho + 2\eta w_{\text{opp}}^T\nabla_w^{\text{cl}}L_\rho\right) + 2\eta(\exp(6H)\|w_\perp\| + \|w_{\text{opp}}\|)\|\nabla_w^{\text{cl}}L_\rho - \nabla_w\hat{L}_\rho\|$$
$$+ \exp(6H)\eta^2\|\nabla_w\hat{L}_\rho\|^2$$

Now we can use Corollary E.8 to bound the first term, and Lemma E.5, Lemma C.12, and Lemma C.13 to bound the second and third terms. Thus we have

$$\exp(6H)\|w_\perp^{(t+1)}\|^2 + \|w_{\text{opp}}^{(t+1)}\|^2 - \left(\exp(6H)\|w_\perp^{(t)}\|^2 + \|w_{\text{opp}}^{(t)}\|^2\right)$$

$$\leq -2\eta\zeta^{2/3}\left(\|w_{\text{opp}}\| + \exp(6H)\|w_\perp\|\right)|a_w| + 2\eta(\exp(6H)\|w_\perp\| + \|w_{\text{opp}}\|)\zeta H_\rho|a_w| + O(\exp(6H)\eta^2\|a_w\|^2)$$

$$\leq -\eta\zeta^{2/3}\left(\|w_{\text{opp}}\| + \exp(6H)\|w_\perp\|\right)|a_w| + O(\exp(6H)\eta^2\|a_w\|^2) \quad (\zeta \leq \exp(-10H))$$

$$\leq -\eta\zeta^{2/3}\left(\|w_{\text{opp}}\| + \exp(6H)\|w_\perp\|\right)|a_w| + O(\exp(6H)\eta^2\|w\|\,|a_w|) \qquad (|a_w| \leq \|w\|)$$

$$\leq -\eta\zeta^{2/3}(\zeta/4)\|w\|\,|a_w| + O(\exp(6H)\eta^2\|w\|\,|a_w|)$$
$$(\zeta\|w_{\text{sig}}\|/2 \leq \exp(6H)\|w_\perp\| + \|w_{\text{opp}}\| \leq \zeta\|w_{\text{sig}}\|)$$

$$\leq 0. \qquad\qquad\qquad (\eta \leq \zeta^3.)$$

Thus it follows that $\exp(6H)\|w_\perp^{(t+1)}\| + \|w_{\text{opp}}^{(t+1)}\| \leq \exp(6H)\|w_\perp^{(t)}\| + \|w_{\text{opp}}^{(t)}\| \leq \|w_{\text{sig}}^{(t)}\| \leq \zeta\|w_{\text{sig}}^{(t+1)}\|$.

$\square$

*Proof of Lemma E.13.* Our approach here is similar to the previous lemma. We will show that for any $\mu \in \{\pm\mu_1, \pm\mu_2\}$ from which $\tilde{\gamma}_\mu \geq \tilde{\gamma}_{\max} - 2\sqrt{\eta}$, we have

$$\tilde{\gamma}_\mu^{(t+1)} \leq \left(1 + \eta\frac{\sqrt{2}}{2}(1 + o(1))g_{\min}\right)\tilde{\gamma}_{\max}^{(t)}.$$

Then by Lemma E.14, for any $\mu$ for which $\tilde{\gamma}_\mu^{(t)} \leq \tilde{\gamma}_{\max}^{(t)} - 2\sqrt{\eta}$, we have

$$\tilde{\gamma}_\mu^{(t+1)} \leq \tilde{\gamma}_\mu^{(t)} + \sqrt{\eta} \leq \tilde{\gamma}_{\max}^{(t)} - \sqrt{\eta} \leq \tilde{\gamma}_{\max}^{(t)}.$$

For any neurons $w \in S_\mu$, using Lemma E.6 to bound $\nabla_w^{\text{cl}} L_\rho$, Lemma E.5 to bound $\|\nabla_w^{\text{cl}} L_\rho - \nabla_w L_\rho\|$, and Lemma C.12 to bound $\|\nabla_w \hat{L}_\rho - \nabla_w L_\rho\|$, we have with probability $1 - d^{-\omega(1)}$,

$$\|w_{\text{sig}}^{(t+1)}\|^2 \leq \|w_{\text{sig}}\|^2\left(1 + 2\eta\frac{|a_w|}{\|w_{\text{sig}}\|}\tau g_\mu(1 + o(1))\right).$$

Similarly, by the same 3 lemmas, we have with probability $1 - d^{-\omega(1)}$,

$$(a_w^{(t+1)})^2 = a_w^2\left(1 + 2\eta\frac{\|w_{\text{sig}}\|}{|a_w|}\tau g_\mu(1 + o(1))\right).$$

Thus

$$\mathbb{E}_{\rho_{t+1}}\mathbf{1}(w^{(t)} \in S_\mu)\|a_w^{(t+1)}w_{\text{sig}}^{(t+1)}\| \tag{E.4}$$

$$\leq \mathbb{E}_\rho\mathbf{1}(w^{(t)} \in S_\mu)\|a_w w_{\text{sig}}\|\left(1 + \eta\frac{|a_w|}{\|w_{\text{sig}}\|}\tau g_\mu(1 + o(1))\right)\left(1 + \eta\frac{\|w_{\text{sig}}\|}{|a_w|}\tau g_\mu(1 + o(1))\right)$$

$$\leq \mathbb{E}_\rho\mathbf{1}(w \in S_\mu)\|a_w w_{\text{sig}}\| + \eta\tau g_\mu(1 + o(1))\mathbb{E}_\rho\mathbf{1}(w \in S_\mu)\mu)\left(a_w^2 + \|w_{\text{sig}}\|^2\right)$$

$$\leq \mathbb{E}_\rho\mathbf{1}(w \in S_\mu)\|a_w w_{\text{sig}}\| + \eta\tau g_\mu(1 + o(1))\left(\mathbb{E}_\rho[\mathbf{1}(w \in S_\mu)2a_w^2] + O(\log(d)\eta H)\right)$$

$$\leq \mathbb{E}_\rho\mathbf{1}(w \in S_\mu)\|a_w w_{\text{sig}}\|\left(1 + 2\eta\tau g_\mu(1 + o(1))\right)$$

$$\text{(}|a_w| \leq \|w\| \leq (1 + o(1))\|w_{\text{sig}}\| \text{ since } w \in S.\text{)}$$

Next we show that $\frac{g_\mu}{g_{\min}}$ is small. We have

$$\frac{g_\mu}{g_{\min}} = \frac{\exp(-\gamma_\mu + \gamma_{\max})(1 + \exp(-\gamma_{\max}))}{1 + \exp(-\gamma_\mu)}$$

$$\leq \exp(-\gamma_\mu + \gamma_{\max}) \qquad\qquad (\gamma_\mu \leq \gamma_{\max})$$

$$\leq \exp(-\tilde{\gamma}_\mu + \tilde{\gamma}_{\max} - 4\zeta H) \qquad\qquad \text{(Eq.E.2)}$$

$$\leq \exp(2\sqrt{\eta} + 4\zeta H) \qquad\qquad \text{(Assumption that } \tilde{\gamma}_\mu \geq \tilde{\gamma}_{\max} - 2\sqrt{\eta})$$

$$\leq 1 + o(1).$$

Plugging this into Eq. E.4 yields

$$\tilde{\gamma}_\mu^{(t+1)} \leq \tilde{\gamma}_\mu^{(t)}\left(1 + 2\eta\tau g_{\min}(1 + o(1))\right),$$

as desired.

$\square$

*Proof of Lemma E.3.* The second and third items of Lemma E.3 follow immediately from Lemmas E.12 and E.13. To prove the first item, we first need to control the growth of $\mathbb{E}_\rho[\mathbf{1}(w \notin S)\|w\|^2]$. We compute

$$\mathbb{E}_{\rho_{t+1}}[\|w\|^2\mathbf{1}(w \notin S)] \leq \mathbb{E}_{\rho_t}[\|w\|^2\mathbf{1}(w \notin S)]\left(1 + 2\eta(1 + 2\zeta H)\tau g_{\max}\right)$$

$$\leq \zeta\tilde{\gamma}_{\min}^{(t)}\left(1 + 2\eta(1 + 2\zeta H)\tau g_{\max}\right) \qquad\qquad \text{(Lemma E.11)}$$

$$\leq \zeta\tilde{\gamma}_{\min}^{(t+1)}\frac{1 + 2\eta(1 + 2\zeta H)\tau g_{\max}}{1 + 2\eta(1 - 2\zeta H)\tau g_{\max}} \qquad\qquad \text{(Lemma E.12)}$$

$$\leq \zeta\tilde{\gamma}_{\min}^{(t+1)}\left(1 + 10\eta\zeta H\right).$$

Letting $\zeta' := \zeta(1 + 10\eta\zeta H)$ be the new signal-heavy parameter, it follows that

$$
\begin{aligned}
\zeta' &\leq \zeta(1 + 10\eta\zeta H) \\
&\leq \zeta_{T_1}(1 + 10\eta\zeta H)^{t-T_1} \\
&\leq \zeta_{T_1}(1 + 10\eta\zeta H)^{\frac{\zeta_{T_1}^{-1/2}}{\eta}} && (t - T_1 \leq \frac{\zeta_{T_1}^{-1/2}}{\eta} \text{ by assumption}) \\
&\leq \zeta_{T_1} e^{10\zeta H \zeta_{T_1}^{-1/2}} \\
&\leq \zeta_{T_1} e \\
&\leq \exp(-10H). && \text{(Lemma E.4)}
\end{aligned}
$$

Further, by Lemma C.13, we have that with probability $1 - d^{-\omega(1)}$, for all neurons, $|a_w^{(t+1)}| \leq \|w^{(t+1)}\|$, and

$$
\begin{aligned}
\mathbb{E}_{\rho_{t+1}}[\|w^{(t+1)}\|^2] - \mathbb{E}_{\rho_{t+1}}[(a_w^{(t+1)})^2] &\leq \mathbb{E}_{\rho_t}[\|w^{(t)}\|^2] - \mathbb{E}_{\rho_t}[(a_w^{(t)})^2] + 4\eta^2 \mathbb{E}_{\rho_t}(a_w^{(t)})^2 \\
&\leq 2\zeta H + 2\eta^2 H \\
&\leq 2\zeta' H.
\end{aligned}
$$

Finally, to bound $\mathbb{E}_{\rho_{t+1}}[(a_w^{(t+1)})^2]$, we have

$$
\begin{aligned}
\mathbb{E}_{\rho_{t+1}}[(a_w^{(t+1)})^2] &\leq \mathbb{E}_{\rho_{t+1}}[\|a_w^{(t+1)} w^{(t+1)}\|] && \text{(E.5)} \\
&\leq \mathbb{E}_{\rho_{t+1}}[\mathbf{1}(w^{(t+1} \in S)\|a_w^{(t+1)} w^{(t+1)}\|] + \mathbb{E}_{\rho_{t+1}}[\mathbf{1}(w^{(t+1} \notin S)\|a_w^{(t+1)} w^{(t+1)}\|] \\
&\leq \sum_{\mu \in \{\pm\mu_1, \pm\mu_2\}} \tilde{\gamma}_\mu^{(t+1)} + \zeta' \tilde{\gamma}_{\min}^{(t+1)} \\
&\leq 5\tilde{\gamma}_{\max}^{(t+1)}.
\end{aligned}
$$

We bound $\tilde{\gamma}_{\max}^{(t+1)}$ in the following claim.

**Claim E.16.**

$$
\tilde{\gamma}_{max}^{(T_1 + \zeta_{T_1}^{-1/2}/\eta)} \leq -\log(\zeta_{T_1})/20.
$$

*Proof.* Let $t^*$ be the last time at which $\tilde{\gamma}_{\max}$ is at most $-\log(\zeta_{T_1})/40$. If $t^* \geq T_1 + \zeta_{T_1}^{-1/160}/\eta$, then we are done. Suppose $t^* \leq T_1 + \zeta_{T_1}^{-1/160}/\eta$. Then by Lemma E.13, we have

$$
\begin{aligned}
\tilde{\gamma}_{\max}^{(T_1 + \zeta_{T_1}^{-1/160}/\eta)} &\leq \tilde{\gamma}_{\max}^{(t^*)} \left(1 + 2\eta(1 + o(1))\tau \max_{t^* \leq s \leq T_1 + \zeta_{T_1}^{-1/160}/\eta} g_{\min}^{(s)}\right)^{\zeta_{T_1}^{-1/160}/\eta} \\
&\leq \frac{-\log(\zeta_{T_1})}{40} \left(1 + 2\eta(1 + o(1))\tau \max_{t^* \leq s \leq T_1 + \zeta_{T_1}^{-1/160}/\eta} g_{\min}^{(s)}\right)^{\zeta_{T_1}^{-1/2}/\eta} \\
&\leq \frac{-\log(\zeta_{T_1})}{40} \exp\left(1.5\zeta_{T_1}^{-1/160} \max_{t^* \leq s \leq T_1 + \zeta_{T_1}^{-1/160}/\eta} g_{\min}^{(s)}\right) \\
&\leq \frac{-\log(\zeta_{T_1})}{40} \exp\left(2\zeta_{T_1}^{-1/160} \exp(-\tilde{\gamma}_{\max}^{(t^*)})\right) \\
&\leq \frac{-\log(\zeta_{T_1})}{40} \exp\left(2\zeta_{T_1}^{-1/160} \zeta_{T_1}^{\frac{1}{40}}\right) \\
&\leq \frac{-\log(\zeta_{T_1})}{20}.
\end{aligned}
$$

Here we have used Lemma E.14 to bound

$$
g_{\min}^{(s)} \leq 2\exp(-\gamma_{\max}^{(s)}) \leq 2\exp(-\tilde{\gamma}_{\max}^{(s)} + 2\zeta H),
$$

and thus

$$\max_{t^* \leq s \leq T_1 + \zeta_{T_1}^{-1/160}/\eta} g_{\min}^{(s)} \leq 2.1 \exp(-\tilde{\gamma}_{\max}^{(t^*)}).$$

$\square$

Plugging this claim into Eq. E.5 yields

$$\mathbb{E}_{\rho_{t+1}}[(a_w^{(t+1)})^2] \leq 5\tilde{\gamma}_{\max}^{(t+1)} \leq -\log(\zeta_{T_1})/20 \leq 2H.$$

The above computations, in addition to Lemma E.12 proves that $\rho_{t+1}$ is $(\zeta', H)$ signal-heavy with the heavy set $S$. $\square$

## F    PROOF OF THEOREM 3.1

We now prove Theorem 3.1 from Lemmas D.1 and Lemmas E.3. We restate the theorem and these two lemmas for the readers convenience.

**Theorem F.1.** *There exists a constant $C > 0$ such that the following holds for any $d$ large enough. Let $\theta := 1/\log(d)^C$. Suppose we train a 2-layer neural network with minibatch SGD as in Section 2.2 with a minibatch size of $m \geq d/\theta$, width $1/\theta \leq p \leq d^C$, step size $\eta \leq \theta$, and initialization scale $\theta$. Then for some $t \leq C\log(d)/\eta$, with probability $1 - d^{-\omega(1)}$, we have*

$$\mathbb{E}_{x \sim P_d}[\ell_{\rho_t}(x)] \leq (\log(d))^{-\Theta(1)}.$$

**Lemma D.1** (Output of Phase 1; Formal). *For any constants $c$ sufficiently large, and $C$ sufficiently large in terms of $c$, for any $d$ large enough, the following holds. Let $\theta := 1/\log(d)^C$.*

*Suppose we train a 2-layer neural network with minibatch SGD as in Section 2.2 with a minibatch size of $m \geq d/\theta^2$, width $1/\theta \leq p \leq d^C$, and step size $\eta \leq \theta$, and initialization scale $\theta$. Then with probability at least $1 - \theta$, after some $T_1 = \Theta(\log(d)/\eta)$ steps of minibatch SGD, the network $\rho_{T_1}$ satisfies:*

1. $\mathbb{E}_{\rho_{T_1}}[\|a_w w\|] \leq 1$;

2. $\mathbb{E}_{\rho_{T_1}}[\|w_\perp + w_{opp}\|^2] \leq 4\theta^2$;

3. *For all $\mu \in \{\pm\mu_1, \pm\mu_2\}$, on at least a $0.1$ fraction of the neurons, we have $\|w_{sig}\| \geq \log(d)^c\theta$ and $w_{sig}^T\mu > 0$.*

*Additionally,*

$$\mathbb{E}_{\rho_{T_1}}[\|w\|^2] \leq \mathbb{E}_{\rho_{T_1}}[|a_w|^2] + \sqrt{\eta},$$

*and for all neurons, we have $|a_w| \leq \|w\|$.*

**Lemma E.3** (Phase 2 Inductive Lemma; Formal). *Suppose $t \leq T_1 + \frac{\zeta_{T_1}^{-1/160}}{\eta}$. If a network $\rho_t$ is $(\zeta, H)$-signal heavy with heavy set $S$ and $\zeta \leq \zeta_{T_1}(1 + 10\eta\zeta H)^{t-T_1}$, then after one minibatch gradient step with step size $\eta \leq \zeta^3$, with probability $1 - d^{-\omega(1)}$,*

1. *$\rho_{t+1}$ is $(\zeta(1 + 10\eta\zeta H), H)$-signal heavy.*

2. *$\tilde{\gamma}_{min}^{(t+1)} \geq (1 + 2\eta\tau(1 - o(1))g_{max})\tilde{\gamma}_{min}^{(t)}$*

3. *$\tilde{\gamma}_{max}^{(t+1)} \leq (1 + 2\eta\tau(1 + o(1))g_{min})\tilde{\gamma}_{max}^{(t)}$.*

*Here $\zeta_{T_1}$ and $H$ are defined in Definition E.2, and $\tau = \frac{\sqrt{2}}{4}$.*

*Proof of Theorem 3.1.* Let $\rho_{T_1}$ be the network output by Lemma D.1. By Lemma E.4, we have that $\rho_{T_1}$ is $(\zeta_{T_1}, H)$-signal-heavy, where $\zeta_{T_1}$ and $H$ are defined in Definition E.2. Further, we have that $\tilde{\gamma}_{\min}^{(T_1)} \geq \zeta_{T_1}^{1/200}$.

Let us iterate Lemma E.3 for $T_2 := -\zeta_{T_1}^{-1/160}/\eta$ steps. We will show that $\tilde{\gamma}_{\min}^{(T_1+T_2)} \geq \frac{1}{200}\log(\zeta_{T_1}^{-1})$. If $\tilde{\gamma}_{\min}^{(t)}$ for $t \in [T_1, T_1 + T_2]$ ever exceeds $\frac{1}{200}\log(\zeta_{T_1}^{-1})$, then we are done since $\tilde{\gamma}_{\min}$ always increases. Suppose it does not exceed this value, and thus $g_{\min}^{(t)}$ is always at least $\zeta_{T_1}^{1/200}$.

Then we can show that $g_{\min}$ is relatively large, and thus by the second item of Lemma E.3, $\tilde{\gamma}_{\min}$ will grow quickly. Indeed for $d$ large enough:

$$
\begin{aligned}
\tilde{\gamma}_{\min}^{(T_1+T_2)} &\geq \tilde{\gamma}_{\min}^{(T_1)} \left(1 + 2\eta\tau(1 - o(1)) \min_{t\in[T_1,T_2]} g_{\min}^{(t)}\right)^{T_2} \\
&\geq \tilde{\gamma}_{\min}^{(T_1)} \left(1 + 2\eta\tau(1 - o(1))\zeta_{T_1}^{1/200}\right)^{T_2} \\
&\geq \tilde{\gamma}_{\min}^{(T_1)} \exp\left(\zeta_{T_1}^{-1/160}(\zeta_{T_1}^{1/199})\right) \\
&\geq \zeta_{T_1}^{\Theta(1)} \exp\left(\zeta_{T_1}^{-1/320}\right) &&\text{(Lemma E.4)} \\
&= \exp(\log^{\Theta(1)}(d)) &&(\zeta_{T_1} = \log^{-\Theta(1)}(d)) \\
&\geq \frac{1}{200}\log(\zeta_{T_1}^{-1}).
\end{aligned}
$$

Here in the second inequality we have lower bounded $g_{\min}^{(t)}$ by $(1 - o(1))\exp(-\tilde{\gamma}_{\min}^{(t)})$ using Lemma E.14.

Finally, we check the loss guarantee of the network $\rho_T$ for $T = T_1 + T_2$. Since $\rho_T$ is $(\zeta', H)$-signal heavy for $\zeta' \leq 2\zeta_{T_1}$ (see the proof of Lemma E.3), and by Definition E.2, $\zeta_{T_1}H = o(1)$, we have

$$
\begin{aligned}
\mathbb{E}_x \ell_{\rho_T}(x) &= -2\log\left(\frac{1}{1 + \exp(-f_\rho(x)y(x))}\right) \\
&\leq \mathbb{E}_x 2\exp(-f_\rho(x)y(x)) \\
&\leq \mathbb{E}_x 2\exp(-\gamma_{\min} + |f_\rho(x) - f_\rho(z)|) \\
&\leq 2\exp(-\gamma_{\min})\mathbb{E}_x \exp(|f_\rho(x) - f_\rho(z)|) \\
&\leq 2\exp(-\gamma_{\min})\mathbb{E}_\xi \exp(\mathbb{E}_\rho|a_w w_\perp^T \xi|) \\
&\leq 2\exp(-\tilde{\gamma}_{\min} + 2\zeta'H)\mathbb{E}_\xi \exp(\mathbb{E}_\rho|a_w w_\perp^T \xi|) &&\text{(Lemma E.14)} \\
&\leq 3\exp(-\tilde{\gamma}_{\min})\mathbb{E}_\xi \exp(\mathbb{E}_\rho|a_w w_\perp^T \xi|)
\end{aligned}
$$

Since $a_w w_\perp^T \xi$ is subguassian with norm $\Theta(\|a_w w_\perp\|)$, we have

$$
\mathbb{E}_\xi \exp(\mathbb{E}_\rho|a_w w_\perp^T \xi|) \leq \exp(\Theta(\mathbb{E}_\rho[\|a_w w_\perp\|^2])) \leq \exp\left(\Theta\left(\sqrt{\mathbb{E}_\rho[\|a_w\|^2]}\sqrt{\mathbb{E}_\rho[\|w_\perp\|^2]}\right)\right) \leq \exp(\Theta(H\zeta')),
$$

so since $\tilde{\gamma}_{\min} \geq \frac{1}{200}\log(\zeta_{T_1}^{-1}) = \Theta(\log\log(d))$, we have

$$
\mathbb{E}_x \ell_{\rho_T}(x) \leq 3\exp(-\tilde{\gamma}_{\min})\exp(\Theta(H\zeta)) \leq 4\exp(-\tilde{\gamma}_{\min}) = \log^{-\Theta(1)}(d).
$$

This yields the theorem. $\qquad\qquad\square$

# G LOWER BOUND OF $\tilde{\Theta}(d)$ FOR LEARNING THE XOR FUNCTION WITH ROTATIONALLY INVARIANT ALGORITHM.

**Proposition G.1.** *Suppose $A : (\{\pm1\}^d)^n \times \{\pm1\}^n \times \{\pm1\}^d \to \Delta(\{\pm1\})$ is an algorithm, which given $n$ labeled samples $X$ from the Boolean hypercube, and an additional unlabeled sample, outputs a distribution over labels. Suppose additionally that $A$ is rotationally invariant, that is, for an orthonormal rotation $U$, we have $A(UX, y, Ux) = A(X, y, x)$, so long as $UX$ and $Ux$ are on the hypercube.*

*Then if $n \leq d/\log^2(d)$, if $X = (x_1, \ldots, x_n)$ and $x$ are sample uniformly at random from the hypercube, and $i, j$ are sampled uniformly without replacement from $[d]$, we have*

$$
\mathbb{P}_{X,x,i,j}\mathbb{P}_{\hat{y}\sim A(X,y(X),x)}[\hat{y} \neq y(x)] \geq 0.03,
$$

*where $y(x) := -(x^T e_i)(x^T e_j)$, and $y(X) \in \{\pm1\}^n$ denotes the labels of the entire set $X$.*

*Proof.* Let $H = \{\pm 1\}^d$ denote the Boolean Hypercube. Let $Q := \{h \in H : h^T x_i = x^T x_i \forall i \in [n]\}$. Because the geometry of the set of points $(x_1, \ldots, x_n, x)$ is the same as the geometry of the points $(x_1, \ldots, x_n, q)$, for each $q \in Q$, for each such $q$, there exists (at least 1) rotation $U$ such that $Ux = q$ and $Ux_i = x_i$ for $i \in [n]$.

Thus by the rotational invariance of $A$, for all $q \in Q$, we must have that $A(X, y, q)$ is the same distribution. Let us denote this distribution by $D_{X,y,Q}$.

We make the following claim:

**Claim G.2.** *For some sufficiently small constant c, if $|Q| \geq 2^{d-cd}$, we have that with probability at least $0.25$ over $i, j$, for any distribution $D$, $\mathbb{P}_{q \sim Q, \hat{y} \sim D}[y(q) \neq \hat{y}] \geq 0.15$.*

*Proof.* Without loss of generality let $i = 1$. It suffices to prove that with probability $0.25$ over $j$, at most a $0.85$ fraction of the points in $Q$ have the same label.

Suppose this was not true. For $j \in [d]$, let $v_j$ denote the majority of $y(q)$ over $q \in Q$ (break ties arbitrarily), where the labeling function $y$ is determined by $i$ and $j$. Let $P \subset [d]$ be the set of all $j$ such that $\mathbb{P}_{q \sim Q}[y(q) = v_j] > 0.85$, such that we must have $|P| \geq 0.75d$. To abbreviate, for $j \in P$, define $y_j(q) := -(q^T a_1)(q^T a_j)$. Then for at least a $0.1$ fraction of points in $q \in Q$, we must have $\mathbb{P}_{j \sim P}[y_j(q) = v_j] \geq 0.8$. Indeed, if not,

$$\mathbb{E}_{j \sim P} \mathbb{E}_{q \sim Q} \mathbf{1}(y_j(q) = v_j) = \mathbb{E}_{q \sim Q} \mathbb{E}_{j \sim P} \mathbf{1}(y_j(q) = v_j) \leq 0.1 * 1.0 + 0.9 * 0.8 < 0.85.$$

Now we consider how many points $h$ there are in $H$ satisfying $\mathbb{P}_{j \sim P}[y_j(h) = v_j] \geq 0.8$. Consider first the number of such points over $H_1 := \{h \in H : e_1 = 1\}$. Having $\mathbb{P}_{j \sim P}[y_j(h) = v_j] \geq 0.8$ implies that $-h_j = v_j$ for at least $0.8|P| \geq 0.55d$ coordinates. By a standard Chernoff bound, the number of such points is smaller that $2^{d-1} \exp(-Cd)$ for some constant $C$. The same holds for the set of all points where $h^T e_1 = -1$. Thus the total number of points in $H$ satisfying $\mathbb{P}_{j \sim P}[y_j(h) = v_j] \geq 0.6$ is at most $2^d \exp(-Cd)$. Thus if $|Q| > \frac{1}{0.1} 2^d \exp(-Cd)$ we have reached a contradiction. This holds for $|Q| \geq 2^{d-cd}$ for $c$ small enough. $\qquad \square$

Now fix $X$ and partition $H$ into $K$ disjoint sets $Q_1, Q_2, \ldots Q_K$ such that that for any $k$, for all $q \in Q_k$, the projection of $q$ onto $X$ is the same. We have

$$\mathbb{P}_{x \sim H, i, j}[\hat{y} \neq y(x)] \geq \sum_{k=1}^{K} \frac{|Q_k|}{2^d} \mathbb{P}_{i,j} \mathbb{P}_{q \sim Q_k}[\hat{y} \neq y(q)]$$

$$\geq \sum_{k=1}^{K} \frac{|Q_k|}{2^d} (0.25)(0.15) \mathbf{1}(|Q_k| \geq 2^{d(1-c)})$$

Now since $K \leq (d+1)^n = 2^{o(d)}$ (indeed, the projection onto each $x_i$ can only take on $d+1$ different integer values $-2d, -2(d-1), \ldots, 2d$), we have that

$$\sum_{k=1}^{K} |Q_k| \mathbf{1}(|Q_k| \geq 2^{d(1-c)}) \geq 2^d - 2^{d(1-c)} K \geq (1 - o(1)) 2^d.$$

Thus for any $X$, we have that

$$\mathbb{P}_{\hat{y} \sim A(X, y(X), x)}[\hat{y} = y(x)] \mathbb{P}_{x \sim H, i, j}[\hat{y} \neq y(x)] \geq (0.25)(0.15)(1 - o(1)) \geq 0.03.$$

The result follows. $\qquad \square$