# OpenReview forum: "SGD Finds then Tunes Features in Two-Layer Neural Networks with near-Optimal Sample Complexity: A Case Study in the XOR problem"
_ICLR.cc/2024/Conference — ICLR 2024 spotlight_

### Official Review · Reviewer_Y9yq · 2023-10-30

**Soundness:** 3 good
**Presentation:** 3 good
**Contribution:** 3 good
**Rating:** 8
**Confidence:** 4

**Summary:**

This paper proves that 2-layer neural networks trained with SGD can learn the XOR function on the boolean hypercube using $\tilde{O}(d)$ samples. Unlike prior work, the analysis doesn't require layer-wise training or the use of random biases, which is thanks to carefully tracking the evolution of the weights in two distinct phases and exploiting the symmetries of the problem. The analysis reveals that in the first phase the weight vectors of a significant number of neurons align with the XOR features, thus feature learning occurs. In the second phase, the neurons that recovered the relevant features will grow and dominate the prediction of the network, thus the XOR function is learned.

**Strengths:**

* This paper presents a novel analysis of feature learning in two-layer neural networks and distinguishes itself from prior work by considering a standard (unmodified) SGD training algorithm and achieving sample complexity that is almost optimal among the class of rotationally invariant algorithms.

* Plenty of intuition is provided on the mechanism of the proofs, and the two-phase dynamics could potentially be used for analyzing feature learning in other problems as well.

**Weaknesses:**

* The considered problem can be a bit restrictive, and for example could be generalized to learning $k$-sparse parities. This is not a major issue however, as this paper is one of the first attempts at providing a feature learning analysis of *standard* SGD without additional modifications.

* As the authors have also pointed out, the batch size chosen is relatively large compared to practical settings, and it is interesting to know whether a smaller batch size can also learn XOR with near-optimal sample complexity.

**Questions:**

* Perhaps the authors could highlight the $\varepsilon$-dependency (test error) of the rate in Theorem 3.1. This question is less relevant in some high-dimensional settings where the goal is only to learn to constant accuracy, but may still provide some insights given that the authors consider the specific problem of learning XOR.

* It seems that the algorithm spends more time in the first phase $\Theta(\log(d) / \eta)$ than the second phase $\Theta(\log\log(d) / \eta)$. This might be interesting to point out, as it is also in line with observations from Ben Arous et al., 2021, who show that SGD spends most of its time in the *weak recovery* phase.

* As a minor comment, it might be helpful to mention the scale of $\zeta$ inside Lemma 4.1 to make the statement of the lemma more self-contained.
***
Ben Arous et al. "Online stochastic gradient descent on non-convex losses from high-dimensional inference." JMLR 2021.

---

> ### Author Response · Authors · 2023-11-20
> **Response to Review**
>
> We thank the reviewer for their positive assessment and insightful comments and questions!
>
> - **Convergence rate of test error**.
> A minor revision to our calculation (which is now in the uploaded revision) yields that $\gamma_{min} = \Omega(\log\log(d)),$ and so the loss is $O(1/polylog(d))$.
>
>   For $\epsilon$ smaller than $O(1/polylog(d))$, we expect the number of iterations should be roughly $\tilde{O}(1/\epsilon)$, matching the convergence rate of logistic regression. However, we do not prove this in our paper because it would require modifying Lemma E.3, the Phase 2 inductive lemma.
>
> - Indeed, our findings are similar to Ben Arous et al. 2021, in that that there is a long search ("weak-recovery") phase in which the directions of the multi-index function are learned but the loss does not decrease, and then after that phase, the loss decays quickly. We will point this comparison in the introduction.
>
> - We have revised to mention the scale of $\zeta$ (it is $1/\log^c(d)$).

---

> > ### Comment · Reviewer_Y9yq · 2023-11-23
> >
> > Thank you for your responses. I keep my positive evaluation of the work.

---

### Official Review · Reviewer_xNLA · 2023-10-31

**Soundness:** 3 good
**Presentation:** 3 good
**Contribution:** 3 good
**Rating:** 8
**Confidence:** 4

**Summary:**

This paper studies the classification problem of learning XOR functions on Boolean hypercube. The setting is standard that using 2-layer ReLU neural networks with vanilla online minibatch SGD on logistic loss. The authors show that neural nets can achieve $o(1)$ population loss with $d \mathrm{polylog}(d)$ samples. This is near-optimal sample complexity in the sense that all rotationally invariant algorithms need at least $d/\mathrm{polylog}(d)$ samples. The proof shows a 2-stage dynamic that in the first stage the neurons learn the features and grow large to reduce loss in the second stage. The analysis approximates the real dynamics with dynamics induced by certain surrogate loss.

**Strengths:**

-	Understanding the optimization for neural networks in feature learning regime is an important and interesting problem. This paper takes a step to go beyond single-index models to more complex XOR functions.
-	The paper is overall well-written and easy to follow. A proof sketch is provided to illustrate some important proof ideas.
-	The paper shows that vanilla online minibatch SGD on 2-layer neural networks can learn XOR function efficiently with $d \mathrm{polylog}(d)$ samples. This seems to be new in literature.
-	The proof techniques introduced in this paper also seem to be novel, especially the observation that the variance of gradient on the signal entries can be shown to be much smaller than the signal (eq (4.5)).

**Weaknesses:**

-	The results essentially still rely on the small initialization to learn features at the beginning, which is similar to many previous works.
-	As discussed in Appendix A, it might be difficult to generalize the techniques to other settings (though it is understandable).

**Questions:**

-	What is the initialization scale in Theorem 3.1? On the top of page 5, the scale is $\theta$. However, it becomes $\theta/\sqrt{p}$ in the statement of Theorem 3.1.
-	I was wondering why stepsize $\eta$ is required to be not very small ($>1/\mathrm{poly}(d)$) in Theorem 3.1. Are there any fundamental difficulties in the proof that that stepsize cannot be too small?
-	I wonder what the order of $o(1)$ test loss is in Theorem 3.1. Is it $1/\mathrm{polylog}(d)$ or $1/\mathrm{poly}(d)$? It seems not very clear in the main text. Or in general, what is the dependency on test loss $\epsilon$ for sample complexity and running time？

Minor:

-	Above eq 4.1: ‘ie.’ -> ‘i.e.,’
-	In Definition 4.2, after ‘Here we have defined’: $w_{sig}^T$ -> $w_{sig}^T x$

---

> ### Author Response · Authors · 2023-11-20
> **Response to Review**
>
> We thank the reviewer for their positive assessment and insightful comments and questions!
>
> **Initialization scale**. We apologize for the typo - the initialization scale should always be $\theta$; we have fixed this in the revision. We note that our assumption on a $1/polylog(d)$ initialization is much weaker than comparable results that leverage a small initialization (eg. Abbe et al. 2022 initializes the first layer weights at 0 exactly --- see "Choice of Hyperparameters" in section E.1). Similarly to Abbe et al 2022 (and other works), our small initialization keeps the output of the network close to 0 for the first phase of training, which indeed simplifies the analysis. However, we note that because our initialization is still within log(d) factor of 1, we still have to be careful .
>
> It may be possible to remove the small initialization assumption by claiming that since the second layer weights are mean-0 and random, the network would still whp output a value close to 0 throughout phase 1, though we leave this question to future work.
>
> **Step size**
> If the step size is smaller, we would need more iterations of SGD (and thus more samples) to learn the XOR function. Escaping the saddle takes $T = \Theta(\log(d))$ time, where $T = \eta t$ and $t$ is the number of iterations. This is because learning the signal mimics a power iteration where the signal components of the neurons grow like $w_{sig}^{(t)} \approx w_{sig}^{(0)} \exp(\eta t)$.
>
> **Test loss**
> An exact bound on the loss can be found by looking at the details of the proof of Theorem 3.1 (at the end of Section F), which repeatedly applies Lemma E.3, the Phase 2 inductive lemma. As stated, the calculation currently given there (which was not optimized to give the best dependence on $d$) shows that we obtain $\gamma_{min} = \Omega(\log\log\log\log(d)),$ and so by the final calculation in this proof, we have that the loss is $O(\exp(-\gamma_{min}) = O(1/\log\log\log(d))$.
>
> A minor revision to our calculation (which is now in the uploaded revision) yields that $\gamma_{min} = \Omega(\log\log(d)),$ and so the loss is $O(1/polylog(d))$.
>
> For $\epsilon$ smaller than $O(1/polylog(d))$, we expect the number of iterations should be roughly $(1/\epsilon)$, matching the convergence rate of logistic regression. However, for simplicity, we do not prove this in our paper because it would require modifying Lemma E.3, the Phase 2 inductive lemma.

---

> > ### Comment · Reviewer_xNLA · 2023-11-23
> >
> > Thanks for the response. I will keep my score.

---

### Official Review · Reviewer_chEd · 2023-11-01

**Soundness:** 3 good
**Presentation:** 4 excellent
**Contribution:** 3 good
**Rating:** 8
**Confidence:** 3

**Summary:**

This paper studies the problem of learning a degree-2 parity function (XOR) with a 2-layer neural network with ReLU activations under the logistic loss function. Particularly, the training is done with mini-batch SGD where both layers are trained simultaneously. It is further shown that the sample complexity is $\tilde O(d)$ which is also shown to be optimal for rotationally-invariant algorithms.

In the proof, it is shown that SGD increases the weights incident to the parity bits for a fraction of neurons. Afterwards, these neurons get reinforced making the learning possible.

**Strengths:**

- Including a rather extensive literature review and also mentioning the limitations of their analyses and avenues for future work (e.g., discussion about batch size and Gaussian setting).
- Providing the analysis in the case that both layers are trained jointly with reasonable learning rates. Also, there are no uncommon modifications to the algorithm such as projections, or changes in the learning rate, ...
- Interestingly they discuss that joint training of the layers may have a regulating effect that makes the learning possible (see Remark 4.3).

**Weaknesses:**

Nothing in particular, the limitations of the work are clearly stated in the paper.

**Questions:**

- Q1. What modifications do you expect to be necessary so that similar analysis can be carried out for higher degree parities?
- Q2. Using $\ell_2$ loss is already discussed in the appendix but for the Gaussian setting. What are the challenges of using $\ell_2$ (or other loss functions) for the Boolean parity analysis?

Remark. It would be beneficial for readers if further discussion about the proof techniques (and their comparison with recent work) is included.

---

> ### Author Response · Authors · 2023-11-20
> **Response to Review**
>
> We thank the reviewer for their positive assessment and insightful comments and questions!
>
> **Modifications for higher degree parities**.
> To represent a higher order parity via relu-activated neurons, you need more neurons, and the roles of the neurons are not ``exchangable'' (e.g. exhibit a certain symmetry with respect to the data distribution). In the 2-parity problem, this means we expect $w_{sig}$ to grow in each of these 4 directions at approximately the same rate. This phenomenon may not occur for higher order parities. For example, for a 3-parity $y = -x_1x_2x_3$, the rate of learning the direction $(1, 1, 1)$ may be differently from the rate of learning the $(1, 1, -1)$ direction.
>
> Thus to carry out this analysis for higher order parities you would need to:
> 1) Determine the set of neurons necessary to represent the parity. [Note, there may be multiple options]
> 2) Use a mean-field like analysis to predict the rates at which these different directions will be learning, near initialization. This step can also confirm which set of neurons will represent the parity, if there were many options in Phase 1.
> 3) If the neurons grow at the same rate, use an analysis similar to our paper with more directions to learn.
> 4) If the neurons do not grow at the same rate, strengthen Phase 2 of our analysis so that it can hold for \Theta(\log(d)) iterations. Then modify our analysis to look like this:
> - Phase 1: Learn the directions learned fastest.
> - Phase 1.5 (A combination of our phase 1 and 2, repeat as needed): Learn the directions which are learned slower, while showing that the signal in the already-learned directions is not forgotten.
> - Phase 2. Grow the signal components of the neurons, as in our Phase 2.
>
> **$\ell_2$ loss for Boolean data.**
> We expect using $\ell_2$ loss could work, though it would require modifying our Phase 2 to yield a slightly stronger result. Namely, we would not just need to show that the margin in each of the 4 cluster directions is large. Rather, we would need to show that the components learned in each of the 4 directions are exactly equal. Presumably the same sort of Phase 2 "signal-heavy" inductive hypothesis should work. There may be some additional challenges due to the fact that the signal may *decrease* if it grows too large.
>
> **Discussion of Comparison of Proof Techniques**
> Due to space limitations, we will add this comparison to the Discussion section in Appendix A. We will discuss:
> - How our decomposition into two phases of analysis is similar to other works, such as Mahankali et al. 2023, Ben Arous et al. 2021.
> - How we leverage the small initialization of the network, similarly (but not to the same extent as) other works such as Abbe et al. 2022.

---

> > ### Comment · Reviewer_chEd · 2023-11-22
> > **Thank you for the answers**
> >
> > I thank the reviewers for their answers and comments. I will maintain my score.

---

### Official Review · Reviewer_kd4Q · 2023-11-01

**Soundness:** 2 fair
**Presentation:** 2 fair
**Contribution:** 2 fair
**Rating:** 6
**Confidence:** 2

**Summary:**

The authors investigate the problem of learning a XOR function on Boolean data using two-layer ReLU networks trained with minibatch stochastic gradient descent (SGD). By investigating the training dynamics of SGD by jointly training the two layers on the logistic loss they identify two phases: a first phase in which the output network is close to zero ("signal-finding"), and a second stage ("signal-heavy") )in which the signal components stay larger than their counterparts. They provably show that with dpolylog(d) samples the training algorithm reaches a o(1) population error.

**Strengths:**

The authors analyze an important problem: learning a target function that depends only on two relevant directions with two-layer neural networks. The training algorithm considers joint updates of the first and second layer, this protocol is more closely connected to real-life settings than the layerwise training explored in many publications in the multi-index learning literature.

**Weaknesses:**

The manuscript's biggest weakness is the clarity of the exposition and its relationship with previous works. A more in-depth discussion is provided below.

**Questions:**

- I believe these works are relevant for the comparison with the literature in this context [1,2,3]. In [1] the authors show that even one step of GD in the $n=O(d)$ regime with a large learning rate beats kernel methods, but only a single index approximation of the target is learned. in [2] the authors expand the findings of [1] to the $n=O(d^l)$ scenario and investigate the learning of multi-index functions with a finite number of GD steps by extending the staircase condition to Gaussian data and a large batch setting. Finally, [3] proves that a smoothed version of SGD matches correlation statistical query lower bound: if the target function has information exponent $k$, smoothed-SGD attains $O(d^{\frac{k}{2}})$ sample-complexity.

-  Ben Arous et al. [2021] is present in the references but not in the text. What is the relationship between the sample complexity obtained in this work for XOR (multi-index) and for $k = 2$ (information-exponent) single index target functions?

- It is just briefly mentioned how the findings differs from the results of Ben Arous et al. [2022], a better comparison must be done. Could the author comment on this point?

-  Suggestion of relevant reference for the Saddle-to-Saddle dynamics [4].

- The analysis of the joint training of the two layers is nice and welcome. Could the author comment on the consequences in terms of transfer learning? After the training procedure, has the first layer learned the relevant feature subspace such that other tasks dependent on the subspace U could be learned?


[1] Jimmy Ba, Murat A Erdogdu, Taiji Suzuki, Zhichao Wang, Denny Wu, and Greg Yang. High-dimensional asymptotics of feature learning: How one gradient step improves the representation [2022].

[2] Yatin Dandi, Florent Krzakala, Bruno Loureiro, Luca Pesce, and Ludovic Stephan. How two-layer neural networks learn: one (giant) step at a time [2023].

[3] Alex Damian, Eshaan Nichani, Rong Ge, and Jason D. Lee. Smoothing the Landscape Boosts the Signal for SGD: Optimal Sample Complexity for Learning Single Index Models [2023].

[4] Arthur Jacot, François Ged, Berfin Şimşek, Clément Hongler, and Franck Gabriel. Saddle-to-saddle dynamics in deep linear networks: Small initialization training, symmetry, and sparsity [2021].

---

> ### Author Response · Authors · 2023-11-20
> **Added requested references.**
>
> We thank the reviewer for their suggestions, particularly to enhance our comparisons to related work.
>
> **We have added reference to [1, 2, 3, 4] in the the introduction:**
> - We discuss [1, 2] in the subsection "Learning Multi-Index Functions via Neural Networks" of the related work, where we discuss works that take 1 step of GD on the first layer. We note that [2] was not available online at the time of our submission.
> - We add [3] and Ben Arous et al. 2021 in our discussion of learning single-index functions in the related work (middle of page 3). Thank you for catching that we missed citing Ben Arous et al. 2021 in the body.
> - We have added [4] to the discussion at the end of the  subsection "Learning Multi-Index Functions via Neural Networks" in the related work, where we discuss differences between kernel and rich regimes.
>
> **What is the relationship between the sample complexity obtained in this work for XOR (multi-index) and for (information-exponent) single index target functions?** As explained in the related work, it is conjectured that the sample complexity of SGD on a 2-layer NN is $\tilde{\Theta}(d^{\max(L-1, 1)})$, where $L$ is the leap complexity, which is $2$ for XOR. *In the case of single-index functions, the leap complexity equals the information exponent*, and thus the sample complexity of learning XOR via SGD matches that of learning a single index function with information exponent $k = 2$. *We have clarified this connection between leap complexity and information exponent in the related work*.
>
> **Better comparison to Ben Arous et al. 2022**. Ben Arous et al. 2022 considers a setting of high-dimensional SGD where there are a *constant* number of summary statistics which are sufficient to track the key features of the SGD dynamics and the loss. There are several differences between the setting they study and our, and the reasons why their tools cannot be applied to our setting.
> - Their work can be applied to a 2-layer network with *constant* width, but not one with width growing with $d$, as we study, because this would not have a constant number of summary statistics.
> - They study the XOR problem in a setting where the cluster separation $\lambda$ grows to infinity.
> - Their coupling between high dimensional SGD and a low-dimension SDE only holds for $\Theta(1)$ time, which is not enough time to learn the XOR function --- because the initialization of the network is near a saddle, $\Theta(\log(d))$ time is necessary.
> *We have clarified this comparison in a footnote where Ben Arous et al. 2022 is referenced.*
>
> Yes, transfer learning could occur after our training procedure since the directions of $U$ will be learned by the first layer weights. This would follow from a similar proof that to Damian et al. 2022. We do not include this because the reasoning would be the same, and is fairly straightforward.
>
> The reviewer also says that a key weakness is the "clarity of exposition", but there are no details provided. If the reviewer has any suggestions that would be appreciated. Otherwise, would the reviewer consider increasing their score?

---

> > ### Comment · Reviewer_kd4Q · 2023-11-21
> > **Thank you for the rebuttal**
> >
> > I sincerely thank the authors for the rebuttal, I believe these changes will improve the clarity of the exposition and I increased my score to 6.

---

### Meta-Review · Area_Chair_NGCC · 2023-12-05

**Metareview:**

This paper analyzes the training of two-layer neural networks with standard minibatch SGD. The target function is an XOR function y = -x_ix_j. The paper shows that the neural network can learn useful features for the XOR problem and gives a good characterization of the training dynamics. The reviewers all agree that the paper addresses an important problem with interesting techniques.

**Justification For Why Not Higher Score:**

To me the results are interesting but not super surprising. There has been other related works (although there are certainly some new ideas in this paper). The reviews also don't sound as excited as their score.

That said, I think the paper is still on the borderline and it wouldn't be wrong to give it an oral.

**Justification For Why Not Lower Score:**

To me this paper is clearly above the bar for spotlight.

---

### Decision · Program_Chairs · 2024-01-16

Accept (spotlight)